# Evaluating the link between the sulphur-rich Laacher See volcanic eruption and the Younger Dryas climate anomaly

James U.L. Baldini[1], Richard J. Brown[1], and Natasha Mawdsley[1]

[1]Department of Earth Sciences, University of Durham, Durham, DH1 3LE, UK.

*Correspondence to*: James Baldini (james.baldini@durham.ac.uk)

**Abstract.** The Younger Dryas is considered the archetypal millennial-scale climate change event, and identifying its cause is fundamental for thoroughly understanding climate systematics during deglaciations. However, the mechanisms responsible for its initiation remain elusive, and both of the most researched triggers (a meltwater pulse or a bolide impact) are controversial. Here we consider the problem from a different perspective, and explore a hypothesis that Younger Dryas climate shifts were catalysed by the unusually sulphur-rich 12.880 ka BP eruption of the Laacher See volcano (Germany). We use the most recent chronology for GISP2 ice core ion dataset from the Greenland ice sheet to identify a large volcanic sulphur spike coincident with both the Laacher See eruption and the onset of Younger Dryas-related cooling in Greenland (i.e., the most recent abrupt Greenland millennial-scale cooling event, Greenland Stadial-1; 'GS-1'). Previously published lake sediment and stalagmite records confirm that the eruption's timing was indistinguishable from the onset of cooling across the North Atlantic, but that it preceded westerly wind repositioning over central Europe by ~200 years. We suggest that the initial short-lived volcanic sulphate aerosol cooling was amplified by ocean circulation shifts and/or sea ice expansion, gradually cooling the North Atlantic region and incrementally shifting the mid-latitude westerlies to the south. The aerosol-related cooling probably only lasted 1-3 years, and the majority of Younger Dryas-related cooling may have been due to the sea ice-ocean circulation positive feedback, which was particularly effective during the intermediate ice volume conditions characteristic of ~13 ka BP. We conclude that the large and sulphur-rich Laacher See eruption should be considered a viable trigger for the Younger Dryas. However, future studies should prioritise climate modelling of high latitude volcanism during deglacial boundary conditions in order to test the hypothesis proposed here.

## 1 Introduction

The Younger Dryas (YD) climate anomaly occurred during the last deglaciation and is often described as a brief return to near-glacial conditions in northern Europe. Research now indicates that the YD was indeed characterised by cold conditions across the North Atlantic and Europe (Carlson et al., 2007; von Grafenstein et al., 1999), but also by a southward-shifted westerly wind belt over Europe (Bakke et al., 2009; Brauer et al., 2008; Lane et al., 2013a; Baldini et al., 2015b), a southward-shifted Intertropical Convergence Zone (Shakun et al., 2007; Chiang and Bitz, 2005; Bahr et al., 2018), increased moisture across the southwest of North America (Polyak et al., 2004; Asmerom et al., 2010), and potential warming in parts of the Southern Hemisphere (Bereiter et al., 2018; Kaplan et al., 2010).

A common explanation for the YD involves meltwater-induced weakening of Atlantic Meridional Overturning Circulation (AMOC) (Berger, 1990; Alley, 2000; Broecker et al., 2010; Johnson and McClure, 1976; Schenk et al., 2018). Initial support for this theory included elevated $\delta^{18}O$ values in Gulf of Mexico sediment dating from the early YD (implying that meltwater was rerouted elsewhere) (Broecker et al., 1988; Flower and Kennett, 1990; Teller, 1990). The meltwater was originally proposed to have travelled to the North Atlantic via the St Lawrence Valley, but results have so far revealed only limited geological evidence for a massive flux of freshwater coincident with the YD initiation (Broecker, 2006a; Rayburn et al., 2011). The freshwater pulse may have followed another route to the ocean, and other research has proposed the Mackenzie Valley (Condron and Winsor, 2012; Murton et al., 2010) as a possible alternative, and the Fennoscandian Ice Sheet as an alternate source (Muschitiello et al., 2015). Very recent surface exposure ages suggest that the route to the North Atlantic from Lake Agassiz was indeed free of ice before the YD initiation (Leydet et al., 2018), coinciding with evidence for freshening of the Gulf of St Lawrence (Levac et al., 2015), illustrating that meltwater could indeed have followed the St Lawrence Valley route to the North Atlantic. Nevertheless, the issue concerning the routing of a meltwater pulse illustrates a key uncertainty inherent to the meltwater pulse hypothesis. Another complication is that efforts to model the AMOC response to freshwater inputs under deglacial boundary conditions have yielded equivocal results (e.g., Meissner, 2007). Consequently, although meltwater forcing remains the most widely researched cause of the YD, it is not universally accepted. Broad agreement does exist that AMOC weakening was associated with the YD onset (Lynch-Stieglitz, 2017), but the driver of this weakening remains unclear.

Other recent research has proposed that a large impact event, or events, over North America may have triggered the YD (Firestone et al., 2007; Kennett et al., 2009). The Younger Dryas Impact Hypothesis (YDIH) is supported by the discovery of iridium, shocked quartz, platinum, and millions of tons of impact spherules at the YD boundary layer (Kennett et al., 2009; Wittke et al., 2013; Wu et al., 2013). Wolbach et al. (2018a) use charcoal and soot evidence from 152 sedimentary sequences and elevated concentrations of platinum at 26 sites (including Greenland) to argue that a cometary impact at the beginning of the YD triggered the largest wildfires of the Quaternary, consuming 9% of global terrestrial biomass. The YDIH has proven remarkably controversial, and different researchers suggest either terrestrial origins for the same evidence, that the evidence is not unique to the YD boundary, or that the YD boundary layer was misidentified (e.g., Pinter et al., 2011; van Hoesel et al., 2015; Holliday, 2015; Scott et al., 2017; Daulton et al., 2017). Finally, still other research proposes that the YD resulted from internal oceanic processes, and that no external forcing was required to trigger the observed climate shifts (Sima et al., 2004).

Here we summarise but do not argue extensively for or against any of these established hypotheses. Instead, we investigate the hypothesis that the YD was triggered by the ~12.9 ka BP eruption of the Laacher See volcano, located in the East Eifel Volcanic Field (Germany). Earlier research briefly alluded to the eruption as a possible causative mechanism for the YD (e.g., Berger, 1990; Bogaard et al., 1989). However, because the meltwater pulse hypothesis was already popularised, and because the effects of volcanic eruptions on climate would escape detailed quantification until after the 1991 Pinatubo eruption, the concept of the

~12.9 ka BP Laacher See eruption as a YD trigger never gained traction. Importantly, the concept was effectively dismissed after lacustrine evidence across central Europe appeared to indicate that the YD's clearest expression appeared ~200 years after the Laacher See Tephra within the same sediments (e.g., Brauer et al., 2008; Brauer et al., 1999a; Hajdas et al., 1995). For example, Schmincke et al. (1999) state that "The Younger Dryas cooling period clearly was not triggered by LSE as formerly thought because it started ca. 180 years after the eruption", reflecting the accepted sequence of events at that time. However, the identification of the Vedde Ash chronostratigraphic unit within Meerfelder Maar (Germany) lake sediments has improved correlations with Greenland ice core records, which contain the same ash (Lane et al., 2013a). This revised chronological framework now strongly suggests that the 12.880 ka BP Laacher See eruption was in fact synchronous with cooling associated with the YD onset (i.e., the most recent abrupt Greenland millennial-scale cooling event, Greenland Stadial-1; 'GS-1'), but preceded major atmospheric circulation shifts over central Europe (Rach et al., 2014).

Additionally, we utilise ion data from the Greenland ice core GISP2 (Zielinski et al., 1997) on the most recent chronological model for the core (the GICC05modelext chronology (Seierstad et al., 2014)) to identify a large volcanogenic sulphate spike whose timing coincides with both the Laacher See eruption and the initiation of GS-1 related cooling. We suggest that the initial, short-lived volcanogenic aerosol cooling triggered a sea-ice/AMOC positive feedback that caused both basin-wide cooling and the dynamical climate shifts most closely associated with the YD. Should future research confirm a volcanically-forced YD, it would strengthen our understanding of the processes operating during deglaciations, and underscore the strong role that volcanic aerosols can play in moderating global climate. Although more research is clearly needed to thoroughly investigate this hypothesis, the apparent coincidence of a large, very sulphur-rich eruption with the beginning of YD cooling is compelling, and well worth exploring further.

## 2 Background

Laacher See volcano, Germany, is situated in the East Eifel Volcanic Field, which is part of the West European rift system (Baales et al., 2002; de Klerk et al., 2008) (Figure 1). The Laacher See Eruption (LSE) occurred at ~12.880 ± 0.040 ka BP based on the position of tephra within regional varved lake sequences (e.g., Wulf et al., 2013; Brauer et al., 1999a; Lane et al., 2015; Bronk Ramsey et al., 2015), consistent with radiocarbon (12.934 ± 0.165 cal ka BP (Baales et al., 2002)) and $^{40}$Ar/$^{39}$Ar (12.9 ± 0.50 ka BP (van den Bogaard, 1995)) ages for the eruption; the absolute age of the eruption is therefore well constrained. The eruption was one of the largest in Europe during the late Quaternary and dispersed over 20 km$^3$ of pumice and ash over >230,000 km$^2$ of central Europe and beyond (Baales et al., 2002; Bogaard and Schmincke, 1985). The eruption consisted of alternating Plinian and phreatomagmatic phases; Plinian columns exceeded 20 km height and injected ash and volcanic gas into the stratosphere (Harms and Schmincke, 2000). Direct effects of the LSE included ash deposition, acid rain, wildfires, and

increased precipitation, all of which could have affected the local and far-field ecology and cultures at the time (de Klerk et al., 2008; Baales et al., 2002; Engels et al., 2016; Engels et al., 2015).

The LSE discharged up to 6.3 km$^3$ of unusually sulphate-rich, evolved phonolite and sulphide-rich mafic phonolite magma from a strongly compositionally zoned reservoir (Harms and Schmincke, 2000). Estimating $SO_2$ release from eruptions preserved in the geologic record is difficult and relies on petrologic comparisons between the sulphur content of melt inclusions (the magma at depth) and the sulphur content of glass in erupted products (the degassed magma) (Devine et al., 1984; Scaillet et al., 2004; Textor et al., 2003; Vidal et al., 2016). Petrologic methods can lead to underestimations of total sulphur release by up to two orders of magnitude relative to satellite data (Gerlach et al., 1996; Scaillet et al., 2004). For example, satellite-derived estimates of the 1991 Pinatubo eruption's $SO_2$ content (15–20 megatonnes (Mt)), are considerably higher than the 0.11 Mt petrologic estimate (Sheng et al., 2015), and similar discrepancies between petrologic and satellite-derived estimates of $SO_2$ emissions exist for other modern eruptions (Scaillet et al., 2004) (Figure 2). The consensus is that this excess sulphur is contained in a sulphur-rich vapour phase that is released prior to and during the eruption (Gerlach et al., 1996; Wallace, 2001). Petrologic studies by Harms and Schmincke (2000) suggest that the LSE released 3.8 Mt of $SO_2$, but considering a sulphur-rich vapour phase they concluded that the eruption released at least 20 megatons of $SO_2$ into the stratosphere. Textor et al. (2003) suggested that the eruption released between 6.76 and 104.8 Mt of $SO_2$, when considering uncertainties regarding the oxidation state of the vapour phase. Despite these unknowns, the amount of $SO_2$ released by the LSE was considerably higher than the petrologic estimates, and Schmincke et al. (1999) speculated a maximum release of 300 Mt $SO_2$, assuming the same relationship between petrologic and observed values as for the Pinatubo eruption. We utilise a similar but more comprehensive approach, taking the mean value of the relationship between petrologic and observed values of the 16 non-basaltic explosive eruptions catalogued by Shinohara (2008) plus estimated values for the large 1257 AD Samalas eruption (Vidal et al., 2016) to estimate that the LSE released ~83 Mt $SO_2$, although the range in possible values is substantial. A variety of estimated values clearly exist, but the eruption almost certainly released more $SO_2$ than the Pinatubo eruption (~20 Mt), and possibly even more than the 1815 Tambora eruption (~70 Mt) (Figure 2). However, unlike the Pinatubo and Tambora eruptions, sulphate aerosols produced by the LSE were largely restricted to the Northern Hemisphere (NH) (Figure 3), leading to strong cooling in the stratosphere that affected the NH disproportionately (Graf and Timmreck, 2001).

Both Pinatubo and the LSE were Magnitude 6 (M6) eruptions, where 'magnitude' is a measure of eruption size referring to the amount of material erupted (Deligne et al., 2010) on a logarithmic scale. However, the cooling effects of a volcanic eruption are controlled by the amount of sulphur released, and not necessarily the eruption size (Rampino and Self, 1982). In general, magnitude and erupted sulphur amounts are well correlated (Oppenheimer, 2003; Carn et al., 2016), and therefore magnitude is often used as a surrogate for sulphur yield. However, variability of almost three orders of magnitude exists in the amount of sulphur released amongst equivalently sized explosive eruptions (Carn et al., 2016), and consequently eruption size is not the only predictor of total sulphur released. In the case of the LSE, all the existing evidence suggests that it was anomalously

enriched in sulphur relative to its magnitude (Baales et al., 2002; Scaillet et al., 2004), and that it therefore should have produced significant NH cooling.

## 3 Results and Discussion

### 3.1 The timing of the Laacher See Eruption relative to the Younger Dryas

Key research using European lake sediment archives containing the Laacher See Tephra (LST) suggested that the LSE preceded the YD onset by ~200 years (Hajdas et al., 1995; Brauer et al., 1999a). For example, the LST appears very clearly in the Meerfelder Maar sediment core (Germany), and it does indeed appear to predate the YD (e.g., Brauer et al., 1999a). However, the recent discovery of the Icelandic Vedde Ash in Meerfelder Maar sediments has revised the relative timing of key climate archives around the North Atlantic (Lane et al., 2013a). It is now apparent that the clearest hydroclimatic

expression of the YD in central Europe lags Greenland cooling associated with GS-1 by 170 years (Figure 4). Specifically, several lake sediment cores from different latitudes have revealed that westerly wind repositioning was time transgressive, with winds gradually shifting to the south following the initiation of GS-1 cooling (Bakke et al., 2009; Lane et al., 2013a; Brauer et al., 2008; Muschitiello and Wohlfarth, 2015).

Although the LSE preceded the most clearly expressed dynamical climatic change associated with the YD in central Europe, its timing ($12.880 \pm 0.040$ ka BP) is indistinguishable from the Greenland temperature decrease leading into GS-1, beginning at $12.870 \pm 0.138$ ka BP (Rach et al., 2014; Steffensen et al., 2008; Rasmussen et al., 2014) (GS-1 is defined as starting at 12.846 in the NGRIP record, but abrupt cooling predates this by ~24 years) (Figure 4). In central Europe, hydrogen isotope ratios of land plant-derived lipid biomarkers from Lake Meerfelder Maar confirm that North Atlantic atmospheric cooling

began at ~12.880 ka BP (Rach et al., 2014), synchronous with both the onset of Greenland GS-1 related cooling and the LSE, but preceding the atmospheric response associated with meridionally displaced westerly winds at the same site (importantly, in MFM the first data point suggesting cooling occurs immediately above the LST (Rach et al., 2014)). This observation is further supported by speleothem $\delta^{18}$O records from La Garma and El Pindal caves in northern Spain (Baldini et al., 2015b; Moreno et al., 2010), Swiss lake sediment $\delta^{18}$O records (Lotter et al., 1992), Lithuanian lake sediment trace element profiles

(Andronikov et al., 2015), and Swiss lake organic molecule-based (BIT and TEX$_{86}$) temperature reconstructions (Blaga et al., 2013), all showing North Atlantic cooling beginning immediately after the LSE (Figure 4). Some lacustrine proxy records suggest that cooling began a few years after the LSE (e.g., the Gerzensee $\delta^{18}$O stack (van Raden et al., 2013)). The existence of a short lag between the temperature signal and the recording of that signal in some lacustrine archives may reflect differences in the type of archive used (i.e., terrestrial versus lacustrine), a decadal-scale residence time of groundwater feeding certain

lakes, or differences in moisture source regions for different lakes. The trigger responsible for the cooling must coincide with

(or predate) the earliest evidence for the cooling, and a lag must logically exist between the forcing and any later response (rather than the effect preceding the cause).

Recent work used volcanic marker horizons to transfer the GICC05modelext to the GISP2 ice core (previously on the Meese/Sowers chronology) (Seierstad et al., 2014), thereby improving comparisons with records of volcanism (Figure 5) and facilitating a direct comparison between the GISP2 volcanic sulphate record (Zielinski et al., 1997) and both the GISP2 and the NGRIP $\delta^{18}$O records (Figure 5). For the interval around the Younger Dryas initiation, very little chronological difference between this timescale and IntCal13 exists (Muscheler et al., 2014), suggesting the timescale is robust. We find that a large sulphate spike at 12.867 ka BP in the GISP2 record on the GICC05modelext timescale is contemporaneous with the LSE's timing based on $i$) dates for the eruption based on varved lake deposits (12.880 ± 0.05 ka BP (Lane et al., 2015)) and $ii$) layer counting from the Vedde Ash within NGRIP (Figure 5); we therefore ascribe this sulphate spike to the Laacher See eruption. Earlier research also briefly considered this spike based on absolute dating (Brauer et al., 1999b), but because of the concept that the LSE preceded the YD boundary by ~200 years, concluded that an earlier sulphate spike most likely represented the LSE. Utilising the GICC05modelext chronology (Seierstad et al., 2014) as well as recent layer counts within the MFM sediment (Lane et al., 2013a), we suggest that the sulphate spike at 12.867 ka BP represents the LSE, and that the earlier one tentatively identified by Brauer et al. (1999b) may represent a small eruption of the Icelandic volcano Hekla, consistent with the interpretation of Muschitiello et al. (2017). Although the coincidence between our identified sulphate spike, the date of the LSE, and the onset of North Atlantic cooling associated with the YD is compelling, detailed tephrochronological analyses are required to definitively ascribe this sulphate spike to the LSE.

## 3.2 A complex response of climate to volcanic eruptions

Explosive volcanic eruptions can inject large amounts of sulphur-rich gases as either $SO_2$ or $H_2S$ into the stratosphere, which are gradually oxidised to form sulphuric acid vapour (Rampino and Self, 1984). Within a few weeks, the vapour can condense with water to form an aerosol haze (Robock, 2000), which is rapidly advected around the globe. Once in the atmosphere, sulphate aerosols induce summer cooling by scattering incoming solar radiation back to space (Timmreck and Graf, 2006; Baldini et al., 2015a). An eruption's sulphur content, rather than its explosivity, determines most of the climate response (Robock and Mao, 1995; Sadler and Grattan, 1999). Furthermore, recent research has also highlighted the role of volcanogenic halogens, such as Cl, Br, and F, in depleting stratospheric ozone (Cadoux et al., 2015; Klobas et al., 2017; Kutterolf et al., 2013; Vidal et al., 2016; LeGrande et al., 2016). Ozone absorbs solar ultraviolet and thermal infrared radiation, and thus depletion in ozone can result in surface cooling, and would act to amplify the radiative effects of volcanogenic sulphate aerosols (Cadoux et al., 2015).

Although the radiative effects associated with volcanic aerosols are reasonably well understood, the systematics of how volcanic eruptions affect atmospheric and oceanic circulation are less well constrained. Research suggests that volcanic eruptions affect a wide variety of atmospheric phenomenon, but the exact nature of these links remains unclear. For example, Pausata et al. (2015) used a climate model to conclude that high latitude NH eruptions trigger an El Niño event within 8-9 months by inducing a hemispheric temperature asymmetry leading to southward Intertropical Convergence Zone (ITCZ) migration and a restructuring of equatorial winds. The model also suggests that these eruptions could lead to AMOC shifts after several decades, consisting of an initial 25-year strengthening followed by a 35-year weakening, illustrating the potential for climate effects extending well beyond sulphate aerosol atmospheric residence times. Several modelling studies based on historical data suggest that eruptions may strengthen AMOC (Ottera et al., 2010; Swingedouw et al., 2014; Ding et al., 2014) but also increase North Atlantic sea ice extent for decades to centuries following the eruption due to the albedo feedback and reductions in surface heat loss (Ding et al., 2014; Swingedouw et al., 2014). Other models suggest that AMOC may intensify initially, but then weaken after about a decade (Mignot et al., 2011). A modelling study by Schleussner and Feulner (2013) suggested that volcanic eruptions occurring during the last millennium increased Nordic Sea sea ice extent, which weakened AMOC and eventually cooled the entire North Atlantic Basin. Importantly, Schleussner and Feulner (2013) concluded that short-lived volcanic aerosol forcings triggered "a cascade of sea ice-ocean feedbacks in the North Atlantic, ultimately leading to a persistent regime shift in the ocean circulation". Other research finds that North Atlantic sea ice growth following a negative forcing weakened oceanic convection and northward heat export during the Little Ice Age (Lehner et al., 2013). Quantifying the long-term influences of single volcanic eruptions is confounded by the effects of subsequent eruptions and other factors (e.g., solar variability, El Niño events), which can overprint more subtle feedbacks. For example, model results looking at recent eruptions found evidence that different types of eruptions can either constructively or destructively interfere with AMOC strength (Swingedouw et al., 2014). Therefore, despite increasingly clear indications that volcanic eruptions have considerable long-term consequences for atmospheric and oceanic circulation, the full scale of these shifts is currently not well understood even over the last two millennia, and is essentially unknown under glacial boundary conditions.

However, indications that volcanism may have had particularly long-term climate effects during the last glacial do exist. Bay et al. (2004) found a very strong statistical link between evidence for Southern Hemisphere (SH) eruptions and rapid Greenland warming associated with Dansgaard-Oeschger (DO) events, although no causal mechanism was proposed. More recent research determined that every large, radiometrically-dated SH eruption (five eruptions) across the interval 30-80 ka BP occurred within dating uncertainties of a DO event (Baldini et al., 2015a). The same research also found a strong statistical correlation between large NH volcanic eruptions and the onset of Greenland stadials over the same 30-80 ka BP interval (Baldini et al., 2015a), and proposed that during intermediate ice volume conditions, a positive feedback involving sea ice extension, atmospheric circulation shifts, and/or AMOC weakening may have amplified the initial aerosol injection. This positive feedback continued to operate until it was superseded by another forcing promoting warming at high latitudes in the NH, or an equilibrium was reached. This also appears consistent with observations during the YD: *i)* a sulphur-rich NH

volcanic eruption, *ii)* long-term NH high latitude cooling, *iii)* NH mid-latitude westerly wind migration to the south, and *iv)* slow recuperation out of the event following rising 65°N insolation and/or a SH meltwater pulse.


The LSE erupted twice the volume of magma as, and potentially injected considerably more sulphur into the stratosphere than, the 1991 AD Pinatubo eruption (Baales et al., 2002; Schmincke, 2004). The Pinatubo eruption cooled surface temperatures by 0.5°C globally, and up to 4°C across Greenland, Europe, and parts of North America, for two years following the eruption (Schmincke, 2004), while aerosols persisted in the stratosphere for ~three years (Diallo et al., 2017). An existing model

suggests that the LSE created a sulphate aerosol veil wrapping around the high northern latitudes (Graf and Timmreck, 2001) (Figure 3). The model suggests that NH summer temperatures dropped by 0.4°C during the first summer following the eruption, though it assumes that the eruption released substantially less $SO_2$ (15 Mt $SO_2$ (Graf and Timmreck, 2001)) than current maximum estimates (104.8 Mt $SO_2$ (Textor et al., 2003)) (Figure 2), which rival the sulphur amount delivered to the stratosphere by the 1815 AD Tambora eruption. Actual aerosol-induced cooling may far exceed these estimates, a perspective

supported by Bogaard et al. (1989) who estimate that the environmental impacts of the eruption of the sulphur-rich phonolite melt may even have exceeded those of the far larger (but silicic) Minoan eruption of Santorini (~1613 BC). It does not seem unreasonable that an eruption of this size, sulphur content, and geographic location, occurring during a transitional climate state, could have catalysed a significant climate anomaly.

**3.3 The nature of the positive feedback**

Volcanogenic sulphate aerosols typically settle out of the atmosphere within three years; aerosol-induced cooling alone therefore cannot explain the YD's extended duration. We suggest that aerosol forcing related to the LSE initiated North Atlantic cooling, which consequently triggered a positive feedback as proposed for earlier Greenland stadials (Baldini et al., 2015a). Existing evidence strongly suggests that North Atlantic sea ice extent increased (Bakke et al., 2009; Baldini et al., 2015b;

Broecker, 2006b; Cabedo-Sanz et al., 2013) and AMOC weakened (Broecker, 2006b; McManus et al., 2004; Bondevik et al., 2006) immediately after the GS-1 onset, and therefore both could have provided a powerful feedback. The feedback may have resided entirely in the North Atlantic, and involved sea ice expansion, AMOC weakening, and increased albedo, as previously suggested within the context of meltwater forcing (Broecker et al., 2010). Alternatively, it has been noted that hemispherically asymmetrical volcanic sulphate loadings induced ITCZ migration away from the hemisphere of the eruption (Ridley et al.,

2015; Hwang et al., 2013; Colose et al., 2016), and it is possible that these ITCZ shifts forced wholesale shifts in atmospheric circulation cells. This hypothesised mechanism is broadly consistent with that advanced by Chiang et al. (2014) where a forcing at a high northerly latitude subsequently drives southward ITCZ migration, which then affects global atmospheric circulation. Within the context of GS-1, LSE aerosol-related NH cooling could have shifted the ITCZ to the south, thereby expanding the NH Polar Cell and shifting the NH Polar Front to the south. Sea ice tracked the southward shifted Polar Front, resulting in

more NH cooling, a weakened AMOC, and a further southward shift in global atmospheric circulation cells (Baldini et al., 2015a). Such a scenario is consistent with recent results based on tree ring radiocarbon measurements suggesting that GS-1

was not caused exclusively by long-term AMOC weakening, but instead was forced by NH Polar Cell expansion and southward NH Polar Front migration (Hogg et al., 2016).

Our hypothesis that the YD was triggered by the LSE and amplified by a positive feedback is further supported by modelling results suggesting that a combination of a moderate negative radiative cooling, AMOC weakening, and altered atmospheric circulation best explain the YD (Renssen et al., 2015). AMOC consists of both thermohaline and wind-driven components, and atmospheric circulation changes can therefore dramatically affect oceanic advection of warm water to the North Atlantic. Recent modelling suggests that reduced wind stress can immediately weaken AMOC, encouraging southward sea ice

expansion and promoting cooling (Yang et al., 2016), illustrating a potential amplification mechanism following an initial aerosol-induced atmospheric circulation shift. Twentieth Century instrumental measurements further support this by demonstrating that westerly wind strength over the North Atlantic partially modulates AMOC (Delworth et al., 2016).

Initiation of the proposed positive feedback would require volcanic aerosols to remain in the atmosphere for at least one

summer season. Evidence based on the seasonal development of vegetation covered by the LST suggests that the LSE occurred during late spring or early summer (Schmincke et al., 1999), and varve studies similarly suggest a late spring or early summer eruption (Merkt and Muller, 1999). Available evidence therefore suggests that the eruption occurred just prior to maximum summer insolation values, maximising the potential scattering effects of the volcanogenic sulphate aerosols. Even if it were a winter eruption, for historical eruptions similar in magnitude to the Laacher See eruption, aerosols remained in the atmosphere

longer than one year, regardless of the eruption's latitude. For example, aerosols remained in the atmosphere for ~three years after the Pinatubo eruption (15°N, 120°E) (Diallo et al., 2017), which probably released considerably less $SO_2$ than the LSE (Figure 2). Measurable quantities of aerosols remained in the atmosphere for approximately three years even after the 1980 Mount St. Helen's eruption (46°N, 122°W) (Pitari et al., 2016), which produced only 2.1 Mt $SO_2$ (Baales et al., 2002; Pitari et al., 2016) and erupted laterally (Eychenne et al., 2015). In short, the LSE eruption probably occurred during the late spring or

early summer, but even if the eruption were a winter eruption, the LSE's aerosols would have certainly persisted over at least the following summer, with the potential to catalyse the positive feedback we invoke.

It is worth highlighting that a similar mechanism may have also contributed to Little Ice Age cooling (Miller et al., 2012), with research suggesting that a coupled sea ice/AMOC mechanism could extend the cooling effects of volcanic aerosols by over

100 years during the Little Ice Age (Zhong et al., 2011). This perspective is supported by modelling results suggesting that a large volcanic forcing is required to explain Little Ice Age cooling (Slawinska and Robock, 2017). Lehner et al. (2013) identify a sea ice/AMOC/atmospheric feedback that amplified an initial negative radiative forcing to produce the temperature pattern characterising the Little Ice Age. However, Rahmstorf et al. (2015) argue that no identifiable change in AMOC occurred during the Little Ice Age, and Thornalley et al. (2018) similarly suggest that AMOC only weakened near the end of the Little Ice Age.

This implies that the drivers of Little Ice Age cooling were related to sea-ice, atmosphere, or oceanic (other than the AMOC,

e.g., horizontal subpolar gyre circulation (Moreno-Chamarro et al., 2017)) changes, although this requires further research to confirm. Large volcanic eruptions in 536, 540, and 547 AD are hypothesised to have triggered a coupled sea ice/ocean circulation feedback that led to an extended cold period (Buntgen et al., 2016). Recent research also highlights the possibility that volcanism followed by a coupled sea ice/ocean circulation positive feedback triggered hemispheric-wide centennial- to millennial-scale variability during the Holocene (Kobashi et al., 2017). If a feedback was active following volcanic eruptions during the 6[th] Century, the Little Ice Age, and the Holocene, the intermediate ice volume and transitional climate characteristic of the last deglaciation should have amplified any volcanic aerosol forcing. This perspective is consistent with previous observations, including those of Zielinski et al. (1996) who noted that when the climate system is in a state of flux it is more sensitive to external forcing, and that any post-volcanic cooling would be longer lived. Importantly, Rampino and Self (1992) stated "Volcanic aerosols may also contribute a negative feedback during glacial terminations, contributing to brief episodes of cooling and glacial readvance such as the Younger Dryas Interval". Our results are entirely consistent with this perspective, and here we highlight a sulphur-rich volcanic eruption whose timing coincided with the onset of YD-related cooling. However, despite increasingly tangible evidence that eruptions can affect ocean circulation and sea ice extent, the exact nature of any positive feedback is still unclear. Future research should prioritize the identification and characterisation of this elusive, but potentially commonplace, feedback that amplifies otherwise subtle NH temperature shifts.

### 3.4 Sensitivity to ice volume

Magnitude 6 (M6) eruptions such as the LSE are large but not rare; for example, over the last two thousand years, twelve M6 or larger eruptions are known (Brown et al., 2014), but none produced a cooling event as pronounced as the YD. Abrupt millennial scale climate change is characteristic of glacial intervals, and the forcing responsible only appears to operate during intervals when large ice sheets are present. The lack of large-scale and prolonged climate responses to external forcing over the Holocene implies either the absence of the forcing (e.g., lack of large meltwater pulses), a reduced sensitivity to a temporally persistent forcing (e.g., volcanism), or the existence of an internal forcing mechanism particular to glacial climates. The subdued climate response to volcanic eruptions over the recent past could reflect low ice volume conditions and the absence (or muting) of the requisite positive feedback mechanism. However, millennial-scale climate change was also notably absent during the Last Glacial Maximum (~20-30 ka BP), implying that very large ice sheets also discourage millennial-scale climate shifts.

The apparent high sensitivity of the climate system to millennial-scale climate change during times of intermediate ice volume is well documented (e.g., Zhang et al., 2014; Zhang et al., 2017), and here we investigate this further by examining the timing of Greenland Stadials relative to ice volume (estimated using Red Sea sea level (Siddall et al., 2003)). The timings of 55 stadial initiations as compiled in the INTIMATE (INTegration of Ice core, MArine, and TErrestrial) initiative (Rasmussen et al., 2014) are compared relative to ice volume, and indeed a strong bias towards intermediate ice volume conditions exists, with 73% of the millennial scale cooling events occurring during only 40% of the range of sea level across the interval from 0-120

ka BP (Figure 6). The distribution of events suggests that the most sensitive conditions are linked to ice volume associated with a sea level of -68.30 m below modern sea level. This intermediate ice volume was commonplace from 35-60 ka BP, and particularly from 50-60 ka BP. However, over the interval from 0-35 ka BP, these ideal intermediate ice volume conditions only existed during a short interval from 11.8-13.7 ka BP, and optimal conditions were centred at 13.0 ka BP (Figure 6). These results are broadly consistent with previous research (e.g., Zhang et al., 2014; Zhang et al., 2017), but the timing and duration

of the most sensitive interval of time, and the likelihood that a forcing produces a longer-term cooling event, may ultimately depend on a complex interplay between ice volume and atmospheric $CO_2$ (Zhang et al., 2017). Atmospheric $pCO_2$ during the YD initiation was relatively high (~240 ppmv) and could therefore affect the timing of ideal conditions for abrupt climate change in conjunction with ice volume, but their precise interdependence is still unclear. Finally, a frequency distribution of the sea level change rate associated with each stadial indicates that whatever mechanism is responsible for triggering a stadial

operates irrespective of whether sea level (i.e., ice volume) is increasing or decreasing (Figure 6b). Because active ice sheet growth should discourage meltwater pulses, this observation seemingly argues against meltwater pulses as the sole trigger for initiating stadials.

### 3.5. Comparison with the climate response to the Toba supereruption

Comparing the LSE to another well-dated Quaternary eruption occurring under different background conditions provides information regarding the nature of the relevant forcings. The magnitude > 8 (i.e., ~100x greater than the LSE) Toba supereruption occurred approximately ~74 ka BP and was the largest eruption of the Quaternary (Brown et al., 2014). Despite its size, the climate response to the eruption remains vigorously debated, with some researchers suggesting that the eruption triggered the transition to Greenland Stadial 20 (GS-20) (Polyak et al., 2017; Baldini et al., 2015a; Carolin et al., 2013;

Williams et al., 2009) but others arguing for only a small climate response (Haslam and Petraglia, 2010). However, it does not appear that the eruption triggered a sustained global volcanic winter (Lane et al., 2013b; Svensson et al., 2013), and therefore globally homogenous long-term aerosol-induced cooling probably did not occur. The Toba supereruption occurred during an orbitally-modulated cooling trend with intermediate ice volume (-71 m below modern sea level, very near the sea level associated with the most sensitive ice volume conditions (-68.3 m below modern sea level, as suggested here)), and it is

possible that the eruption expedited the transition to GS-20 through a combination of an initial short-lived aerosol-induced cooling amplified by a positive feedback lasting hundreds of years.

In this analysis we use the timing of the Toba eruption relative to the NGRIP record suggested by Svensson et al., (2013) (their NGRIP sulphate spike 'T2'), but the timing remains uncertain. If future research revises the timing relative to NGRIP, this

would necessarily affect the interpretations discussed below. Assuming the Toba eruption was responsible for the sulphate spike T2, the eruption was followed by high latitude NH cooling accompanied by southward ITCZ migration and Antarctic warming, consistent with sea ice growth across the North Atlantic and a weakened AMOC. The cooling rates into GS-1 (the YD) after the LSE and GS-20 after the Toba eruption are nearly identical (Figure 7), consistent with the possibility that a

similar process was responsible for the cooling in both instances. The cooling rate may reflect the nature of the post-eruptive positive feedback, which was potentially independent of the initial radiative cooling effects of the two eruptions, provided that they were large enough to trigger a feedback. The reconstructed climate responses following the LSE and Toba eruptions diverge after the achievement of maximum cooling. In the case of Toba, cold conditions persisted for ~1.7 ka before the very rapid temperature increase characteristic of the onset of Greenland Interstadial (an abrupt Greenland warming event) 19 (GI-19) (Figure 7). GS-1 ended after ~1.2 ka, but unlike GS-20 which had stable cold temperatures throughout, $\delta^{18}O$ (Baldini et al., 2015b; Steffensen et al., 2008) and nitrogen isotope data (Buizert et al., 2014) suggest that the coldest conditions around the North Atlantic occurred early within GS-1 and were followed by gradual warming. The LSE and Toba eruptions occurred under different orbital configurations, and the contrasting topologies characteristic of GS-1 and GS-20 may reflect differing insolation trends. At ~13 ka BP, summer insolation at 65°N was increasing, rather than decreasing as during the Toba eruption ~74 ka BP. The Toba eruption may have triggered a shift to the insolation-mediated baseline (cold) state, whereas the LSE may have catalysed a temporary shift to cold conditions opposed to the insolation-driven warming trend characteristic of that time interval, resulting in a short-lived cold event followed by gradual warming. Radiative cooling events in the Southern Hemisphere (e.g., a SH eruption, an Antarctic meltwater pulse, etc.) may have abruptly terminated both GS-1 and GS-20; the GS-1 termination is broadly consistent with the putative timing of Meltwater Pulse 1B (MWP-1B) (Ridgwell et al., 2012), whereas any SH trigger for the termination of GS-20 is unidentified. MWP-1B (or another SH cooling event) may have cooled the SH and strengthened AMOC, prompting northward migration of the ITCZ and NH mid-latitude westerlies to achieve equilibrium with high insolation conditions, thereby rapidly reducing sea ice extent and warming Greenland, but this requires further research, particularly because the source, duration, timing, and even existence of MWP-1B are still debated. Reduced oceanic salt export within the North Atlantic subtropical gyre, characteristic of stadials, may have preconditioned the North Atlantic toward vigorous AMOC following the initial migration of atmospheric circulation back to the north (Schmidt et al., 2006).

The residence time of aerosols within the atmosphere is not critical within the context of this model provided the positive feedback is activated, and a sufficiently high aerosol-related cooling over only one summer and one hemisphere could suffice. The feedback's strength may also depend on the amount of hemispheric temperature asymmetry caused by the eruption. Consequently, the high latitude LSE may actually have induced a stronger hemispheric temperature asymmetry than the low latitude Toba eruption, although the Toba eruption would have resulted in considerably more overall aerosol-induced cooling. The long-term (e.g., hundreds to thousands of years) climate response may depend on the climate background conditions; if a NH eruption occurs during an orbitally-induced cooling trend (as may have been the case for Toba), the eruption may catalyse cooling towards the insolation-mediated baseline. If the NH eruption occurs during rising insolation and intermediate ice volume (e.g., Laacher See), we suggest that any post-eruption feedback will continue until it is overcome by a sufficiently high positive insolation forcing, or a SH cooling event. Although the size of the two eruptions differed by two orders of magnitude, the residence time of the respective aerosols in the stratosphere was probably broadly similar: ~three years for the LSE (similar

to that of the Pinatubo eruption) and ~10 years for Toba (Timmreck et al., 2012; Robock et al., 2009). Although modelling results suggest that Toba resulted in ~10 degrees C of cooling (Timmreck et al., 2012), compared to probably around ~1 degree

Celsius for the LSE, we argue that both eruptions were able to cool climate sufficiently to trigger sea ice growth, weaken AMOC, and thereby initiate the positive feedback.

A useful way of visualising this is to consider two extreme scenarios: *i*) very high $CO_2$ concentrations and no ice (e.g., the Cretaceous) and *ii*) very low $CO_2$, low insolation, and very high ice volume (e.g., the Last Glacial Maximum). An eruption the

size of the LSE would probably not significantly affect climate beyond the atmospheric residence time of the sulphate aerosols during either scenario. Under the background conditions characteristic of the first scenario, insufficient aerosol forcing would occur to trigger ice growth anywhere, and consequently no positive feedback would result. In the case of the second scenario, the eruption would cause ice growth and cooling for the lifetime of aerosols in atmosphere before conditions returned to the insolation-mediated equilibrium. In contrast to these extreme scenarios, an injection of volcanogenic sulphate aerosols into the

NH atmosphere would most effectively trigger a feedback during intermediate $CO_2$ and ice volume conditions; in other words, during a transition from one ice volume state to another when the climate system was in a state of disequilibrium. Under these conditions, we suggest that activation of the feedback would occur even if the sulphate aerosols settled out of the atmosphere after just one year. Therefore, although the Toba and Laacher See eruptions were of very different magnitudes, the nature of the positive feedback would depend largely on the background conditions present during the individual eruptions. We argue

that both eruptions were large enough, and contained enough sulphur, to activate the positive feedback under their respective background conditions. The positive feedback's strength was then controlled by background conditions rather than eruption magnitude (or amount of sulphur released). Furthermore, we predict that the more asymmetric the hemispheric distribution of the aerosols, the stronger the feedback. For these reasons, the long-term (centennial- to millennial-scale) climate response following the extremely large but low latitude Toba eruption may have approximated those of the far smaller but high latitude

Laacher See eruption. Therefore, the long-term climate repercussions to very large volcanic eruptions may not consist of prolonged radiative cooling (i.e., 'volcanic winter') but rather of geographically disparate dynamical shifts potentially not proportional to eruption magnitude.

### 3.6 Response to other late Quaternary eruptions

The strongest argument against the LSE contributing to YD climate change is that several other similar or larger magnitude eruptions must have occurred during the last deglaciation, and that therefore the LSE was not unusual. However, the LSE *i)* was unusually sulphur-rich, *ii)* was high-latitude, and *iii)* coincided with ideal ice volume conditions. It is in fact the only known sulphur-rich, high latitude eruption coinciding with the most sensitive ice volume conditions during the last deglaciation. Early work noted that because the 1963 AD Agung eruption was very sulphur-rich, it forced cooling of a similar

magnitude as the 1815 Tambora eruption, despite Tambora ejecting considerably more fine ash (~150 times more) into the upper atmosphere (Rampino and Self, 1982). Although estimates of total sulphur released vary, it is clear that the LSE was

also unusually sulphur-rich (probably releasing more than Agung (Figure 2)), so had the potential to exert a considerable forcing on climate disproportionate to its magnitude. Furthermore, other eruptions may also have contributed to climate change but current chronological uncertainties preclude establishing definitive links. For example, the ~14.2 ka BP Campi Flegrei eruption may have caused some cooling, but this is impossible to confirm with eruption age uncertainties of ±1.19 ka.

Other large sulphate spikes exist in the GISP2 sulphate record, and in fact the two spikes at 12.6 ka BP and 13.03 ka BP are both larger than the potential LSE sulphate spike. The size of the sulphate spike is not necessarily proportional to the size of the eruption, and the sulphate spike at 13.03 ka BP is likely related to the small eruption of Hekla (Iceland) (Muschitiello et al., 2017), which deposited sulphate in the nearby Greenland ice but was climatologically insignificant. Similarly, the spike at 12.6 ka BP may also reflect the influence of a small but proximal unknown eruption. Alternatively, a large eruption of Nevado de Toluca, Mexico, is dated at 12.45 ± 0.35 ka BP (Arce et al., 2003), and could correspond to the large sulphate spike at 12.6 ka BP within the GISP2 ice core. The eruption was approximately the same size as the LSE, so the lack of long-term cooling may reflect a different climate response due to the eruption's latitude, which caused a more even distribution of aerosols across both hemispheres. Although there is no long-term cooling following the 12.6 ka BP sulphate spike, it is associated with a short cooling event; therefore the lack of long-term cooling at 12.6 ka BP may simply reflect the fact that temperatures had already reached the lowest values possible under the insolation and carbon dioxide baseline conditions characteristic of that time.

### 3.7 Compatibility with other hypotheses

A substantial amount of research has now characterised the YD climate, and several hypotheses exist attempting to explain the excursion. The most well-researched hypothesis involves a freshwater pulse from the large proglacial Lake Agassiz substantially reducing AMOC strength (Alley, 2000; Broecker, 1990; Broecker et al., 2010). Weakened AMOC would result in a colder North Atlantic, a southward displaced ITCZ, and Antarctic warming, all features characteristic of the YD. However, direct geological evidence for a meltwater pulse originating from Laurentide proglacial lakes coincident with the YD onset remains elusive (Broecker et al., 2010; Petaev et al., 2013). Recent research suggests that meltwater from Lake Agassiz could have had an ice-free route to the North Atlantic at the time of the YD initiation (Leydet et al., 2018), revising earlier ice sheet reconstructions suggesting that the Laurentide Ice Sheet blocked the most obvious meltwater pathways to the North Atlantic (Lowell et al., 2005). Although direct evidence that a meltwater pulse occurred along this route immediately preceding the YD's onset is limited, some sediment core evidence suggesting that floods through the St Lawrence Valley coincided with the YD initiation does exist (Levac et al., 2015; Rayburn et al., 2011). A northward meltwater flowpath along the MacKenzie Valley to the Arctic Ocean is also possible (Murton et al., 2010; Not and Hillaire-Marcel, 2012; Tarasov and Peltier, 2005). This is supported by both geological and modelling evidence, which suggests that a freshwater pulse into the Arctic Ocean could have weakened AMOC by >30%, considerably more that the <15% resulting from a similar pulse injected into the North Atlantic (Condron and Winsor, 2012). Still other evidence implicates meltwater from the Fennoscandian Ice Sheet (FIS) as the YD trigger (Muschitiello et al., 2015; Muschitiello et al., 2016), highlighting meltwater provenance as a key uncertainty

intrinsic to the meltwater hypothesis. Intriguingly, Muschitiello et al. (2015) find that the FIS-derived meltwater pulse ended at 12.880 ka BP, coinciding with the LSE. It is therefore conceivable that the meltwater pulse resulted from insolation-controlled melting of the FIS, until the cooling associated with the LSE and associated positive feedback reversed this trend, though currently this remains speculative. Finally, evidence from a sediment site SW of Iceland very strongly suggests that

cooling into stadials precedes evidence for iceberg discharges, and that therefore North Atlantic iceberg-related surface water freshening was not the trigger for stadial events (Barker et al., 2015). Despite these issues, a meltwater pulse is still widely considered as the leading hypothesis for the initiation of the YD, although the meltwater source, destination, and duration are still vigorously debated.

Iridium-rich magnetic grains, nanodiamonds, and carbon spherules coinciding with a carbon-rich black layer at approximately 12.9 ka BP were interpreted by Firestone et al. (2007) as evidence that an extraterrestrial impact (or impacts) may have contributed to the YD. The consequences of the impact (possibly a cometary airburst) were hypothesised to have destabilised the Laurentide Ice Sheet, cooled NH climate, and contributed to megafaunal extinction characteristic of the period (Firestone et al., 2007; Kennett et al., 2009). The discovery of significant amounts of impact-derived spherules scattered across North

America, Europe, Africa, and South America at ~12.80 ± 0.15 ka BP further supports the Younger Dryas Impact Hypothesis (YDIH) (Wittke et al., 2013), as does apparently extraterrestrially-derived platinum, found initially in Greenland ice (Petaev et al., 2013) and subsequently globally (Moore et al., 2017), coincident with the YD onset. However, other research questions the evidence for an impact, focussing on perceived errors in dating of the YD boundary layer (Holliday, 2015; Meltzer et al., 2014), misidentification of terrestrially-derived carbon spherules, shocked quartz, and nanodiamonds as extraterrestrial (Pinter

et al., 2011; van Hoesel et al., 2015; Tian et al., 2011), non-uniqueness of YD nanodiamond evidence (Daulton et al., 2017), and inconsistencies regarding the physics of bolide trajectories and impacts (Boslough et al., 2013). Despite these criticisms, independent researchers have linked spherules found within the YD boundary layer to an impact (LeCompte et al., 2012), and identified the source of the spherules as northeastern North America (Wu et al., 2013), not only supporting the hypothesis, but also suggesting an impact site proximal to the southern margin of the Laurentide Ice Sheet (Wu et al., 2013). Most recently,

Wolbach et al. (2018a, 2018b) catalogue extensive evidence for global wildfires they argue were ignited by the impact of fragments of a >100 km diameter comet. According to Wolbach et al. (2018a, 2018b), 9% of global biomass burned, considerably more than that following the end of Cretaceous impact event. Despite this growing catalogue of evidence for an impact event, the YDIH remains extremely controversial. Possibly due to this controversy, the majority of research relating to the YDIH has focussed on verifying the evidence for an impact, and the possible climate repercussions stemming from an

impact remain relatively poorly constrained.

Of these proposed triggers, only the Laacher See eruption is universally accepted as having occurred near the YD/Allerød boundary, and strong evidence now exists dating the eruption precisely at the onset of YD-related North Atlantic cooling (GS-1). However, we emphasise that these hypotheses are not mutually exclusive. For example, meltwater releases into the North

Atlantic and Arctic Ocean undoubtedly did occur during the last deglaciation, and a meltwater pulse and a volcanic eruption occurring within a short time of each other could conceivably result in an amplified climate response. Similarly, available evidence suggests that an impact event may have occurred within a few decades of the LSE, and in fact Wolbach et al. (2018) date the impact to 12.854 ± 0.056 ka BP based on a Bayesian analysis of 157 dated records interpreted as containing evidence for the impact. This date is just 26 years after the 12.880 ka BP LSE (and well within dating uncertainty) and the two events

could therefore have both contributed to environmental change associated with the YD. Conversely, the LSE could also be directly responsible for some of the evidence for an impact, a possibility that has not been investigated thoroughly due to the widespread, but incorrect, concept that the eruption predates the evidence for an impact by ~200 years. An impact event (independently or in conjunction with YD cooling) is suggested to explain North American megafaunal extinctions (Firestone et al., 2007; Wolbach et al., 2018a; Wolbach et al., 2018b), though recent research has built a compelling case that

anthropogenic factors such as overhunting and disease were largely responsible for the demise of many species of large mammalian fauna (Sandom et al., 2014; Bartlett et al., 2016; van der Kaars et al., 2017; Cooper et al., 2015; Metcalf et al., 2016). Notably, megafaunal extinction dates appear correlated with the timing of human colonisation of the Western Hemisphere (Surovell et al., 2016), a result inconsistent with an impact-driven extinction. However, even if future research confirms that megafaunal extinctions were caused by human activity rather than a bolide impact, this does not reduce the

YDIH's explanatory power as a YD trigger. Again, it is conceivable that the LSE, a bolide impact, and/or a meltwater pulse occurred within a short interval, reinforcing each other. A similar eruption-impact mechanism has been proposed for major mass extinctions through geological time (Tobin, 2017; White and Saunders, 2005; Keller, 2005), but these involved long-term flood basalt (e.g., the Deccan Traps) emplacement rather than a single, short-lived explosive eruption. The probability of multiple triggers occurring within a short time of each other is low, but the possibility cannot be excluded outright.


A key issue with the YDIH however is that YD-type events apparently occurred during previous terminations, such as TIII, (Broecker et al., 2010; Cheng et al., 2009; Sima et al., 2004) that are not reasonably attributable to other extraterrestrial impacts. However, TIII would also likely have experienced at least one high latitude Northern Hemisphere M > 6 volcanic eruption. Termination II lacked a YD-type event, possibly because extremely rapid sea level rise (and ice volume decreases) and very

high insolation levels (Carlson, 2008) discouraged a strong post-eruptive positive feedback and/or shortened the 'window' of ideal ice volume conditions. Another issue with the YDIH is that the YD was simply not that anomalous of a cold event, and therefore does not require an unusually powerful trigger. Over the last 120 ka, 26 Greenland Stadial (GS) events occurred (Rasmussen et al., 2014), of which the YD was the most recent. This does not exclude the possibility that the YD was forced by an impact event, but the most parsimonious explanation is that most stadial events had similar origins, implying a much

more commonplace trigger than an impact. Furthermore, nitrogen isotopes suggest that Greenland temperature was 4.5°C colder during the Oldest Dryas (18 to 14.7 ka BP) than the YD (Buizert et al., 2014), again suggesting the transition to the YD does not require an extreme forcing. It is also worth noting that the YD may actually represent a return to the insolation-mediated baseline, and that potentially the Bølling-Allerød (B-A) warm interstadial (i.e., GI-1, from 14.642 to 12.846 ka BP),

immediately preceding the YD, was the anomaly (Thornalley et al., 2011; Sima et al., 2004). Although an in-depth discussion of the B-A is outside the scope of this study, the B-A may represent an interval with a temporarily invigorated AMOC (an 'overshoot') (Barker et al., 2010), which, after reaching peak strength, began to slow down back towards its glacial state because of the lack of a concomitant rise in insolation (Knorr and Lohmann, 2007; Thornalley et al., 2011). The rate of this slowdown was independent of the final trigger of the YD, and indeed much of the cooling back to the glacial baseline was achieved by ~13 ka BP. Consequently, only a small 'nudge' may have been required to expedite the return to the cold baseline state, consistent with a very sulphur-rich volcanic eruption (occurring every few hundred years) but not necessarily with a rare, high-consequence event. In other words, the B-A may represent the transient anomaly, and the conditions within the YD represented typical near-glacial conditions, obviating the need for an extreme YD triggering mechanism. However, the gradual cooling after the B-A differs from the rapid cooling observed at around 12.9 ka BP clearly visible in the NGRIP $\delta^{18}$O and nitrogen isotope data as well as in numerous other North Atlantic records, raising the possibility that the LSE (or other external forcing) expedited the final cooling into the YD. It is also possible however that the increased cooling rate reflects non-linearity inherent in Atlantic oceanic circulation, potentially linked to transitions across key thresholds between different modes of oceanic circulation (e.g., Ganopolski and Rahmstorf, 2001).

A possible scenario is therefore that many glacial terminations were characterised by B-A-type events that were then followed by gradual cooling back to near-glacial conditions. This gradual cooling was sometimes interrupted and accelerated by YD-type events forced by high latitude volcanism, possibly linked to crustal stresses induced by deglaciation (Zielinski et al., 1996). Some deglaciations did not experience a YD-type event, and we speculate that this was perhaps due to short-lived ideal ice volume conditions not coinciding with an eruption. We stress however that deglaciations were also characterised by large meltwater pulses, and consequently the presence of a YD-type event during multiple deglaciations is consisted with either mechanism. Additionally, the absence of a YD-type event during some terminations may simply reflect the absence of preceding B-A-type event and the presence of a very long stadial interval which then transitioned directly into an interglacial (Cheng et al., 2009; Carlson, 2008). Terminations lacking a YD-type event tend to have a strong insolation forcing that may have suppressed AMOC throughout the termination, preventing the development of a B-A-type event and decreasing the possibility of a YD-type event following after peak B-A-type warming (Wolff et al., 2010; Barker et al., 2011; Barker et al., 2010).

## 4 Conclusions

We propose that the unusually sulphur-rich 12.880 ± 0.040 ka BP Laacher See volcanic eruption initiated GS-1 cooling and the atmospheric reorganisation associated with the YD event. Recent revisions to the chronological framework of key European climate archives now strongly suggest that the onset of GS-1 related North Atlantic cooling occurred simultaneously

with the LSE. We have identified a large volcanic sulphur spike within the GISP2 ion data (Zielinski et al., 1997) on the recent GICC05modelext chronology that coincides with both the onset of GS-1 cooling as recorded within the same ice core and the date of the LSE. Lipid biomarker hydrogen isotope ratios from Lake Meerfelder Maar further corroborate that GS-1 atmospheric cooling began at 12.880 ka BP, coincident with the Laacher See Tephra within the same sediment but preceding the larger dynamical atmospheric response associated with the YD in central Europe by ~170 years (Rach et al., 2014). Aerosol-induced cooling immediately following the eruption may have caused a positive feedback involving sea ice expansion and/or AMOC weakening, as previously proposed for other Greenland stadials over the interval 30-80 ka BP (Baldini et al., 2015a), the 6[th] Century AD (Buntgen et al., 2016), the Little Ice Age (Zhong et al., 2011; Miller et al., 2012), and the Holocene (Kobashi et al., 2017). Viewed from this perspective, the YD was simply the latest, and last, manifestation of a Last Glacial stadial.

The strongly asymmetric nature of the sulphate aerosol veil released by the LSE cooled the Northern Hemisphere preferentially, inducing a strong hemispheric temperature asymmetry and potentially triggering a cascade of dynamical climate shifts across both hemispheres, including a southward shifted ITCZ and Hadley Cells. Intermediate ice volume conditions around ~13 ka BP, driven by rising insolation during the Last Glacial termination, may have promoted a positive feedback following the LSE's injection of between 6.76 and 104.8 megatonnes of $SO_2$ into the stratosphere (Textor et al., 2003). This is also consistent with observations that YD-type events were not unique to the last deglaciation, but existed during older deglaciations as well (Broecker et al., 2010). At least one high-latitude M6 eruption likely occurred during most deglacial intervals, and therefore GS-1 and earlier YD-type events may simply reflect the convergence of large, sulphate-rich high latitude NH eruptions with intermediate ice volume conditions. NH continental ice sheet decay induced continental lithospheric unloading, and may have triggered high latitude NH volcanism (Zielinski et al., 1997; Sternai et al., 2016), highlighting the intriguing possibility that eruptions such as the LSE were not randomly distributed geographically and temporally, but instead were intrinsically linked to deglaciation. This perspective is strongly supported by a previous observation that the three largest eruptions (including the LSE) in the East Eifel Volcanic Field (Germany) were all associated with warming during glacial terminations and the reduction in ice mass in northern Europe (Nowell et al., 2006).

The hypothesis that the Laacher See eruption triggered the YD is testable. Detailed tephrochronological studies of the ice containing the sulphate spike identified here could confirm the source of the sulphate, and, if it is confirmed as the LSE rather than a smaller Icelandic eruption, this would provide an important step towards attributing GS-1 and the YD to volcanic forcing (although evidence for GS-1 cooling already seems to occur immediately above the LST in central Europe). Similarly, volcanic sulphate 'triple' isotope ratios of sulphur and oxygen provide information regarding the residence time of volcanic plumes in the stratosphere. The majority of atmospheric processes encourage mass dependent fractionation; however rare mass-independent fractionation processes produce isotope ratios that do not behave according to predictions based on mass dependent processes (Martin et al., 2014). Historical volcanic eruptions where sulphate aerosols reached the stratosphere have been successfully identified in ice cores (Baroni et al., 2008; Savarino et al., 2003), indicating that the technique is effective

at distinguishing large explosive eruptions from smaller local ones. This technique could also determine if the sulphate in the potential LSE sulphate spike reached the stratosphere. Although this would not necessarily confirm the LSE as the source, it would strongly suggest that the sulphate was derived from a climatologically significant eruption rather than a smaller Icelandic one. Most importantly, more climate modelling studies of high latitude, large, and sulphur-rich eruptions under deglacial boundary conditions are needed to constrain the climate effects of volcanism during deglaciation. The role of halogen emissions are particularly understudied, and modelling efforts should also quantify their effects on climate. Finally, accurate dating of other large volcanic eruptions during intermediate ice volume conditions is key to testing the link between volcanism and other Greenland stadials; this information could eventually also support, or refute, the LSE's role in triggering the YD.

More research is clearly necessary to better characterise the sensitivity of Last Glacial climate to volcanic eruption latitude, sulphur content, and magnitude. Due to perceived chronological mismatches, the concept that the YD was triggered by the LSE is vastly understudied compared to both the Younger Dryas Impact Hypothesis and the meltwater forcing hypothesis. However, the concept that the LSE triggered the YD has clear advantages compared to other hypotheses: *i)* there is no disagreement that the eruption occurred, *ii)* available evidence suggests that the eruption occurred synchronously with the initiation of YD cooling, *iii)* a volcanic trigger is consistent with the relatively high frequency of similar, and often more severe, cooling events during intermediate ice volume conditions, *iv)* volcanic aerosol cooling followed by a prolonged positive feedback has been implicated in other cooling events, and *v)* events similar to the YD occurred over other deglaciations, supporting a relatively commonplace trigger such as volcanism. Future research may well demonstrate that the Laacher See eruption did not play any role in catalysing the YD, but the coincidence of a large, high-latitude, and anomalously sulphur-rich eruption with the initiation of YD-related cooling merits serious further consideration.

*Acknowledgments*: We thank four anonymous reviewers and David Pyle, Xu Zhang, and Evžen Stuchlik for constructive comments that greatly improved the manuscript. We also thank David Thornalley for his editorial handling and suggestions which also helped improved the manuscript. Early versions of the manuscript benefitted from critical comments by Bob Hilton, Erin McClymont, and Lisa Baldini. Ed Llewellin is thanked for comments that helped finalise the manuscript. We also thank Dirk Sachse for useful discussions, and Paul Mayewski for providing the GISP2 ion datasets.

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

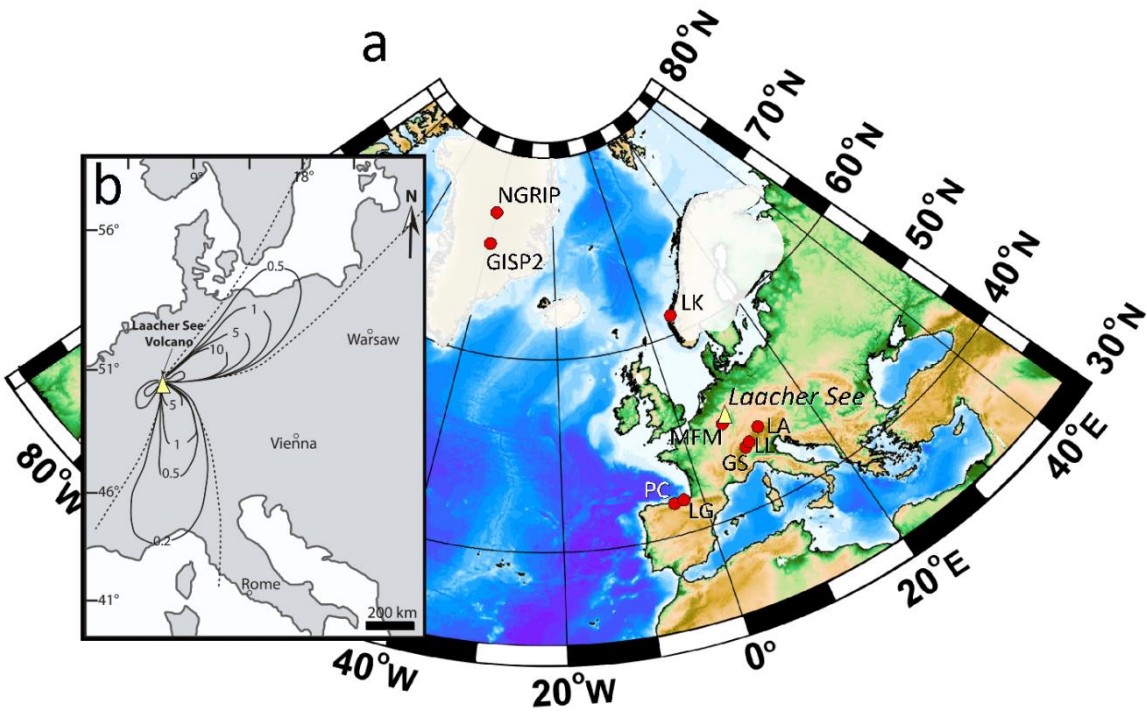

**Figure 1. a) Map with the locations of sites discussed and (b) an isopach map of the Laacher See tephra fall deposits across central Europe. Semi-transparent white areas in (a) demarcate continental glaciers, and the reconstruction and map are adapted from Baldini et al. (2015b). The sites shown are: Laacher See volcano (yellow triangle); LG, La Garma Cave; MFM, Meerfelder Maar; LA, Lake Ammersee; PC, El Pindal Cave; LK, Lake Krakenes; LL, Lake Lucerne; GS, Gerzensee; NGRIP, NGRIP ice core. The dashed line in (b) is the outer detection limits of the distal tephra layers (adapted from Bogaard and Schmincke, 1985). Isopach line labels in (b) are in cm.**

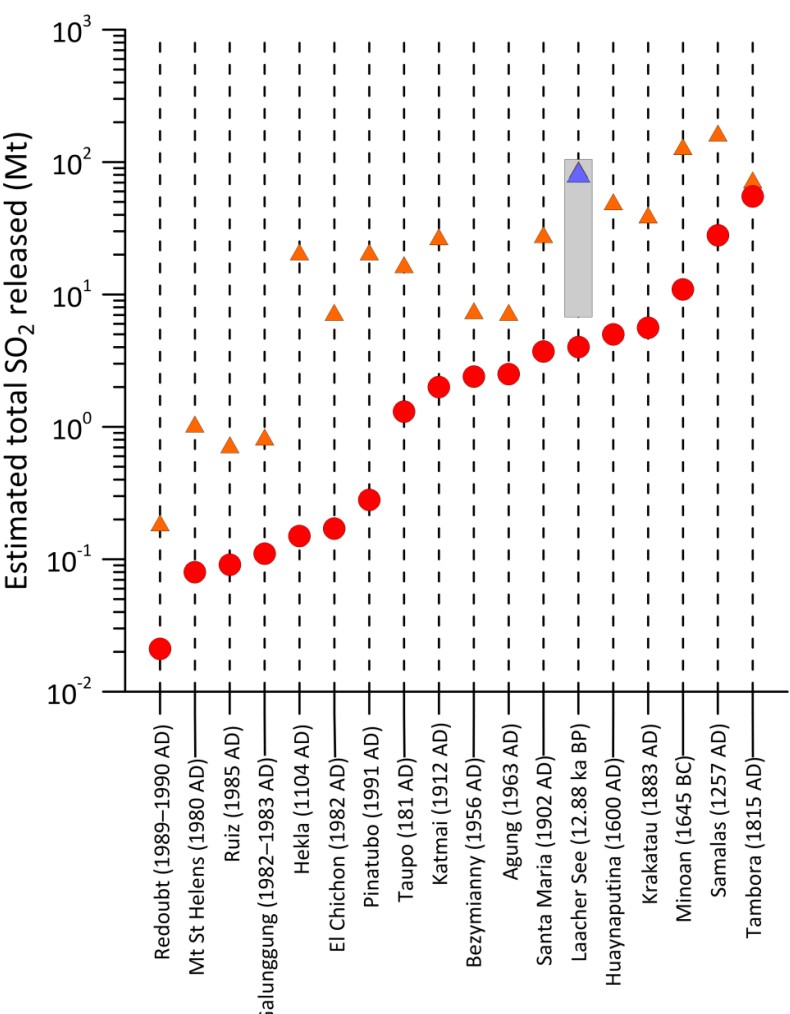

**Figure 2: The SO₂ yield of climatologically significant late Holocene eruptions (Shinohara, 2008). Red circles represent petrologic estimates of SO₂ release, and orange triangles represent estimated actual values, calculated from either satellite or ice core data (Shinohara, 2008). The grey bar represents the range of values for the Laacher See eruption as suggested by Textor et al. (2003; Schenk et al., 2018; Bahr et al., 2018), and the blue triangle in the Laacher See column represents the total estimated SO₂ emitted by the Laacher See eruption (83.6 Mt) assuming that the actual SO₂ emitted is 20.9 times the petrologic estimate, calculated here as the mean value of 17 non-basaltic explosive eruptions (the 16 listed in Shinohara (2008) plus the 1257 AD Samalas eruption (Vidal et al., 2016)).**


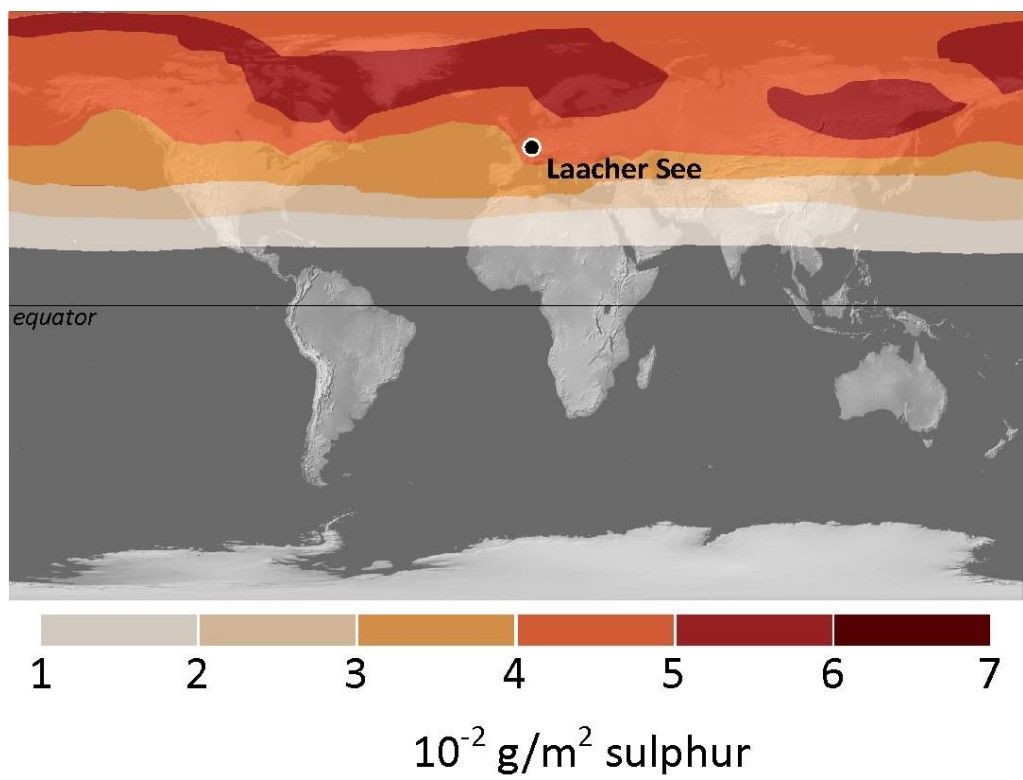

**Figure 3. The distribution of the Laacher See volcanic cloud six months after the ~12.9 ka BP eruption based on existing climate model outputs (MAECHAM4 model) (Graf and Timmreck, 2001). The figure is redrawn based on the original in Graf and Timmreck (2001), which contains details regarding the model and simulation. This simulation assumed that the eruption injected 15 Mt SO$_2$ into the lower stratosphere, ~7x less than the maximum estimate of Textor et al. (2003a).**


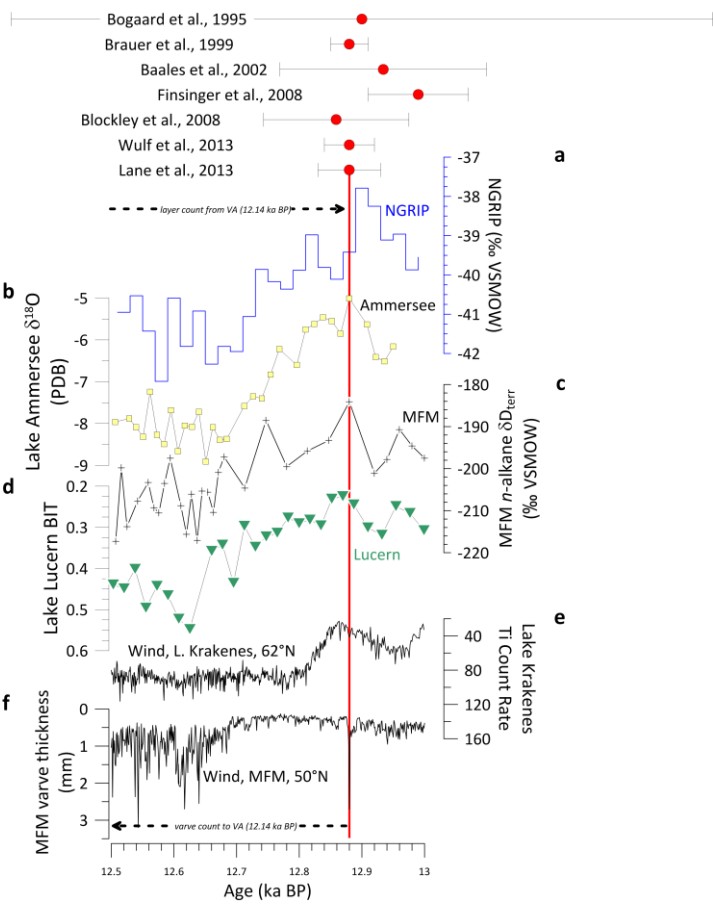

**Figure 4. Temperature (a-d) and wind strength (e, f) proxy records from the North Atlantic and Europe. a) NGRIP ice core δ18O (Greenland, GICC05) (Rasmussen et al., 2006). b) precipitation δ18O reconstructed using deep lake ostracods from Lake Ammersee, southern Germany. c) Meerfelder Maar (MFM) (Eifel Volcanic Field, Germany) n-alkanes δDterr (terrestrial lipid biomarkers) (Rach et al., 2014; Litt et al., 2001), d) Lake Lucern, Switzerland, isoprenoid tetraether (BIT) record (Blaga et al., 2013). e) wind strength as reconstructed using Ti count rate in Lake Kråkenes (Norway); three-point moving average shown. f) Wind strength as reconstructed using MFM varve thickness data (Brauer et al., 2008; Mortensen et al., 2005). The records are arranged so that cooling is down for all the records. The LST (vertical red line) is present or inferred within the MFM, Lake Ammersee, and Lake Lucern cores. The LST is not evident in the NGRIP or Lake Kråkenes cores, and the eruption's timing relative to NGRIP δ18O and Lake Kråkenes Ti is based on layer counting from the Vedde Ash ('VA'), a tephrochronological marker (12.140 ± 0.04 ka BP) also found in MFM (Lane et al., 2013a; Brauer et al., 2008). The published chronologies for Lake Ammersee and Lake Lucern were shifted slightly by a uniform amount (0.115 and -0.093 ka, respectively) to ensure the contemporaneity of the LSE in all the records. This adjustment does not affect the LSE's timing relative to the Lake Ammersee or Lake Lucern climate records. Published radiometric dates for the LSE are shown (red circles) with errors, although the absolute age is not as important as its timing relative to the apparent climate shifts.**

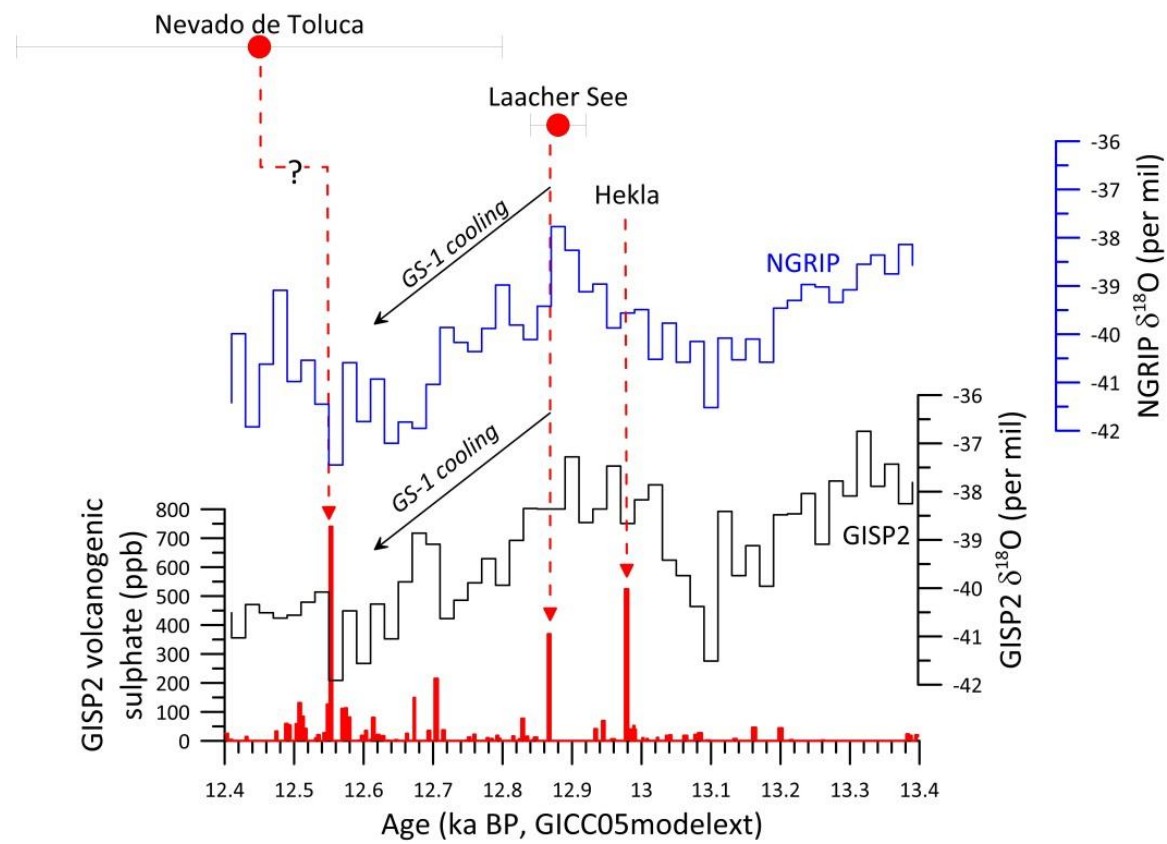

**Figure 5.** The GISP2 (bidecadal (Stuiver et al., 1995)) (black) and NGRIP (blue) ice core $\delta^{18}O$ records synchronised on the GICC05modelext chronology (Seierstad et al., 2014). The red bars indicate GISP2 volcanological sulphate record (also synchronised on the GICC05 timescale). The maximum counting error of the ice core chronology is 0.138 ka at 12.9 ka BP (Seierstad et al., 2014). The best age estimate of the LSE is shown (red circle) (Brauer et al., 2008; Lane et al., 2015) and a possible correlation to a synchronous sulphate spike highlighted by the vertical dashed arrow. The large spike at ~13 ka BP represents a smaller but more proximal Icelandic eruption (Hekla) associated with a volcanic ash layer (Mortensen et al., 2005; Bereiter et al., 2018).


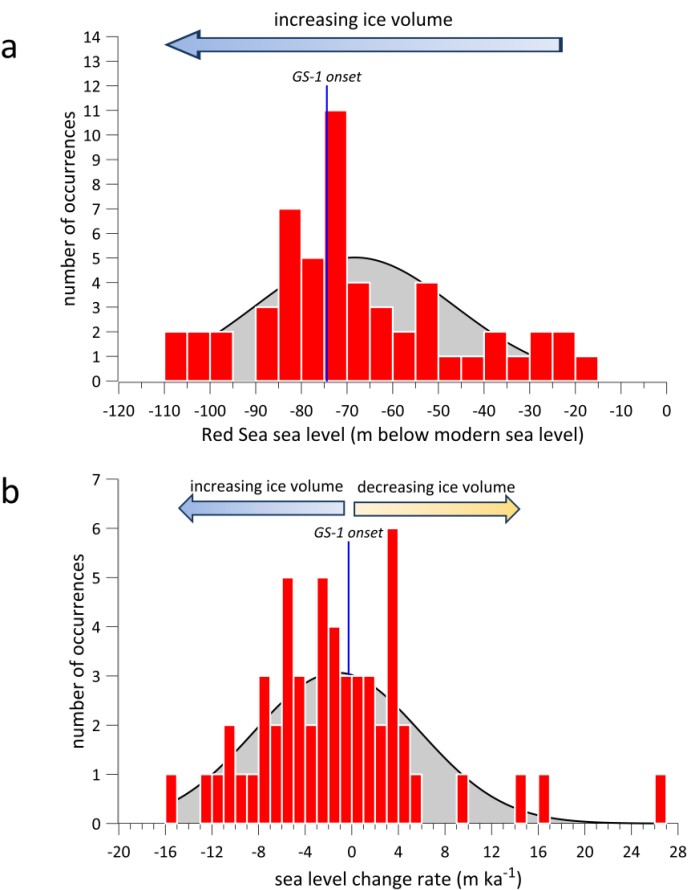

**Figure 6: Histogram of the frequency of abrupt Greenland cooling events relative to ice volume and ice volume change.** Red Sea sea level (Siddall et al., 2003) is used as a proxy for ice volume and to determine (a) the distribution of ice volume conditions associated with all Greenland cooling events identified by Rasmussen et al. (2014) over the last 120 ka (n = 55). A Gaussian best-fit curve describes the distribution (grey-filled curve) (mean = -68.30 m, σ = 21.85 m), and sea level during the GS-1 onset is marked with a blue dashed line. b) The distribution of sea level change rates (reflecting global ice volume shifts) associated with 55 Greenland cooling events over the last 120 ka. This distribution is also described by a Gaussian best-fit curve (grey-filled curve) (mean = -1.35493 m ka$^{-1}$, σ = 7.19 m ka$^{-1}$); 21 events are associated with decreasing ice volume, and 34 are associated with increasing ice volume.


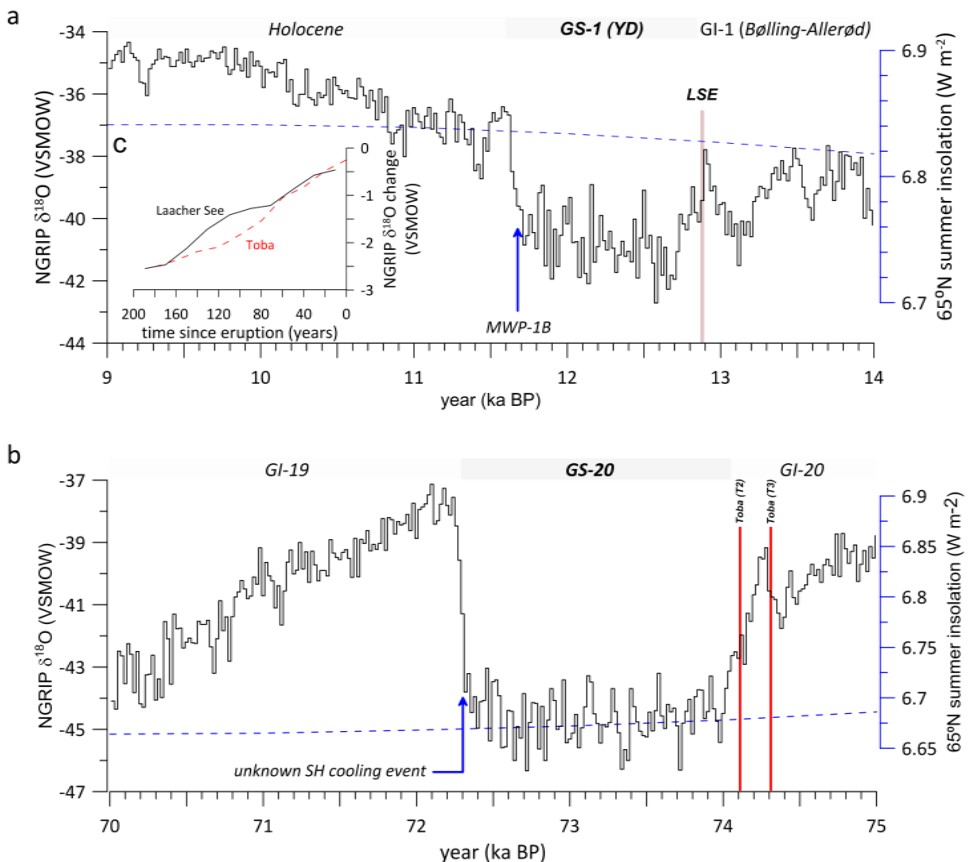

**Figure 7: Two five-ka intervals of the NGRIP δ¹⁸O record bracketing (a) the ~12.9 ka BP Laacher See Eruption and (b) the ~74 ka BP Toba supereruption (Rasmussen et al., 2006). The two eruptions are among the most well-dated eruptions of the Quaternary. The LST is not apparent in the NGRIP core (although we identify a candidate sulphate spike in the GISP2 record; see Figure 5), and the timing of the LST relative to the NGRIP δ¹⁸O record is based on the difference in ages between the LST and the Vedde Ash found in European lake sediments (Lane et al., 2013a). The timing of the Toba eruption relative to NGRIP is based on the position of the most likely sulphate spike identified by Svensson et al. (2013). Blue arrows indicate the timing of possible SH cooling events. The timing of Meltwater Pulse-1B is based on Ridgwell et al, 2012, but the source, timing, and even occurrence of the meltwater pulse are still debated. The inset panel (c) shows NGRIP δ¹⁸O during the 200 years immediately following the two eruptions (the sulphate spike T2 is assumed to represent Toba, as suggested by Svensson et al. (2013)).**