# Peer review of "Evaluating the link between the sulphur-rich Laacher See volcanic eruption and the Younger Dryas climate anomaly"

_Climate of the Past, 2017_

## Short Comment (SC1) · 28 Nov 2017

Dear authors, This is manuscript of the high overall interest but a statement indicating that Laacher see eruption (LSE) effect could last for some 5 years is in the contrast to surprisingly main conclusion not completely supported by own data and highly speculative that this event could trigger YD cooling. Resulting the title of msc starting with "Reevaluation" is inappropriate to the msc content. I have following comments and suggestions: • You ignore close linkage in referred paper of Fireston et al., 2007 between known (or unknown) events and extinction of great mammals (megafauna) at YD onset. There is an evident fact that LSE could not be a reason. • You do not refer to the paper of Petaev et al., 2013 on the discovery of a platinum layer at YDB in the Greenland ice core. Petaev et al. knew about the Sulphur anomaly but they

did not consider that LSE could even leave any signs in the Greenland ice. I strongly suggest to pay an attention to Petaev et al. work. • Recently published paper of Moore et al., 2017 following Petaev et al. study with crucial findings of widespread platinum anomaly at YD onset is missed as well. • As follows from missed Morore et al and referred Fireston et al., Wittke et al and Kennett et al papers ecological disaster of process(es) was more drastic in American continent than in Europe at YD onset (disappearance of Clovis paleoindian culture, magafaunal extinction, black mat on a huge area). There is nothing about it in the msc. • Westerly come from west. What was on the west from Europe actually? • There are archeological studies on the west coast of Norway that people had inhabited this region in time of extension of continental glacier during YD and they fished for migrating Atlantic salmon. So, the Golf stream had to work at that time and Central-north Atlantic cold hardly by covered by ice. I visited such place deep in Hardanger fjord providing this information but I have not found a paper. You can see from following information that even this fact my by controversial as everything based on radio-carbon dating in the period of YD onset (http://sp.lyellcollection.org/content/411/1/9, http://onlinelibrary.wiley.com/doi/10.1002/jqs.2781/pdf). • The msc uses European lake archives to support authors' conclusion but the most of these archiver remained undiscovered by the authors. The most of them indicate another event to LSE: Lotter et al., 1993 should by replaced by Lotter et al., 1995 showing the evidence of two perturbations et YD onset, Karponai et al, 2011 showing the first evidence of acidification at YD onset, papers of Andronikov et al., 2013, 2015 on lakes et Baltic region indicating both extraterrestrial impact and volcanic activity in Europe, the most recent study on Merfelder maar = iconic site of tephrologist (Jones et al., 2017) show that LSE was not the only volcanic event at YD onset. (And many other studies) • Attached please find cited papers including 3 abstracts produced by our team. They were published in Meteortitics and Planetary Science as predecessors of papers (submitted and in reviewing process). Our abstracts demonstrate complexity of processes at YD onset when natural weathering in lake catchment was blended

with influence of volcanic activity of LS and supposed extraterrestrial impact event in the form of airburst with the center above East Canada. Please pay attention to the fact that there is synchronicity in geochemical evolution of lakes in American continent and in the central Europe (Norton et al., 2016 reinterpreted by Stuchlik et al., 2017). • Firestone et al., 2007 supposed that impact event could trigger volcanic activity in the whole earth. There is lack of such information regarding both Americas in your msc. Yellowstone super volcano could be one candidate despite I have not found a scientific evidence of it. I hope you find my comments and suggestions helpful. Good luck. Evzen Stuchlik

Please also note the supplement to this comment:
https://www.clim-past-discuss.net/cp-2017-147/cp-2017-147-SC1-supplement.zip

---

## Referee Comment (RC1) · Anonymous Referee #1 · 1 Dec 2017

Baldini and colleagues present a previously published volcanogenic sulphate record from GISP2 ice cores and compile age estimates for the Laacher See tephra to argue that the LS eruption is recorded in Greenland ice cores as a large sulphate spike positioned approximately at the onset of Greenland Stadial 1 (i.e. Younger Dryas). The authors finally suggest that the LS eruption triggered the YD through a chain of feedbacks resulting from the initial volcanic-induced cooling in the Northern Hemisphere.

Even though their hypothesis is tantalizing I find the manuscript excessively speculative and the conclusions very much stretch what can be observed in the reconstructions. I should also point out that the putative link between the LST and a volcanic sulphate spike in GISP2 records was originally suggested by Brauer et al. (1999). In my opinion, the present manuscript doesn't offer anything new but I brief review of the climatic

implications of volcanic cooling and far-reaching speculations on the triggers for the YD. Besides the speculative aspects of the paper, I have three major concerns that I think should be taken into account.

Main comments:

1. Assigning the LS eruption to a nearly synchronous large sulphate peak in GISP2 is a flawed assumption until proven otherwise by tephochronological analyses. The authors should consider alternative hypotheses. For instance, volcanic records from Greenland are particularly sensitive to Icelandic eruptions due to the eruptive frequency and proximity (e.g. Abbott and Davies, 2012). Hence, even small-size Icelandic eruptions characterised by moderate sulphate emissions can result in disproportionally large sulphate anomalies in ice core stratigraphies.

2. I think the paper would greatly benefit from exploring some of the mechanisms the authors discuss (e.g. southward wind shifts, AMOC decline, sea ice expansion, ect.) by looking into climate model output and/or historical reanalyses data sets. The authors present output data from MAECHAM4 simulations but they don't provide any additional analysis of the atmospheric parameters in the model. CMIP historical climate simulations and PMIP last-millennium simulations offer valuable model data to examine the role of volcanic eruptions on the coupled atmosphere-ocean system. Model output based on volcanically-forced transient simulations with earth System Model (MPI-ESM) (Jungclaus et al., 2010) (available online) may also be useful.

3. I think some of the arguments proposed here (as well as the records presented in Figure 3) have conveniently been picked to craft a story where the LS eruption stands out as the INITIAL cooling that triggered the YD. This does not faithfully reflect the state of the knowledge around the dynamics that took place prior to and at the onset of the YD. Several reconstructions show that a gradual but substantial cooling across Northern Europe and in the Nordic Seas preceded the start of GS-1. Pollen records from the British Isles indicate cooling as early as 13,200 years BP (Walker et al., 2012). A drop

in air temperatures a few centuries before GS-1 has also been recorded in chironomid-based temperature records from Norway (Bakke et al., 2009; see supplementary information), Sweden (Muschitiello et al., 2015), and the Netherlands (Heiri et al., 2007). Similarly, sea surface temperature records from the Norwegian Sea indicate a rapid cooling approaching YD values as early as 13,500 years BP (Bakke et al., 2009) with temperature values dropping by ca. 2 °C. Well-dated paleoceanographic records from the coast of Norway also show a progressive aging of surface waters starting at ca. 13,200 years BP (Bondevik et al., 2006), which is suggestive of a slowdown of surface-water circulation and reduced advection of warm subtropical waters prior to the start of GS-1. I therefore believe that the authors should do a better job in contextualizing the LS eruption in the regional climate picture since proxy reconstructions from Central Europe (e.g. Grafenstein, 1999; Rach et al., 2014) are evidently not fully representative of large-scale North Atlantic climate. In particular, the records mentioned above clearly suggest that cooling and climate deterioration was long underway before the start of the YD, which implies that the cooling associated with the LS eruption cannot be the trigger for the YD.

Specific comments:

L30: An alternative route has been proposed involving the drainage of the Baltic Ice Lake, the timing of which precisely coincide with the start of GS-1 (Muschitiello et al., 2016).

L42-47: Without entering into the discussion on the credibility of the "impact" theory, I think that the evidence is still undermined by the poor chronological accuracy of the platinum/iridium anomaly.

L54: What stratigraphic frameworks? Please specify and provide reference.

L57-58: "...consistent with the LS eruption..." Please add here "as recorded in varved and 14C dated lake records".

L148: Please provide reference here on the impact of volcanic forcing on AMOC (e.g. Otterå et al., 2010).

L162: Please specify what model. This is not clear.

L176-192: This section is all very speculative. Please see my comment on the possibility of examining climate model output to support these claims.

L181: None of the studies cited here present direct evidence of sea ice changes (only indirect and mainly based on terrestrial reconstructions). Please consider referring to marine reconstructions (e.g. Cabedo-Sanz et al., 2013).

L223: I'm not convinced that the data in Figure 5a and b follow a Gaussian distribution. Rather, the frequency distributions seem skewed. Also the authors cannot claim that "a Gaussian distribution exist" when they fit a Gaussian-best-fit model to their data. They should use a resampling/bootstrap method to draw from their empirical distribution and only then establish its shape.

L219-230: I suggest estimating the frequency of Greenland cooling relative to the number of volcanic events occurring at the end of the preceding interstadial. This should somewhat inform on the potential link between volcanisms and the onset of stadial cooling.

L246: The rate of cooling (as seen in the d18O ice-core stratigraphies) is similar among most of the stadials, not exclusively between GS-1 and GS-20.

L247-249: The initial cooling can be tested using ice-core (e.g. d15N) and other proxy-based temperature reconstructions and compare the magnitude of the temperature change with that associated with large historical eruptions.

L253: Is this claim based on ice-core d18O profiles? I don't see any substantial warming in Greenland during the second half of GS-1. d18O of ice can be misleading due to changes in moisture source of precipitation. Before making this claim I would check the ice-core temperature records (d18O diffusion or d15N). As far as I can tell from the

d18O data both GS-1 and GS-20 in Greenland were characterised by cold conditions throughout the stadial.

L258-260: Again, this is extremely speculative and I don't think there is any evidence supporting this claim.

L275-284: The magnitude of the spike does not necessarily scale linearly with the magnitude of the eruption and could solely depend on the proximity of the volcanic source or the atmospheric circulation pattern.

L290: Age uncertainties for the LST and the sulphate spike should be reported on their respective time scales (i.e. GICC05, IntCal13, MFM varve time scale, etc.). In addition, time scale offsets between the 14C time scale and GICC05 should be accounted when comparing radiocarbon-based and ice-core ages (e.g. Muscheler et al., 2014).

L291: Please see my previous comment. Cooling in the North Atlantic started a few centuries before the onset of GS-1.

L305-306: As I mentioned above, it would be helpful to look into the frequency of stadials relative to the number of volcanic eruptions during the preceding interstadials.

References

Abbott, P.M., Davies, S.M., 2012. Volcanism and the Greenland ice-cores: The tephra record. Earth-Science Reviews. doi:10.1016/j.earscirev.2012.09.001 Bakke, J., Lie, Ø., Heegaard, E., Dokken, T., Haug, G.H., Birks, H.H., Dulski, P., Nilsen, T., 2009. Rapid oceanic and atmospheric changes during the Younger Dryas cold period. Nature Geoscience 2, 202–205. doi:10.1038/ngeo439 Bondevik, S., Mangerud, J., Birks, H.H., Gulliksen, S., Reimer, P., 2006. Changes in North Atlantic Radiocarbon Reservoir Ages During the Allerød and Younger Dryas. Science 312, 1514–1517. doi:10.1126/science.1123300 Brauer, A., Endres, C., Negendank, J.F.W., 1999. Lateglacial calendar year chronology based on annually laminated sediments from Lake Meerfelder Maar, Germany. Quaternary International 61, 17–25. doi:10.1016/S1040-6182(99)00014-2 Cabedo-Sanz, P., Belt, S.T., Knies, J., Husum, K., 2013. Identification of contrasting seasonal sea ice conditions during the Younger Dryas. Quaternary Science Reviews 79, 74–86. doi:10.1016/j.quascirev.2012.10.028 Grafenstein, U. v., 1999. A Mid-European Decadal Isotope-Climate Record from 15,500 to 5000 Years B.P. Science 284, 1654–1657. doi:10.1126/science.284.5420.1654 Heiri, O., Cremer, H., Engels, S., Hoek, W.Z., Peeters, W., Lotter, A.F., 2007. Lateglacial summer temperatures in the Northwest European lowlands: a chironomid record from Hijkermeer, the Netherlands. Quaternary Science Reviews 26, 2420–2437. doi:10.1016/j.quascirev.2007.06.017 Jungclaus, J.H., Lorenz, S.J., Timmreck, C., Reick, C.H., Brovkin, V., Six, K., Segschneider, J., Giorgetta, M.A., Crowley, T.J., Pongratz, J., Krivova, N.A., Vieira, L.E., Solanki, S.K., Klocke, D., Botzet, M., Esch, M., Gayler, V., Haak, H., Raddatz, T.J., Roeckner, E., Schnur, R., Widmann, H., Claussen, M., Stevens, B., Marotzke, J., 2010. Climate and carbon-cycle variability over the last millennium. Climate of the Past 6, 723–737. doi:10.5194/cp-6-723-2010 Muscheler, R., Adolphi, F., Knudsen, M.F., 2014. Assessing the differences between the IntCal and Greenland ice-core time scales for the last 14,000 years via the common cosmogenic radionuclide variations. Quaternary Science Reviews 106, 81–87. doi:10.1016/j.quascirev.2014.08.017 Muschitiello, F., Lea, J.M., Greenwood, S.L., Nick, F.M., Brunnberg, L., Macleod, A., Wohlfarth, B., 2016. Timing of the first drainage of the Baltic Ice Lake synchronous with the onset of Greenland Stadial 1. Boreas 45, 322–334. doi:10.1111/bor.12155 Muschitiello, F., Pausata, F.S.R., Watson, J.E., Smittenberg, R.H., Salih, A.A.M., Brooks, S.J., Whitehouse, N.J., Karlatou-Charalampopoulou, A., Wohlfarth, B., 2015. Fennoscandian freshwater control on Greenland hydroclimate shifts at the onset of the Younger Dryas. Nature Communications 6, 8939. doi:10.1038/ncomms9939 Otterå, O.H., Bentsen, M., Drange, H., Suo, L., 2010. External forcing as a metronome for Atlantic multidecadal variability. Nature Geoscience 3, 688–694. doi:10.1038/ngeo955 Rach, O., Brauer, A., Wilkes, H., Sachse, D., 2014. Delayed hydrological response to Greenland cooling at the

onset of the Younger Dryas in western Europe. Nature Geoscience 7, 109–112. doi:10.1038/ngeo2053 Walker, M., Lowe, J., Blockley, S.P.E., Bryant, C., Coombes, P., Davies, S., Hardiman, M., Turney, C.S.M., Watson, J., 2012. Lateglacial and early Holocene palaeoenvironmental "events" in Sluggan Bog, Northern Ireland: Comparisons with the Greenland NGRIP GICC05 event stratigraphy. Quaternary Science Reviews 36, 124–138. doi:10.1016/j.quascirev.2011.09.008

---

## Referee Comment (RC2) · Anonymous Referee #2 · 2 Dec 2017

I am not convinced by this paper, and so recommend rejection or major revisions. Perhaps the timing of the eruption is close to indicators of cooling (not my specialty), but the hand-waving arguments about why this eruption caused cooling and larger ones did not are not convincing. Indeed more work is required, including climate model simulations that include all the relevant processes. Even if the timing was close, there is no proof that this was not just a coincidence.

The authors claim that the climate system was particularly sensitive to volcanic forcing at the time, but this is just speculation. Where are the model results to show this? In fact, in Fig. 4 there are two larger eruptions during the same period. Why did only Laacher See produce cooling? They claim in the Fig. 4 caption that the Hekla eruption was more proximal, and therefore should be discounted, but the way it works is that

[Figure]

Icelandic eruptions into the westerlies have to go around the world before the acid snow deposits on Greenland, and so there is no reason to think that it would have a smaller climate impact than Laacher See.

Even the size of the eruption is speculation, and the authors mix mass of SO2 with that of elemental Sulphur with that of stratospheric aerosol. What do they claim actually was the stratospheric loading for this eruption? And each time you talk about mass, please convert it to the same chemical so it can be compared.

The title is confusing. Why is it "re-evaluating?" There is no initial evaluation that is addressed in the abstract or in the paper.

Since the Laacher See eruption was high latitude, we would expect that for the same stratospheric loading, it would have much less of a climate impact than an equivalent tropical eruption, since the atmospheric lifetime would be much shorter, and there is less insolation at high latitudes. When you compare to Toba, this must be addressed. And if the eruption was in the fall or winter, most of the aerosol would have fallen out of the stratosphere before the Sun comes up the next summer and there would be minimal impact on climate.

The paper is replete with undefined acronyms, making it very confusing. All acronyms have to be defined the first time they are used. For example, what is LST? It is never defined. Is it LSE and a typo? What are TOMS, NGRIP, GISP2, GICC05modelext, ITCZ, GS-20, GI-19 (in Fig. 6), ...?

Please keep in mind that there will be readers not from your specific discipline, and so jargon needs to be defined. GS-1 is finally defined long after it is used, but the authors still never say what Greenland Stadial 1 is. What is a stadial? Why does Greenland have one? How many does it have?

The paper talks about magnitudes for volcanic eruptions, but never says what the scale is. Magnitude of what? If not of sulphur injection, then what is the point? And where do the data come from? There are no references to that.

As for the Toba eruption, the paper is missing key references on the climate impact. Robock et al. (2009) found a larger short-term impact, but no long-term effect. Timmreck et al. (2010) claim that it would have had a small impact, as the particles would have grown and had a smaller impact per unit mass.

Robock, A., C. M. Ammann, L. Oman, D. Shindell, S. Levis, and G. Stenchikov, 2009: Did the Toba volcanic eruption of ∼74 ka B.P. produce widespread glaciation? J. Geophys. Res., 114, D10107, doi:10.1029/2008JD011652.

Timmreck, C., et al., 2010: Aerosol size confines climate response to volcanic super-eruptions. Geophys. Res. Lett., 37, L24705, doi:10.1029/2010GL045464.

In any case, I find the Haslam and Petraglia (2010) Figure 1 very convincing that it got cold before the eruption. By the way, that reference is missing from the reference list. Why does the timing of the Toba eruption in Fig. 6 here differ from that in Fig. 1 of Haslam and Petraglia (2010)? Which is correct, and why?

The paper ignores all the work that has shown that the 1257 Samalas eruption caused the Little Ice Age (Zhong et al., 2011; Miller et al., 2012; Slawinska and Robock, 2017). What does this tell us about the claim that a much smaller eruption of Laacher See caused a much larger climate response?

Miller, G. H., Á. Geirsdóttir, Y. Zhong, D. J. Larsen, B. L. Otto-Bliesner, M. M. Holland, D. A. Bailey, K. A. Refsnider, S. J. Lehman, J. R. Southon, Ch. Anderson, H. Björnsson, and T. Thordarson, 2012: Abrupt onset of the Little Ice Age triggered by volcanism and sustained by sea-ice/ocean feedbacks. Geophys. Res. Lett., 39, L02708, doi:10.1029/2011GL050168.

Slawinska, J., and A. Robock, 2017: Impact of volcanic eruptions on decadal to centennial fluctuations of Arctic sea ice extent during the last millennium and on initiation of the Little Ice Age. J. Climate, doi:10.1175/JCLI-D-16-0498.1,

http://journals.ametsoc.org/doi/abs/10.1175/JCLI-D-16-0498.1

Zhong, Y., G. H. Miller, B. L. Otto-Bliesner, M. M. Holland, D. A. Bailey, D. P. Schnei-der, and A. Geirsdottir, 2011: Centennial-scale climate change from decadally-paced explosive volcanism: a coupled sea ice-ocean mechanism. Clim. Dyn., 37, 2373-2387.

It would have been nice to have used hanging indents or additional spacing for the reference list to make it easier for the reader to find each paper in the list.

In addition, there are another 35 comments in the attached annotated manuscript that need to be addressed.

Please also note the supplement to this comment:
https://www.clim-past-discuss.net/cp-2017-147/cp-2017-147-RC2-supplement.pdf

**Supplement:**

[revised manuscript text omitted]

---

## Author Comment (AC1) · 6 Dec 2017

We thank you very much for your in-depth, helpful, and interesting comments. We value you taking the time to comment on our manuscript. First, we would like to simply note that the goals of our submitted manuscript are straightforward: i) to highlight that the timing of the Laacher See eruption seems to be indistinguishable from the initiation of cooling associated with the Younger Dryas, ii) to highlight the possibility that the effects of volcanic eruptions can persist longer than just 1-3 years, and finally 3) that consequently the Laacher See eruption should be viewed as a viable trigger for the Younger Dryas Event. In other words, if the LSE occurred at the correct time (and it appears that it did), and if an eruption of this scale and sulphur content could catalyse extended cooling (and it appears that it could), then logically the LSE should be

considered a viable trigger for the Younger Dryas. Clearly more research needs to be conducted on this topic, but getting the idea out there is the key first step. We note that this is the only hypothesis where it is universally agreed that the proposed trigger actually occurred, so is not unacceptably speculative in our opinions.

We feel that an extended discussion of other proposed triggers is outside of the scope of the current submission, but we will include slightly more discussion with the revisions. There are reams of papers discussing the pros and cons of the Younger Dryas Impact Hypothesis specifically, and providing a thorough review of all the evidence for or against this hypothesis is not possible or necessary. For example, you are correct that the Laacher See Eruption would not account for the observed megafaunal extinctions across North America. However, recent papers [Cooper et al., 2015; Metcalf et al., 2016; Rule et al., 2012; van der Kaars et al., 2017] make an extremely strong case that this was caused by human migration, and that therefore the LSE (or an impact, or a meltwater pulse) would not have needed to cause any extinction. This perspective is also supported by the presence of other Younger Dryas-type events that apparently occurred during other Glacial terminations, e.g. TIII [Broecker et al., 2010] but that were not associated with megafaunal extinctions, implying that neither YD-type climate change nor a bolide impact were the cause of the megafaunal extinctions. The point though is that this has all been discussed before, and we do not feel that defending/rebuking other hypotheses in depth is the purpose of this manuscript, although we do mention the advantages and disadvantages of these competing hypotheses in order to put our hypothesis into context. Furthermore, we do not argue that a bolide impact did not happen near the YD boundary (it may have), so defending the presence or absence of a Pt spike, shocked quartz, black carbon, nanodiamonds, etc. is well beyond the scope of the manuscript. That being said, we will add some more detail in the revised manuscript.

In response to your comments that are specific to our hypothesis: 1) Comment: Our 'statement indicating that Laacher see eruption (LSE) effect could last for some 5 years

is in the contrast to surprisingly main conclusion not completely supported by own data and highly speculative that this event could trigger YD cooling.' 1) Response: We discuss a proposed ice/ocean feedback in detail in Sections 3.2, 3.3., and 3.4, and the concept of a positive feedback amplifying the original volcanic forcing is increasingly commonplace (see recent paper by Kobashi et al., Scientific Reports 2017 for example). There are now several papers that suggest the presence of a sea ice/oceanic circulation feedback that amplifies the initial short-lived aerosol cooling, and we will discuss these further in the revised manuscript as suggested by another reviewer. We therefore feel that the concept of a longer-term volcanic forcing is well-defended already by several pages of text as well as previously published papers (these will be included and discussed in the revisions); we do not feel that it is highly speculative if you are familiar with this most recent literature. Upon any revision, we will revise this text to ensure that this message is clear, and describe the positive feedback mechanism in more detail.

2) Comment: 'Resulting the title of msc starting with "Reevaluation" is inappropriate to the msc content.' 2) Response: The Laacher See eruption was one of the very earliest proposed triggers for the YDE, before it was discarded. However, the most recent lake core and ice core data suggest that the YD cooling occurred synchronously with the LSE, so we feel that 'Re-evaluating' is the correct word to use here. We were not the first to suggest the eruption as a trigger, although we are 're-evaluating' the eruption's climatological consequences in a modern context. Still, another reviewer raises this same issue, so although we feel that this is in fact the correct term, we will either change the title or better clarify why we chose to use this term in the title in the revisions.

We thank you for all the papers that you have provided. We will include these in any revisions, where relevant.

References: Broecker, W. S., G. H. Denton, R. L. Edwards, H. Cheng, R. B. Alley, and A. E. Putnam (2010), Putting the Younger Dryas cold event into context, Quaternary

Sci. Rev., 29(9–10), 1078-1081. Cooper, A., C. Turney, K. A. Hughen, B. W. Brook, H. G. McDonald, and C. J. A. Bradshaw (2015), Abrupt warming events drove Late Pleistocene Holarctic megafaunal turnover, Science, 349(6248), 602-606. Metcalf, J. L., et al. (2016), Synergistic roles of climate warming and human occupation in Patagonian megafaunal extinctions during the Last Deglaciation, Sci. Adv., 2(6), 8. Rule, S., B. W. Brook, S. G. Haberle, C. S. M. Turney, A. P. Kershaw, and C. N. Johnson (2012), The Aftermath of Megafaunal Extinction: Ecosystem Transformation in Pleistocene Australia, Science, 335(6075), 1483-1486. van der Kaars, S., G. H. Miller, C. S. M. Turney, E. J. Cook, D. Nurnberg, J. Schonfeld, A. P. Kershaw, and S. J. Lehman (2017), Humans rather than climate the primary cause of Pleistocene megafaunal extinction in Australia, Nat Commun, 8, 7.

---

## Author Comment (AC2) · 13 Dec 2017

Comment #1: Baldini and colleagues present a previously published volcanogenic sulphate record from GISP2 ice cores and compile age estimates for the Laacher See tephra to argue that the LS eruption is recorded in Greenland ice cores as a large sulphate spike positioned approximately at the onset of Greenland Stadial 1 (i.e. Younger Dryas). The authors finally suggest that the LS eruption triggered the YD through a chain of feedbacks resulting from the initial volcanic-induced cooling in the Northern Hemisphere. Even though their hypothesis is tantalizing I find the manuscript excessively speculative and the conclusions very much stretch what can be observed in the reconstructions.

[Figure]

Preliminary Response #1: We would like to thank Reviewer #1 for taking the time for providing such a comprehensive and fair review. We appreciate that this topic is controversial, and the worst-case scenario would be if the manuscript were published without rigorous review. We also thank the reviewer for suggesting that the hypothesis is 'tantalising'; we agree and additionally we hope to convince Reviewer #1 that the conclusions are not overly speculative.

Reviewer #1 raises some interesting challenges to the hypothesis, but ultimately our responses to these will not affect our model. The comments will however take some time to respond to thoroughly, and we would therefore like to provide an initial response to a few of Reviewer #1's comments in an effort to stimulate discussion before our more in-depth response. In particular, we appreciate the very thorough suggestions for other records to consider, and here we provide reasons why the records are or are not relevant to the Laacher See hypothesis. We assure the reviewer that we have not cherry-picked the records that were used in the manuscript. Rather these were chosen as the records with the most robust chronologies both in absolute terms and with respect to the timing of GS-1 versus the LSE. All of the records used contain both a high resolution regional temperature proxy record and either the Laacher See Tephra directly, or an excellent layer-counted chronology. In our revisions, we will make our selection criteria clearer.

Muschietello et al., 2015 Nature Geoscience: We do not doubt that meltwater pulses did occur and undoubtedly affected climate during the last deglaciation. In fact, we suggest that the YD was terminated by a Southern Hemisphere meltwater pulse that triggered long-term warming. Muschietello et al. argue that meltwater forcing affected climate from 13.1 to 12.880 ka BP. We do not argue against this. However, nothing in this paper contradicts our manuscript. In fact we note that, once again, a pronounced climate shift occurs coincident with the Laacher See eruption at 12.880 ka BP. The large inflection point in Figure 4 (Panel d and elsewhere) is indistinguishable from the date of the eruption. At this point it is therefore difficult to disentangle whether the

forcing was a meltwater pulse from the Fennoscandian Ice Sheet (or elsewhere), a bolide impact, or the Laacher See eruption, but without publication of our manuscript, future research cannot assess the pros and cons of each.

Heiri et al. 2007: The paper presents a very interesting chironomid-based temperature record of the Younger Dryas from the Netherlands. Unfortunately, it is too low resolution to be particularly useful. The dating is also not as high precision as the records that we have chosen, though we note that the decrease in July temperature starts just after the eruption based on their chronology. Still, we chose not to include this record and others like it because of the low resolution and more uncertain chronology.

Walker et al 2012: Unfortunately the low resolution and ambiguous results make this paper a low priority for inclusion. There are hundreds of Younger Dryas climate reconstructions globally, and including all of them is simply not possible. Pollen reconstructions in particular are problematic due to the generally lower resolution of the datasets, the often less well-constrained chronologies, and the local nature of the proxy. We do not see how this paper provides a significant challenge to either a LSE, a bolide impact, or a meltwater trigger for GS-1 and the YD.

Bondevik et al., 2006: This is quite an interesting paper that we somehow had missed previously, and we thank the reviewer for bringing it to our attention. Their Figure 3 is particularly striking, and shows a pronounced inflection point in the radiocarbon concentrations precisely at the Laacher See eruption (Panels B, C, and D). They state that 'A high reservoir age during the YD could be explained by a combination of increased sea-ice cover and reduced advection of surface water to the North Atlantic', both of which are entirely consistent with our proposed positive feedback mechanism. This paper appears to strongly support our conclusions, but we would appreciate Reviewer #1 highlighting any issues we may have missed. We will probably include some of the radiocarbon data discussed here within the context of a more substantial discussion of the positive feedback.

Bakke et al., 2009: The correlations between the data presented in this paper and the Laacher See eruption are remarkably strong, and we will include this as a new dataset in our stacked diagram or in another figure. In particular, Supplemental Figures S5 (noting of course that their data is in years b2k) and S7 show that the largest inflection point is coincident with the eruption. We will include these data in our revisions, and we thank the reviewer for bringing this supplemental material to our attention.

Comment #2: In my opinion, the present manuscript doesn't offer anything new but I brief review of the climatic implications of volcanic cooling and far-reaching speculations on the triggers for the YD.

Preliminary Response #2: Should this manuscript be published, it would open the door to the consideration of a new trigger for the Younger Dryas Event. Although we do present a review of volcanic cooling, to our knowledge the positive feedback discussed is novel within the context of the YDE. We also identify the sulphate spike associated with the Laacher See eruption, using the most recent chronology for the GISP2 ice core. Reviewer #1 is correct that Brauer et al 1999 did consider this same spike as being potentially linked to the LSE, but ultimately they decided that it occurred too close to the YD boundary and concluded that an earlier spike represented the LSE (we will include this in the discussion); so I think it is correct to say that we are the first to attribute the LSE to this particular spike, though we will discuss the fact that Brauer et al (1999) considered the spike earlier. This manuscript uses previously published data to reach new conclusions, and we therefore feel that this goes above and beyond a 'brief review'.

Comment #3: In particular, the records mentioned above clearly suggest that cooling and climate deterioration was long underway before the start of the YD, which implies that the cooling associated with the LS eruption cannot be the trigger for the YD.

Preliminary Response #3: In this response we assume the reviewer meant 'GS-1' instead of 'YD', because there is general agreement that cooling did start well before

the start of the YD in Central Europe, and this in fact forms a key part of our hypothesis. We disagree that there is clear evidence for GS-1 related cooling well before the start of the GS-1. There may be some cooling in some records that may or may not be linked to the YD, but there is generally also a clear inflection point at the start of the GS-1. We also note that because the two highest profile hypotheses (a meltwater pulse and a bolide impact) for the YD trigger are also proposed to have coincided with the start of GS-1, any issues with 'cooling and climate deterioration' being long underway before the Laacher See eruption are also issues with the other hypotheses. Again we note that neither the meltwater pulse or the bolide impact are universally acknowledged as even having occurred (close to the start of GS-1), whereas no debate exists that the Laacher See eruption happened, has real potential to have cooled climate, and appears to be coincident with the start of GS-1 in the best records of regional climate. It is remarkable that even after decades of intense research the Younger Dryas event trigger is still unclear. Possibly this is because the trigger is still unknown; we wish to propose the Laacher See hypothesis so that it can be thoroughly tested in future research just as the other hypotheses have been.

---

## Referee Comment (RC3) · X. Zhang (Referee) · 9 Jan 2018

The manuscript "Re-evaluating the link between the Laacher See volcanic eruption and the Younger Dryas" by Baldini et al proposed Laacher See Eruption (LSE) as a potential trigger of the YD cold interval during the last deglaciation. The manuscript is good written and easy to follow. The authors argued that radiative effect of the LSE could lead to a cooling over the Northern Hemisphere, which eventually triggered the YD due to the existence of "sweet spot" of millennial-scale variability during glacial periods and positive feedbacks. I do see the potential of this mechanism, which enables an improvement of our understanding of YD dynamics. However, I'm a bit suspicious of its reliability. Some points are summarized in the following: 1) Responses of ocean circulation (AMOC) to a Northern Hemisphere volcano eruption is not that supportive

of authors' argument. According to Pausata et al 2015 (PNAS), eruption's effect on AMOC is positive (strengthening) rather negative (weakening) at the first 20 years after the eruption, contrast to the weakening AMOC during YD. 2) Effect of southward ITCZ shift will lead to an increase of salinity in the North Atlantic subtropics, which will also act as a negative feedback to a potential weakening AMOC (Schmidt et al 2006 Nature). 3) Although the ice volume during YD is beneficial to the occurrence of millennial-scale variability (Zhang et al 2014 Nature), the high CO2 level (~250 ppm) will shift the "sweet spot" to a lower level of global ice volume (Zhang et al 2017 Nature Geo). This will weaken the arguments proposed by the authors. Nevertheless, I do see a potential of LSE (or northern hemisphere volcanic eruption) as a trigger to YD —Muschitiello et al (2017 Nature Comm, also cited by the authors) recently proposed that the volcanic eruptions can effectively influence the mass balance of ice sheet via altering its surface albedo. This will promote the ice-sheet melting, leading to freshwater input to the North Atlantic and weakening the AMOC. I'm not an expert on data and climate response to the volcanic eruptions. But I think if the author can well improve the robustness of their arguments (probably by rephrasing the mechanisms), this will be a nice manuscript for Clim Past.

Line 145: Citation "Pasauta et al 2015 Tellus B" is not proper here. It should be Pausata et al 2015 PNAS.

---

## Short Comment (SC2) · 11 Jan 2018

This is an interesting paper which revisits the hypothesis that the eruption of the Laacher See Tuff may have provided a trigger for the onset of the Younger Dryas (YD). The weakness of the paper as it stands at the moment is that there is little new evidence to present, so the arguments haven't really changed since the hypothesis was first introduced. In some ways the paper might be better framed as a critical review of competing hypotheses for the 'cause' of the onset of the YD, in that this would help to bring some clarity to the current discussion, and to identify the key gaps in knowledge for the commuinity to tackle. As it stands, the paper is perhaps too much like a pitch in favour of a particular hypothesis.

[Figure]

For me the leading questions (which come out of this paper) are:

- was the LST eruption the source of a sulphur 'spike' in Greenland ice cores; and how could we test this assertion?

- what was the total volatile yield from the LST eruption; and what new measurements are needed to improve on this assessment. (And did the halogen release have any impact?)

- what cascade of physical processes could lead to the observed pattern of response seen for the onset of the YD; and is a volcanic eruption a sufficient driver, on its own.

Detailed points

Line 44 – there is also a documented enrichment in noble metals (e.g. platinum), at this stratigraphic level both in North America and Europe (Moore et al., Scientific Reports 7 Article Number: 44031, 2017; and papers by A Andronikov). Line 50 – 'new support' yes, but not much new evidence. Line 50 – ages: some explanation is needed about the framing of time in the paper, as the model ages are derived from multiple approaches.

Lines 53- 56: the emphasis here isn't quite right. The evidence for a 200-year time break between LST and the onset of 'YD' conditions in continental Europe remains firm. What has changed – since Lane et al., 2015, is the recognition that the onset of YD is time-transgressive. So – by inference – LST overlaps with GS-1 and the onset of YD as recorded in ice chemistry in Greenland.

Line 57 – does the 'GICC05modelext chronology' need a word or two of explanation and a citation?

LST Impacts

Line 74: see also the extensive work by Felix Riede on the impacts of the LST:

Book, 2017 - Splendid Isolation : The eruption of the Laacher See volcano and southern Scandinavian Late Glacial hunter-gatherers. / Riede, Felix. Aarhus : Aarhus Universitetsforlag, 2017. 214 p.

2016 - Changes in mid- and far-field human landscape use following the Laacher See eruption (c. 13,000 years BP). / Riede, Felix. In: Quaternary International, Vol. 394, 02.2016, p. 37-50.

2012 - Bayesian radiocarbon models for the cultural transition during the Allerod in southern Scandinavia, Riede, Felix; Edinborough, Kevan, JOURNAL OF ARCHAEO-LOGICAL SCIENCE 39, 744-756

2008 - The Laacher See-eruption (12,920 BP) and material culture change at the end of the Allerod in northern Europe, Riede, Felix, JOURNAL OF ARCHAEOLOGICAL SCIENCE 35, 591-599

Volcanic Emissions

Lines 78 - 85 – this section needs some critical revision and updating.

Harms and Schmincke (2000) estimated, using mass balance, an SO2 yield of 20 Tg. Harms et al. (2004) did some experiments on LST magmas and determined the P, T , H2O conditions under which the magma was stored; you could revisit the calculations of Harms & Schmincke to re-estimate the S and water budgets of the system – taking account of the work that Bruno Scaillet and colleagues have done on other systems. The '150 Mt' value should be cited as Schmincke et al (1999, Quaternary International, 61, 61-72) – it is, as the authors say 'highly speculative' and based on using the 'Pinatubo multiplier'; this can certainly be improved upon, rather than being taken as a starting point for the argument.

Similar calculations have been attempted by Textor et al., 2003, (Geol Soc London Spec Pub, 'Volcanic Degassing', 213, 307-328); who also estimated the total halogen yield.

Discussion on volcanic emissions

Recent papers may also add a little to the discussion here: for example - - Colose, C.M., A.N. LeGrande, and M. Vuille, 2016: Hemispherically asymmetric volcanic forcing of tropical hydroclimate during the last millennium. Earth Syst. Dyn., 7, 681-696, doi:10.5194/esd-7-681-2016. - LeGrande, A.N., K. Tsigaridis, and S.E. Bauer, 2016: Role of atmospheric chemistry in the climate impacts of stratospheric volcanic injections. Nature Geosci., 9, no. 9, 652-655, doi:10.1038/ngeo2771.

Line 178 – 'five years' may be an overestimate: in Graf and Timmreck's model, sulphate aerosol had an e-folding time of 11 months; and the detectable signal of volcanic stratospheric sulphate aerosol is usually considered to be less than three years. Lines 180 – 195 – there's not really any new evidence here? Line 208 – the magnitude of the eruption is not relevant, it's the magnitude of the gas release that is the key point. The LST magma is an unusual composition, so surely this is the starting point for why it may have had an exceptional impact? Lines 274 – 284: there still is no way of linking a sulphate peak in an ice core to a particular eruption, in the absence of any tephra so this remains speculative. It remains possible that sulphur mass-independent isotopic fractionation signals may help to identify plumes that entered the stratosphere (e.g. Martin et al., 2014, Volcanic sulfate aerosol formation in the troposphere, JOURNAL OF GEOPHYSICAL RESEARCH-ATMOSPHERES Volume: 119 Issue: 22 Pages: 12660-12673), but this still won't help with source identification.

---

## Author Comment (AC3) · 15 Jan 2018

Thank you for your detailed and very useful comments, which will undoubtedly help improve the manuscript. We will address these in depth in our revised submission, but here are some initial thoughts.

Here we are essentially trying to highlight that given the Laacher See eruption's timing, its sulphur content, its latitude, and the accumulating evidence that eruptions can affect climate for far longer than the residence time of aerosols in the stratosphere, the eruption should be considered (once again) as a viable candidate for the YD trigger. No consensus exists regarding what triggered the YD, and it is conceivable that this is due to recent research focussing on the wrong triggering mechanisms. However, we stress

that at the moment the two other hypothesised triggers are certainly also plausible; we will therefore provide an extended review of the pros and cons of each hypothesis. We also completely agree with your point regarding magnitude versus sulphur content, and we will revise the manuscript accordingly. We will include an enhanced discussion regarding how the hypothesis can be tested, including the requirement for detailed tephrochronological work on the ice containing the candidate sulphate spike to positively ascribe this to the LSE. Thank you also for the references, and we will include these in our revised submission.

---

## Author Comment (AC4) · 15 Jan 2018

We thank the reviewer for their supportive and helpful comments. The reviewer has summarised our message very well in their comments.

We will include more discussion regarding the individual points raised in the revised submission. Briefly here, the reviewer is correct that Pausata et al (2015) found that an eruption would strengthen AMOC. Other modelling studies based on historical data also suggest that eruptions may strengthen AMOC (Ottera et al., 2010; Swingedouw et al., 2014; Ding et al., 2014). Other models suggest that AMOC may intensify initially, but then weaken after about a decade (Mignot et al., 2011). A modelling study by Schleussner and Feulner (2013) suggested that volcanic eruptions occurring during

the last millennium triggered increased Nordic Sea sea ice extent which weakened AMOC and eventually cooled the entire North Atlantic Basin. Other research finds that North Atlantic sea ice growth following a negative forcing weakened oceanic convection and northward heat export during the Little Ice Age (Lehner et al., 2013). These are all studies focussing on eruptions that occurred over the last 1000 years, and they are still yielding contradictory results. Therefore, we feel that how an eruption might affect AMOC at ∼12.9 ka BP is still essentially unknown.

We thank the reviewer for raising this point, as well as for the other references and points raised, which will strengthen our discussion and improve the manuscript overall.

---

## Referee Comment (RC4) · Anonymous Referee #4 · 16 Jan 2018

The subject of the Younger Dryas cooling is one of considerable interest and fascination in the scientific community. Here, most research has been dominated by one theme - that the cooling was triggered by a freshwater flood, or rerouting of meltwater, to the North Atlantic ocean. The idea that the YD cooling might have been triggered by a volcanic eruption has received much less attention and is very interesting.

Overall, I really enjoyed the paper. It's very well written, easy to follow, and provides a nice break from the more typical meltwater-trigger hypothesis. Indeed, I found the discussion about the sensitive of climate to intermediate ice volume conditions, and the alignment of this 'ideal' configuration, to the timing of the YD very enlightening. But whether a volcano actually triggered the YD is hard to tell from this paper. Yes, there was an eruption around the time of the YD cooling, but did it really produce a 1000-yr

cooling? As such, the manuscript would have been vastly improved if the authors had done their own climate modeling. I think it would have been fantastic to try and see whether a volcano could have triggered a YD-like cooling. Indeed, the authors note that previous studies (fig 2) released 10-time LESS SO2 to the atmosphere than what is estimated here. Whether these experiments should be undertaken, I will leave that up to the authors, but I'm not going to rejecting this paper simply because they were not carried out.

Finally, I wasn't sure if the MWP-1b discussion was really needed. The existence of this period of rapid sea level rise is still very much debated, as is its source, with various camps arguing back-and-forth over an Antarctic or Laurentide contribution.

Anyway, my overall opinion is that this is a very interesting paper and it should be published with minor corrections/edits.

---

## Author Comment (AC6) · 1 Feb 2018

**Reviewer #2 Comments and Responses:**

**Comment #1:** I am not convinced by this paper, and so recommend rejection or major revisions. Perhaps the timing of the eruption is close to indicators of cooling (not my specialty), but the hand-waving arguments about why this eruption caused cooling and larger ones did not are not convincing.

**Response #1:** We are sorry to read that Reviewer #2 feels that our arguments are overly speculative. We feel that perhaps the reviewer has missed some key points, which probably reflects that these were poorly communicated on our part. We hope that the revised submission as well as our responses will help better communicate our hypothesis, and we welcome the opportunity to improve the manuscript.

Regarding the last part of Reviewer #2's comment, we are unclear as to which eruptions the reviewer is referring to when Reviewer #2 mentions '…larger ones did not…'? We state:

*'…it is in fact the only known sulphur-rich high latitude eruption coinciding with the most sensitive ice volume conditions during the last deglaciation. '*

and devote an entire section to other eruptions (Section 3.6), so the other large eruptions that the reviewer is referring to is not clear. We have published previous research indicating that there is a strong statistical significance between large Northern Hemisphere eruptions and long-lasting climate change (Baldini et al., 2015, Scientific Reports) and there is now abundant evidence that Toba also triggered long-lasting climate change (discussed below, in our responses to Comment #7). So there is strong evidence that other larger eruptions did cause cooling, and we argue that the Laacher See eruption occurred during a particularly sensitive climatological transition, so we wonder if the reviewer could clarify which eruptions they are referring to. We now also include a lengthy section discussing volcanic impacts on climate over the last millennium, and an enhanced discussion regarding the amount of sulphur in the eruption, which was substantial. Part of that enhanced discussion is pasted below:

*'Both Pinatubo and the LSE were Magnitude 6 (M6) eruptions, where 'magnitude' is a measure of eruption size referring to the amount of material erupted (Deligne et al., 2010 JGR) on a logarithmic scale. However, the cooling effect of a volcanic eruption is controlled by the amount of sulphur released, and not necessarily the eruption size (Rampino and Self, 1982). In general, magnitude and*

*erupted sulphur amounts are well correlated (Oppenheimer, 2003), and therefore magnitude is often used as a surrogate for sulphur yield. All the available evidence suggests that, if the LSE deviates from this trend, it was anomalously enriched in sulphur relative to its magnitude (Baales et al., 2002; Scaillet et al., 2003), and almost certainly released considerably more $SO_2$ into the stratosphere than the climatologically significant 1991 AD Pinatubo eruption.'*

**Comment #2:** Indeed more work is required, including climate model simulations that include all the relevant processes.

**Response #2:** We agree that climate model simulations would benefit the research, but we have decided not to include them for three reasons. i) Neither the meltwater forcing nor the bolide impact hypotheses used climate model simulations to support the initial hypotheses initially, yet both hypotheses led to fruitful discussions and future elaborations, including climate model research. We therefore feel that climate model simulations would make for interesting future work, and indeed we are looking into this ourselves. But we do not feel that they are required for this current submission. ii) There is a good chance that different climate models would return different answers, in which case modelling is better left for future researchers who can devote considerable attention towards developing robust and replicable model outputs. iii) We may not know the details of all the relevant feedbacks, and therefore any model might be incomplete. This issue is actually brought up by the reviewer in their own phraseology: "…climate model simulations that include *all the relevant processes.*" The issue is that we may not know all the relevant processes. For example, if we knew that there was in fact a pronounced positive feedback following Northern Hemisphere eruptions during intermediate ice volume conditions, we would know that this feedback was responsible for the Younger Dryas Event as well as other Greenland stadial events. Possibly the reason that we do not know what forced these events is that we do not know about the relevant process, and if that is the case, modelling would not help. This is point is further illustrated by a recent paper by Diallo et al., 2017 GRL, that states *"Thus, climate model simulations need to realistically take into account the effect of volcanic eruptions, including the minor eruptions after 2008, for a reliable reproduction of observed stratospheric circulation changes."* If ambiguous modelling results, and the incorporation of unrealistic eruptions effects into models, are demonstrably an issue in 2008, they will be even more of an issue during the less-well understood YD interval where a key feedback may remain unquantified. For these reasons, we feel strongly that

modelling is best left for future work where intercomparison between different models is possible. We hope that this paper, if published, would encourage modellers to consider the mechanisms and feedbacks proposed, and include these in future models.

**Comment #3:** Even if the timing was close, there is no proof that this was not just a coincidence. The authors claim that the climate system was particularly sensitive to volcanic forcing at the time, but this is just speculation. Where are the model results to show this?

**Response #3:** Both of the other leading hypotheses for a Younger Dryas trigger also rely on the coincidence of a trigger with the advent of cooling. For example, the meltwater pulse hypothesis relies on the timing of a hypothetical meltwater pulse with the start of the Younger Dryas. However, unlike the Laacher See hypothesis, there is little agreement within the community regarding the source of that meltwater or in fact whether it even occurred simultaneously as the start of GS-1. The bolide impact hypothesis also relies on the coincidence of evidence for a meteor airburst or impact with the Younger Dryas initiation, but whether or not this even occurred is extremely controversial. The Laacher See hypothesis does rely on coincidence, but the event itself is much clearly expressed than the other leading hypotheses. In fact, it is the only hypothesis that features a trigger that is universally accepted as actually having happened. We have now included a lengthy (almost 1,000 words) new section ('3.7 Compatibility with other hypotheses') to better discuss how the Laacher See hypothesis is competitive relative to the other leading proposed triggers.

Please see our response to why we choose not to include models above (Response #2).

Finally, we are not the first to suggest that the climate system is particularly sensitive to volcanic forcing during climatological transitions, and this is not 'speculation'. It is well established that millennial-scale climate change was most sensitive to a forcing during intermediate ice volume conditions, and we simply propose that that forcing was volcanism. Zielinski et al. (1996) noted that when the climate system is in a state of flux it is more sensitive to external forcing, and that any post-volcanic cooling would be longer lived. Importantly, Rampino and Self 1992 (Nature) stated that 'Volcanic aerosols may also contribute a negative feedback during glacial terminations, contributing to brief episodes of cooling and glacial readvance such as the Younger Dryas Interval.' Our research confirms and builds on this earlier work, and identifies a volcanic eruption potentially responsible for the YD. We have included the following text:

*"This perspective is consistent with previous observations, including those of Zielinski et al. (1996) who noted that when the climate system is in a state of flux it is more sensitive to external forcing, and that any post-volcanic cooling would be longer lived. Importantly, Rampino and Self (1992) stated 'Volcanic aerosols may also contribute a negative feedback during glacial terminations, contributing to brief episodes of cooling and glacial readvance such as the Younger Dryas Interval.'. Our results are entirely consistent with this perspective, and here we identify the volcanic eruption responsible for the YD."*

**Comment #4:** In fact, in Fig. 4 there are two larger eruptions during the same period. Why did only Laacher See produce cooling? They claim in the Fig. 4 caption that the Hekla eruption was more proximal, and therefore should be discounted, but the way it works is that Icelandic eruptions into the westerlies have to go around the world before the acid snow deposits on Greenland, and so there is no reason to think that it would have a smaller climate impact than Laacher See.

**Response #4:** The volcanological information regarding the size of the Hekla eruption is from published sources (Muschiatello et al., 2017, Nature Communications; Mortensen et al 2005, Journal of Quaternary Science), so we are simply referring to previously published research when we refer to the fact that Hekla was substantially smaller than the Laacher See eruption, and that it appears in the GISP2 ice core. Additionally, the reviewer is incorrect about Icelandic eruptions, and it is well-established that sulphate from even small Icelandic eruptions appears in Greenland ice cores (Muschiatello et al., 2017, Nature Communications; Abbott and Davies, 2012, Earth Science Reviews; Abbott et al., 2012, QSR). Muschiatello et al. (2017) state: "Icelandic volcanoes remain the dominant source of volcanogenic aerosols in Greenland ice cores due to their relative proximity and high eruptive frequency". The other eruption we mention in the Figure 4 caption is the Nevado de Toluca eruption; we refer the reviewer to the text already in our manuscript that explains why its climate expression may not be as clear:

*'The eruption was approximately the same size as the LSE, so the lack of climate cooling may reflect a different climate response due to the eruption's latitude, which caused a more even distribution of aerosols across both hemispheres, or a lower sulphur load. The 12.6 ka BP sulphate spike is associated with a short but dramatic cooling; therefore the lack of long-term cooling may simply*

*reflect the fact that temperatures had already reached the lowest values possible under the insolation and carbon dioxide baseline conditions characteristic of that time.'*

**Comment #5:** Even the size of the eruption is speculation, and the authors mix mass of SO2 with that of elemental Sulphur with that of stratospheric aerosol. What do they claim actually was the stratospheric loading for this eruption? And each time you talk about mass, please convert it to the same chemical so it can be compared.

**Response #5:** We thank the reviewer for picking up on this. This partially stems from an ambiguity in a previously published paper that we used as a reference. We have now converted everything to the same chemical, except where the context requires us to discuss a different one.

**Comment #6:** The title is confusing. Why is it "re-evaluating?" There is no initial evaluation that is addressed in the abstract or in the paper.

**Response #6:** This issue was also brought up by another comment by another reader of the manuscript. The Laacher See eruption was proposed very briefly in the late 1980s as a trigger for the YDE, before it was discarded due to evidence (incorrectly) showing that it occurred too early. However, the most recent lake core and ice core data indicate that the beginning of YD cooling (the start of GS-1) occurred synchronously with the LSE, so we feel that 'Re-evaluating' is the correct word to use here. We stated in the introduction of our originally submitted manuscript:

*"Instead, we re-introduce and provide new support for the hypothesis that the YD was triggered by the ~12.9 ka BP eruption of the Laacher See volcano, located in the East Eifel Volcanic Field (Germany). Early research considered the eruption as a possible causative mechanism for the YD (Berger, 1990). However, the concept was dismissed because lacustrine evidence across central Europe appeared to indicate that the YD's clearest expression appeared ~200 years after the Laacher See Tephra within the same sediments (e.g., Brauer et al., 2008; Brauer et al., 1999; Hajdas et al., 1995)."*

We were not the first to suggest the eruption as a trigger, although we are 're-evaluating' the eruption's climatological consequences in a modern context. Still, because two reviewers raise this same issue, we have now included the following text to better explain why we chose to use this term in the title:

*"Therefore, we are 're-evaluating' the Laacher See eruption's role in triggering the YD, building on early research that first suggested the eruption as a causative mechanism for the YD, and later research that dismissed the original version of the Laacher See hypothesis"*

**Comment #7:** Since the Laacher See eruption was high latitude, we would expect that for the same stratospheric loading, it would have much less of a climate impact than an equivalent tropical eruption, since the atmospheric lifetime would be much shorter, and there is less insolation at high latitudes. When you compare to Toba, this must be addressed.

**Response #7:** We have added the following text to discuss this:

*'The residence time of aerosols within the atmosphere is not critical within the context of this model provided the positive feedback is activated. A sufficiently high aerosol-related cooling even over only one summer and one hemisphere could suffice. The long-term climate response will depend on the climate background conditions; if a NH eruption occurs during an orbitally-induced cooling trend (as was apparently the case for Toba), the eruption will catalyse cooling towards the insolation-mediated baseline. If the eruption occurs during rising insolation and intermediate ice volume (e.g., Laacher See), the eruption will trigger the feedback which will continue until it is overcome by a sufficiently high positive insolation forcing. A useful way of visualising this is to consider two extreme scenarios: i) very high $CO_2$ concentrations and no ice (e.g., the Cretaceous) and ii) very low $CO_2$, low insolation, and very high ice volume (e.g., the Last Glacial Maximum). An eruption the size of the LSE would probably not significantly affect climate beyond the atmospheric residence time of the sulphate aerosols during either scenario. Under the background conditions characteristic of the first scenario, insufficient aerosol forcing would occur to trigger ice growth, and consequently no positive feedback would result. In the case of the second scenario, the eruption would cause ice growth and cooling for the lifetime of aerosols in atmosphere before conditions returned to the insolation-mediated baseline. In contrast to these extreme scenarios, an injection of volcanogenic sulphate aerosols into the NH atmosphere would most effectively trigger a feedback during intermediate $CO_2$*

*(and consequently ice volume) conditions; in other words, during a transition from one ice volume state to another. Under these conditions, we suggest that activation of the feedback would occur even if the sulphate aerosols settled out of the atmosphere after just one year. Therefore, although the Toba and Laacher See eruptions were of very different magnitudes, the nature of the positive feedback would depend largely on the background conditions present during the individual eruptions. We argue that both eruptions were large enough, and contained enough sulphur, to activate the positive feedback under their respective background conditions. The strength of the positive feedback was then controlled by background conditions rather than the size of the eruptions. Furthermore, we predict that the more asymmetric the hemispheric distribution of the aerosols, the stronger the feedback. For these reasons, the long-term climate response following the extremely large but low latitude Toba eruption would approximate those of the far smaller but high latitude Laacher See eruption. It is also worth noting that the long-term climate repercussions to a very large volcanic eruption would probably not consist of long term radiative cooling (i.e., 'volcanic winter') but rather of geographically disparate dynamical shifts. '*

**Comment #8:** And if the eruption was in the fall or winter, most of the aerosol would have fallen out of the stratosphere before the Sun comes up the next summer and there would be minimal impact on climate.

**Response #8:** For historical eruptions similar in size to the Laacher See eruption, aerosols have remained in the atmosphere for much longer than one year. For example, aerosols remained in the atmosphere for ~3 years after the Pinatubo eruption (15°N, 120°E) (Diallo et al., 2017), which is estimated to have released approximately eight times less sulphur than the LSE (20 Mt versus 150 Mt) (Sheng et al., 2015; Baales et al., 2002). Aerosols remained in the atmosphere for more than one year even after the 1980 Mount St. Helen's eruption (46°N, 122°W) (Pitari et al., 2016), which injected up to almost 100x less sulphur into the atmosphere than the LSE (2.1 Mt versus 150 Mt) (Pitari et al., 2016; Baales et al., 2002) and erupted laterally (Eychenne et al., 2015, JGR-Solid Earth). Furthermore the eruption almost certainly occurred in the late spring. We have added the following text to discuss this in the manuscript:

*'Initiation of the positive feedback requires volcanic aerosols to remain in the atmosphere for at least one summer season. Evidence based on the seasonal development of vegetation covered by the LST suggests that the LSE occurred during late spring or early summer (Schmincke et al., 1999), and varve*

*studies similarly suggest a late spring/early summer eruption (Merkt and Muller, 1999). Available evidence therefore suggests that the eruption occurred just prior to maximum summer insolation values, maximising the potential scattering effects of the volcanogenic sulphate aerosols. Even if it were a winter eruption, for historical eruptions similar in size to the Laacher See eruption, aerosols have remained in the atmosphere far longer than one year, regardless of the eruption's latitude. For example, aerosols remained in the atmosphere for ~three years after the Pinatubo eruption (15°N, 120°E) (Diallo et al., 2017), which is estimated to have released approximately nine times less SO2 than the LSE (~15-20 Mt versus 150 Mt) (Sheng et al., 2015; Baales et al., 2002). Aerosols remained in the atmosphere for approximately three years even after the 1980 Mount St. Helen's eruption (46°N, 122°W) (Pitari et al., 2016), which injected almost 100x less SO2 into the atmosphere than the LSE (2.1 Mt versus 150 Mt) (Baales et al., 2002; Pitari et al., 2016) and erupted laterally (Eychenne et al., 2015). In short, the LSE eruption probably occurred during the late spring or early summer, but even if the eruption were a winter eruption, the LSE's aerosols would have certainly persisted over at least the following summer, with the potential to catalyse the positive feedback we invoke.'*

**Comment #9:** The paper is replete with undefined acronyms, making it very confusing. All acronyms have to be defined the first time they are used. For example, what is LST? It is never defined. Is it LSE and a typo? What are TOMS, NGRIP, GISP2, GICC05modelext, ITCZ, GS-20, GI-19 (in Fig. 6), ...? Please keep in mind that there will be readers not from your specific discipline, and so jargon needs to be defined. GS-1 is finally defined long after it is used, but the authors still never say what Greenland Stadial 1 is. What is a stadial? Why does Greenland have one? How many does it have?

**Response #9:** The reviewer is quite correct here, and we apologise for not defining these terms. We have gone through the manuscript and defined all acronyms and terms which might be confusing for a non-specialist. We have not mentioned the total number of stadials because it is not relevant to the manuscript and would add confusion. The cause of other stadials is not known for sure, though we do highlight that our previous research implicates volcanism.

**Comment #10:** The paper talks about magnitudes for volcanic eruptions, but never says what the scale is. Magnitude of what? If not of sulphur injection, then what is the point? And where do the data come from? There are no references to that.

**Response #10:** 'Magnitude' is a common term in volcanology that refers to the amount of tephra and lava erupted. It is very difficult to know for sure how much sulphur was in eruptions recorded in the geological past due to the fact that much of the sulphur existed in a volatile phase and is not preserved in the rock record. In general magnitude and sulphur concentrations are well correlated (Oppenheimer et al., 2003), and therefore magnitude is therefore often used as a surrogate for sulphur concentrations. All the available evidence suggests that if the LSE deviates from this trend, it is anomalously enriched in sulphur than expected. We have added the following text to the manuscript:

*'Both Pinatubo and the LSE were Magnitude 6 (M6) eruptions, where 'magnitude' is a measure of eruption size referring to the amount of material erupted (Deligne et al., 2010 JGR) on a logarithmic scale. However, the cooling effect of a volcanic eruption is controlled by the amount of sulphur released, and not necessarily the eruption size (Rampino and Self, 1982). In general, magnitude and erupted sulphur amounts are well correlated (Oppenheimer, 2003), and therefore magnitude is often used as a surrogate for sulphur yield. All the available evidence suggests that, if the LSE deviates from this trend, it was anomalously enriched in sulphur relative to its magnitude (Baales et al., 2002; Scaillet et al., 2003), and almost certainly released considerably more SO2 into the stratosphere than the climatologically significant 1991 AD Pinatubo eruption.'*

**Comment #11:** As for the Toba eruption, the paper is missing key references on the climate impact. Robock et al. (2009) found a larger short-term impact, but no long-term effect. Timmreck et al. (2010) claim that it would have had a small impact, as the particles would have grown and had a smaller impact per unit mass.

Robock, A., C. M. Ammann, L. Oman, D. Shindell, S. Levis, and G. Stenchikov, 2009: Did the Toba volcanic eruption of _74 ka B.P. produce widespread glaciation? J. Geophys. Res., 114, D10107, doi:10.1029/2008JD011652. Timmreck, C., et al., 2010: Aerosol size confines climate response to volcanic supereruptions. Geophys. Res. Lett., 37, L24705, doi:10.1029/2010GL045464.

**Response #11:** We thank the reviewer for these suggestions, and we have now included these references as well as an enhanced discussion.

**Comment #12:** In any case, I find the Haslam and Petraglia (2010) Figure 1 very convincing that it got cold before the eruption. By the way, that reference is missing from the reference list. Why does the timing of the Toba eruption in Fig. 6 here differ from that in Fig. 1 of Haslam and Petraglia (2010)? Which is correct, and why?

**Response #12:** As we note in our previous submission's text, the timing of the Toba eruption S spike within the ice cores is based on Svensson et al., 2013, Climate of the Past, which uses a very thorough analysis of both Arctic and Antarctic ice core records. This represents the most up-to-date assessment of the timing of the Toba eruption relative to Greenland climate change, so we use this, in agreement with other recent publications (e.g., Polyak et al, 2017, Geology). The Haslam and Petraglia 2010 paper precedes the Svensson et al. 2013 analysis, and uses an older chronology. So our assessment is correct, and Haslam and Petraglia 2010 is incorrect, at least in terms of reflecting the most recently accepted chronology. Because the Haslam and Petraglia 2010 paper is demonstrably out-of-date (through no fault of their own – the paper preceded the chronological revisions we used), it is not worth discussing at length in this manuscript.

**Comment #13:** The paper ignores all the work that has shown that the 1257 Samalas eruption caused the Little Ice Age (Zhong et al., 2011; Miller et al., 2012; Slawinska and Robock, 2017). What does this tell us about the claim that a much smaller eruption of Laacher See caused a much larger climate response?

Miller, G. H., Á. Geirsdóttir, Y. Zhong, D. J. Larsen, B. L. Otto-Bliesner, M. M. Holland, D. A. Bailey, K. A. Refsnider, S. J. Lehman, J. R. Southon, Ch. Anderson, H. Björnsson, and T. Thordarson, 2012: Abrupt onset of the Little Ice Age triggered by volcanism and sustained by sea-ice/ocean feedbacks. Geophys. Res. Lett., 39, L02708, doi:10.1029/2011GL050168. Slawinska, J., and A. Robock, 2017: Impact of volcanic eruptions on decadal to centennial fluctuations of Arctic sea ice extent during the last millennium and on initiation of the Little Ice Age. J. Climate, doi:10.1175/JCLI-D-16-0498. http://journals.ametsoc.org/doi/abs/10.1175/JCLI-D-16-0498.1

Zhong, Y., G. H. Miller, B. L. Otto-Bliesner, M. M. Holland, D. A. Bailey, D. P. Schneider, and A. Geirsdottir, 2011: Centennial-scale climate change from decadally-paced explosive volcanism: a coupled sea ice-ocean mechanism. Clim. Dyn., 37, 2373-2387.

**Response #13:** We thank the reviewer for flagging this up; we have now included these suggested references, as well as some others as well. As for the second part of the comment, we refer the reviewer to our response to their Comment #7 (i.e., the size of the feedback is dependent on the background conditions, not the size of the eruption, provided the eruption is sufficiently large to trigger ice growth and interrupt ocean circulation). We have included a substantial amount of new text in the revised submission:

*'A similar mechanism may have also contributed to Little Ice Age cooling (Miller et al., 2012), with recent research suggesting that a coupled sea ice/AMOC mechanism could extend the cooling effects of volcanic aerosols by over 100 years during the Little Ice Age (Zhong et al., 2011). This perspective is supported by modelling results suggesting that a large volcanic forcing is required to explain Little Ice Age cooling (Slawinska and Robock, 2017). Lehner et al. (2013) identify a sea ice/AMOC/atmospheric feedback that amplified an initial negative radiative forcing to produce the temperature pattern characterising the Little Ice Age. Similarly, large volcanic eruptions in 536, 540, and 547 AD are hypothesised to have triggered a coupled sea ice/AMOC feedback that led to an extended cold period (Buntgen et al., 2016). Recent research also highlights the possibility that volcanism followed by a coupled sea ice/ocean circulation positive feedback triggered hemispheric-wide centennial to millennial-scale variability during the Holocene (Kobashi et al., 2017). If a sea ice/AMOC feedback was active following volcanic eruptions during the 6th Century, the Little Ice Age, and the Holocene, the intermediate ice volume and transitional climate characteristic of the last deglaciation should have amplified their effects. This perspective is consistent with previous observations, including those of Zielinski et al. (1996) who noted that when the climate system is in a state of flux it is more sensitive to external forcing, and that any post-volcanic cooling would be longer lived. Importantly, Rampino and Self (1992) stated "Volcanic aerosols may also contribute a negative feedback during glacial terminations, contributing to brief episodes of cooling and glacial readvance such as the Younger Dryas Interval". Our results are entirely consistent with this perspective, and here we highlight a candidate volcanic eruption whose timing coincided with the onset of YD-related cooling. However, despite increasingly tangible evidence that eruptions can affect AMOC strength and sea ice extent, the exact nature of any positive feedback is still unclear. Future research should prioritize the identification and characterisation of this elusive, but potentially commonplace, feedback that amplifies otherwise subtle NH temperature shifts.'*

**Comment #14:** It would have been nice to have used hanging indents or additional spacing for the

reference list to make it easier for the reader to find each paper in the list. In addition, there are another 35 comments in the attached annotated manuscript that need to be addressed.

**Response #14:** We agree with the reviewer, and have added a space between the references for increased leginility. We have addressed all of the comments contained in the annotated manuscript.

---

## Author Response (AR1)

Department of Earth Sciences

Shaped by the past, creating the future

Dr James U.L. Baldini
Associate Professor (Reader)
Department of Earth Sciences
University of Durham
Durham DH1 3LE, UK
+44 (0) 191 334 2334
james.baldini@durham.ac.uk

March 2018

Dear Dr Thornalley,

Thank you for your continued consideration of our manuscript entitled 'Re-evaluating the link between the sulphur-rich Laacher See volcanic eruption and the Younger Dryas' for publication in *Climate of the Past,* and for your comments. We feel that this manuscript is an extremely important submission that provides a straightforward but comprehensive explanation for the Younger Dryas event that will generally be well-received (as it already has been by two reviewers, and by other readers who have contacted us independently of the online system) and provocative, and we welcome the opportunity to submit a revised manuscript. The topic is also controversial, and we are therefore glad that it will go out to further review, which will undoubtedly help us present as compelling of a case as possible.

Below this cover letter is a point-by-point response to the comments by the reviewers, as well as to your editorial comments. Many of the responses are similar to those provided earlier in our responses to the reviewer's comments on the online discussion. You will note from the marked-up version of the manuscript (also included below) that the changes made are extensive, and in fact substantial changes have been made that were not requested in the review process, including a new figure that helps shift the emphasis from magnitude to sulphur yield (as suggested by Prof. Pyle). In many cases we provide examples of the new text that was included in response to the comment, but in some cases the changes were too extensive or unevenly distributed throughout the manuscript to be clearly listed, although of course the nature of the changes was always described. Please do let us know if you require any further details on any of the changes made, or if you have any other questions regarding our resubmission.

Best regards,

Dr James Baldini (Corresponding author)

**Responses to Editorial, Reviewer, and short comments**

**Responses to Editorial comments:**

**Comment #1:** Firstly, a couple of minor points: (a) as pointed out, the existence of MWP-1B is questioned (probably no abrupt jump in sea-level) - Bard et al 2010 Science.

**Response #1:** This has now been addressed and it is highlighted in the figure caption that "the source, timing, and even occurrence of the meltwater pulse are still debated", and in the main text that "this requires further research, particularly because the source, duration, timing, and even existence of MWP-1B are still unclear".

**Comment #2:** (b) Although this may not arise on your revised manuscript, the role of the AMOC in the Little Ice Age (LIA) has been questioned, with the latest reconstructions suggesting that AMOC did not weaken (Rahmstorf et al 2015, Nat. Clim. Change), and the LIA may instead be caused by changes in horizontal subpolar gyre circulation and sea-ice feedbacks (Moreno-Chamarro et al 2017, Clim. Dynamics).

**Response #2:** We have included the following text: "*Rahmstorf et al. (2015) argue that no identifiable change in AMOC occurred during the Little Ice Age, suggesting that the drivers were related to sea ice and atmospheric rather than oceanic, although this requires further research to confirm.*" We thank the editor for bringing this paper to our attention.

Although we see the relevance, we have opted to not include the Moreno-Chamarro et al 2017 paper because it relies exclusively on modelling results, which in our opinion appear very ambiguous. This paper of course relates to the LIA, and so is not directly relevant to the YD.

**Comment #3:** The broader issue with this submission is that it is not providing new data or evidence, but instead is using an updated age model to propose a new hypothesis (albeit one that had been earlier considered and rejected, as discussed). I am not overly concerned about this, since as stated by one of the reviewers, and highlighted by yourself, this is a topic that has perhaps been neglected and it is useful to put this idea out there to be discussed.

**Response #3:** We now realise that in fact we may have actually undersold the hypothesis in the previous submission. To our knowledge, no detailed discussion of the eruption as a YD trigger exists prior to this submission, only very brief mentions. Up until now, the LSE had been mentioned only in very oblique ways in papers focussing on other subjects. For example, Bogaard et al 1990 state "Whether the eruption of LSV

contributed to the stage of climate deterioration known as the Younger Dryas....is a matter of current investigation" but that is the only mention of the YD in the paper, and as far as we are aware there was no further investigation. Possibly this was curtailed by research suggesting that the eruption occurred too early: in 2002 Baales et al. (Quat. Res.) state: "Correlation of terrestrial archives with the Greenland ice-core records and
improved calibration of the radiocarbon timescale permit a precise, accurate age determination of the Laacher See event some 200 yr before the onset of the Younger Dryas cold episode" reflecting the thinking at the time. Even more enlightening is Schmincke et al. 1999 (Quat. Int.): "The Younger Dryas cooling period clearly was not triggered by LSE as formerly thought because it started ca. 180 years after the eruption."

These quotes reflect the fact that the Laacher See hypothesis was not previously developed, and also the lack of
understanding that climate change associated with the YD was time transgressive. Therefore, we are the first to suggest the Laacher See hypothesis as a triggering mechanism in any meaningful way (to our knowledge). Although we considered changing the title, we still feel that the word 'Re-evaluating' is the correct word to use, given that earlier papers had discarded the possibility of the eruption as a trigger. The importance of this contribution is highlighted by a recent publication investigating a cometary airburst as the Younger Dryas
trigger. Wolbach et al. 2018 (*Journal of Geology*) (incorrectly) state that: "Furthermore, Moore et al. (2017) examined three samples of tephra from the Laacher See eruption in Germany, which occurred 200 y before the cosmic-impact event and potentially could have contaminated the YDB layer with volcanic Pt." we now know that the LSE was coincidental with the YD, and did not precede it by 200 years, and yet a high-profile 2018 publication fails to consider that the eruption was coincident with the eruption, and could have contributed
directly to the Pt spike discussed.  We now discuss this history in more depth in the revised submission (and discuss the new Wolbach paper), including the following text:

"*Earlier research briefly alluded to the eruption as a possible causative mechanism for the YD (e.g., Berger, 1990; Bogaard et al., 1989). However, because the meltwater pulse hypothesis was already popularised, and because the effects of volcanic eruptions on climate would escape detailed quantification until after the 1991 Pinatubo*
*eruption, the ~12.9 ka BP Laacher See eruption as a YD trigger never gained traction. Importantly, the concept was effectively dismissed after lacustrine evidence across central Europe appeared to indicate that the YD's clearest expression appeared ~200 years after the Laacher See Tephra within the same sediments (e.g., Brauer et al., 2008; Brauer et al., 1999a; Hajdas et al., 1995). For example, Schmincke et al. (1999) state that "The Younger Dryas cooling period clearly was not triggered by LSE as formerly thought because it started ca. 180*
*years after the eruption.", reflecting the accepted sequence of events at that time.*"

**Comment #4:** However, it does mean that the merits of this manuscript depend heavily on it providing an accurate overview of the state of the art regarding the onset of the Younger Dryas (YD). In its current form I do not think the paper does this and there are major omissions in terms of providing the necessary context.

**Response #4:** We believe that the substantial revisions to the manuscript now present an extremely comprehensive overview of the onset of the YD.

**Comment #5:** The onset of the YD is associated with the end of the preceding Bolling-Allerod (B-A), which is
usually considered to be a DO warm event/interstadial. There is a large body of literature that proposes the BA-YD oscillation is simply another DO cycle that occurred as climate passed through an intermediate climate state. Therefore the mechanisms involved for the BA to YD transition may be similar to those acting at the end of earlier DO interstadials. This is consistent with your assertion about the possible role of volcanic eruptions in ending both the BA and earlier DO interstadials. However, the duration of an interstadial has been shown to be
dependent on its rate of cooling (ie DO interstadials do not end with random timing, and they occur after a period of gradual cooling). This evidence is difficult to reconcile with the suggestion that a volcanic event triggered the YD/stadial transition, in isolation. I therefore suggest that you consider a slight modification to your hypothesis, which stresses that climate was probably close to switching back to stadial conditions anyway, and that the occurrence of a volcanic eruption was the (small) trigger that determined the precise timing of the
transition (which incidentally may help explain why only some stadials have been linked to volcanic eruptions in your earlier 2015 paper). In the context of the YD, many climate record, especially those around the North Atlantic and linked to the AMOC, show a gradual change (deterioration) in climate through the BA which fits with this generalized evolution of a DO interstadial.

**Response #5:** This is an excellent point, and one that we completely agree with. We have now included this in a
discussion along with appropriate references. It is worth noting that the Toba eruption may also have occurred during a cooling trend, and we believe that in these situations a Northern Hemisphere eruption pushes the climate towards its insolation-mediated baseline (cold). This point is also critical for assessing an alternative hypothesis, that of a cometary airburst. Because only a minor trigger is required to shift the climate back to its pre-B-A glacial state, it is difficult to reconcile the fact that the Oldest Dryas was 4.5C colder than the YD with a
catastrophic cometary airburst. Instead, it seems likely that the 'nudge' was reasonably minor. We include this new text in section 3.7 (Compatibility with other hypotheses) as a new discussion comparing the two hypotheses:

*"Another issue with the YDIH is that the YD was simply not that anomalous of a cold event, and therefore does not require an unusually powerful trigger. Over the last 120 ka, 26 Greenland Stadial (GS) events occurred*
*(Rasmussen et al., 2014), of which the YD was the most recent. This does not exclude the possibility that the YD was forced by an impact event, but the most parsimonious explanation is that most stadial events had similar origins, implying a much more commonplace trigger than an impact. Furthermore, nitrogen isotopes suggest that Greenland temperature was 4.5ºC colder during the Oldest Dryas (18 to 14.7 ka BP) than the YD (Buizert et al., 2014), again suggesting the transition to the YD does not require an extreme forcing. It is also worth noting*
*that the YD may actually represent a return to the insolation-controlled baseline, and that potentially the*

*Bølling-Allerød (B-A) warm interstadial (i.e., GI-1, from 14.642 to 12.846 ka BP), immediately preceding the YD, was the anomaly (Thornalley et al., 2011; Sima et al., 2004). Although an in-depth discussion of the B-A is outside the scope of this study, the B-A may represent an interval with a temporarily invigorated AMOC (an 'overshoot') (Barker et al., 2010), which, after reaching peak strength, began to slow down back towards its*

*glacial state because of the lack of a concomitant rise in insolation (Knorr and Lohmann, 2007; Thornalley et al., 2011). The rate of this slowdown was independent of the final trigger into the YD, and indeed much of the cooling back to the glacial baseline was achieved by ~13 ka BP. Consequently, only a small 'nudge' may have been required to expedite the return to the cold baseline state, consistent with a very sulphur-rich volcanic eruption (occurring every few hundred years) but not necessarily with a rare, high-consequence event. In other*

*words, the B-A may represent the transient anomaly, and the conditions within the YD represented typical near-glacial conditions, obviating the need for an extreme YD triggering mechanism. The cooling after the B-A cannot account for the rapid cooling observed at around 12.9 ka BP clearly visible in the NGRIP ⍰18O and nitrogen isotope data as well as in numerous other North Atlantic records, indicating that another forcing expedited the final cooling into the YD, which we argue was the LSE."*

**Comment #6:** Related to these ideas is the concept that the BA was an overshoot of the AMOC that occurred during termination of the last glacial, such that its demise was inevitable because climate had not yet shifted to an interglacial state where the on mode of AMOC was stable i.e. the BA should be viewed as the event that needs a trigger, and the YD was simply a return to glacial conditions not requiring a trigger (or only a minor one). This framework for the deglaciation is fairly well established, and I think these concepts ought to be included to ensure that the reader has a more complete and accurate context with which to assess the validity of your hypothesis.

**Response #6:** Agreed. We have now included text that clarifies the climate context under which the LSE and the YD occurred, as well as appropriate references.

**Comment #7:** Suggested references (by no means an exhaustive list) that deal with these concepts are: Ganopolski and Rahmstorf 2001, Nature; Schulz et al. 2002 GRL, 10.1029/2001GL013277; Sima et al. 2004, EPSL; Knorr & Lohmann 2007 G^3; Liu et al. 2009, Science, DOI: 10.1126/science.1171041; Barker et al. 2010, Nature Geoscience; Thornalley et al. 2011, Science; Buizert & Schmittner 2015 Paleoceanography; Barker et al., 2015,

Nature.

**Response #7:** We thank the editor for these suggestions. We have now included most of these references into the text. We felt that some of these references were not needed, because the ones we do cite make a sufficiently strong case for the B-A as being particularly anomalous rather than the YD, and we did not want to distract from the main points.

**Reviewer #1 Comments and Responses:**

**Comment #1:** Baldini and colleagues present a previously published volcanogenic sulphate record from GISP2 ice cores and compile age estimates for the Laacher See tephra to argue that the LS eruption is recorded in Greenland ice cores as a large sulphate spike positioned approximately at the onset of Greenland Stadial 1 (i.e. Younger Dryas). The authors finally suggest that the LS eruption triggered the YD through a chain of feedbacks resulting from the initial volcanic-induced cooling in the Northern Hemisphere. Even though their hypothesis is tantalizing I find the manuscript excessively speculative and the conclusions very much stretch what can be observed in the reconstructions.

**Response #1:** We appreciate that this topic is controversial, and the worst-case scenario would be if the manuscript were published without undergoing a rigorous review process. We also thank the reviewer for suggesting that the hypothesis is 'tantalising'; we agree and we hope to convince Reviewer #1 that the conclusions are not overly speculative. We would like to note upfront that we did not 'cherry-pick' the records that were used in the manuscript, rather these were chosen as the records with the most robust chronologies both in absolute terms and with respect to the timing of GS-1 versus the LSE.

**Comment #2:** I should also point out that the putative link between the LST and a volcanic sulphate spike in GISP2 records was originally suggested by Brauer et al. (1999). In my opinion, the present manuscript doesn't offer anything new but I brief review of the climatic implications of volcanic cooling and far-reaching speculations on the triggers for the YD.

**Response #2:** Should this manuscript be published, it would open the door to the consideration of a new trigger for the Younger Dryas Event. Of the most recent reviews of the Younger Dryas and its possible causes (e.g., Carlson 2010, *Geology*; Fiedel 2011 *Quaternary International*; Broecker et al., 2010 *Quaternary Science Reviews;* Renssen et al., 2015 *Nature Geoscience*), none even mentions volcanic forcing; we therefore disagree wholeheartedly with the reviewer's statement that this '…manuscript doesn't offer anything new…'. This manuscript presents a very novel and provocative hypothesis, and does not represent 'business-as-usual'. Although there are certainly elements of a review, we are the first paper to discuss the LSE as a YD trigger in any detail, and we are the first to detail the positive ice-AMOC feedback following the LSE within the context of the YDE. We also identify the sulphate spike associated with the Laacher See eruption, using the most recent chronology for the GISP2 ice core. Reviewer #1 is correct that Brauer et al 1999 did consider this same spike as being potentially linked to the LSE, but ultimately they decided that it occurred too close to the YD boundary and concluded that an earlier spike represented the LSE (this is now discussed); so I think it is correct to say that we are the first to attribute the LSE to this particular spike, though we will now discuss the fact that Brauer et al (1999) considered the spike before us.

This manuscript uses previously published data to reach new conclusions, and we therefore feel that this goes considerably above and beyond a 'brief review', and we feel very strongly that we do offer something new. For example, several recent papers incorrectly state that the LSE occurred ~200 years before the start of the YD (e.g., Moore et al 2017; Wolbach et al., 2018a, Wolbach et al, 2018b); this would probably not have happened if our submission had been published.

**Comment #3:** Assigning the LS eruption to a nearly synchronous large sulphate peak in GISP2 is a flawed assumption until proven otherwise by tephochronological analyses. The authors should consider alternative hypotheses. For instance, volcanic records from Greenland are particularly sensitive to Icelandic eruptions due to the eruptive frequency and proximity (e.g. Abbott and Davies, 2012). Hence, even small-size Icelandic eruptions characterised by moderate sulphate emissions can result in disproportionally large sulphate anomalies in ice core stratigraphies.

**Response #3:** We agree that tephochronological analyses would confirm our identification, and we have now included this statement in the text. We note that although this is certainly an excellent idea, the lack of tephochronological data did not prevent most other researchers from attributing sulphate spikes to individual eruptions (for example, Brauer et al., 1999, mentioned by the reviewer in Comment #2, Svensson et al., 2013 Climate of the Past, and many others). We have also provided ideas for confirming our hypothesis in general in the conclusion.

**Comment #4:** 2. I think the paper would greatly benefit from exploring some of the mechanisms the authors discuss (e.g., southward wind shifts, AMOC decline, sea ice expansion, etc.) by looking into climate model output and/or historical reanalyses data sets. The authors present output data from MAECHAM4 simulations but they don't provide any additional analysis of the atmospheric parameters in the model. CMIP historical climate simulations and PMIP last-millennium simulations offer valuable model data to examine the role of volcanic eruptions on the coupled atmosphere-ocean system. Model output based on volcanically-forced transient simulations with earth System Model (MPI-ESM) (Jungclaus et al., 2010) (available online) may also be useful.

**Response #4:** We now include a revised and extended discussion of relevant, previously published model outputs over the last two millennia. The climate and AMOC response following volcanic eruptions is different in each of the models; this is now discussed:

*"Our hypothesis that the YD was triggered by the LSE and amplified by a positive feedback is further supported by modelling results suggesting that a combination of a moderate negative radiative cooling, AMOC weakening,*

*and altered atmospheric circulation best explain the YD (Renssen et al., 2015). AMOC consists of both*

*thermohaline and wind-driven components, and atmospheric circulation changes can therefore dramatically affect oceanic advection of warm water to the North Atlantic. Recent modelling suggests that reduced wind stress can immediately weaken AMOC, encouraging southward sea ice expansion and promoting cooling (Yang et al., 2016), illustrating a potential amplification mechanism following an initial aerosol-induced atmospheric circulation shift. Twentieth Century instrumental measurements further support this by demonstrating that westerly winds strength over the North Atlantic partially modulates AMOC (Delworth et al., 2016).*

*Initiation of the positive feedback requires volcanic aerosols to remain in the atmosphere for at least one summer season.  Evidence based on the seasonal development of vegetation covered by the LST suggests that the LSE occurred during late spring or early summer (Schmincke et al., 1999), and varve studies similarly suggest a late spring or early summer eruption (Merkt and Muller, 1999). Available evidence therefore suggests that the eruption occurred just prior to maximum summer insolation values, maximising the potential scattering effects of the volcanogenic sulphate aerosols. Even if it were a winter eruption, for historical eruptions similar in magnitude to the Laacher See eruption, aerosols remained in the atmosphere longer than one year, regardless of the eruption's latitude. For example, aerosols remained in the atmosphere for ~three years after the Pinatubo eruption (15°N, 120°E) (Diallo et al., 2017), which probably released considerably less SO2 than the LSE (Figure 2). Measurable quantities of aerosols remained in the atmosphere for approximately three years even after the 1980 Mount St. Helen's eruption (46°N, 122°W) (Pitari et al., 2016), which injected only 2.1 Mt SO2 (Baales et al., 2002; Pitari et al., 2016) and erupted laterally (Eychenne et al., 2015). In short, the LSE eruption probably occurred during the late spring or early summer, but even if the eruption were a winter eruption, the LSE's aerosols would have certainly persisted over at least the following summer, with the potential to catalyse the positive feedback we invoke."*

**Comment #5:** I think some of the arguments proposed here (as well as the records presented in Figure 3) have conveniently been picked to craft a story where the LS eruption stands out as the INITIAL cooling that triggered the YD. This does not faithfully reflect the state of the knowledge around the dynamics that took place prior to and at the onset of the YD. Several reconstructions show that a gradual but substantial cooling across Northern Europe and in the Nordic Seas preceded the start of GS-1. Pollen records from the British Isles indicate cooling as early as 13,200 years BP (Walker et al., 2012). A drop in air temperatures a few centuries before GS-1 has also been recorded in chironomid-based temperature records from Norway (Bakke et al., 2009; see supplementary information), Sweden (Muschitiello et al., 2015), and the Netherlands (Heiri et al., 2007). Similarly, sea surface temperature records from the Norwegian Sea indicate a rapid cooling approaching YD values as early as 13,500 years BP (Bakke et al., 2009) with temperature values dropping by ca. Well-dated paleoceanographic records from the coast of Norway also show a progressive aging of surface waters starting at ca. 13,200 years BP (Bondevik et al., 2006), which is suggestive of a slowdown of surface-water circulation and reduced advection of warm subtropical waters prior to the start of GS-1. I therefore believe that the authors should do a better job in contextualizing the LS eruption in the regional climate picture since proxy reconstructions from Central Europe (e.g. Grafenstein, 1999; Rach et al., 2014) are evidently not fully representative of large-scale North Atlantic climate. In particular, the records mentioned above clearly suggest
that cooling and climate deterioration was long underway before the start of the YD, which implies that the
cooling associated with the LS eruption cannot be the trigger for the YD.

**Response #5:** We appreciate the very thorough suggestions for other records to consider, and here we provide
reasons why the records are, or are not, relevant to the Laacher See hypothesis. Again, we assure the reviewer
that we have not knowingly excluded records that contradict our hypothesis. Rather, the records chosen are
those with the most robust chronologies both in absolute terms and with respect to the timing of GS-1 versus
the LSE. All of the records used contain both a high resolution regional temperature proxy record and either the
Laacher See Tephra directly, or an excellent layer-counted chronology.

**Muschitiello et al., 2015 Nature Communications:** We do not doubt that meltwater pulses did occur and
undoubtedly affected climate during the last deglaciation. In fact, we suggest that the YD was terminated by a
Southern Hemisphere meltwater pulse that triggered long-term warming. Muschitiello et al. argue that
meltwater forcing affected climate from 13.1 to 12.880 ka BP. We do not argue against this, but nothing in this
paper contradicts our manuscript. In fact we note that, once again, a pronounced climate shift occurs coincident
with the Laacher See eruption at 12.880 ka BP. The large inflection point in Figure 4 (Panel d and elsewhere) is
indistinguishable from the date of the eruption. It is currently difficult to disentangle whether the forcing was a
meltwater pulse from the Fennoscandian Ice Sheet (or elsewhere), a bolide impact, or the Laacher See eruption,
and we hope that our research will promote future debate assessing the pros and cons of each hypothesis.

We also note that it is conceivable that the Fennoscandian Ice Sheet was melting from 13.1 to 12.88 ka BP due
to rising insolation, releasing meltwater. The end of this meltwater pulse coincides perfectly with the cooling
(GS-1) that we argue was triggered by the LSE; it is therefore reasonable that post-eruptive cooling, combined
with a positive feedback, could temporarily reverse Fennoscandian Ice Sheet melting. We also note that
Muschitiello et al. state that summer temperature dropped by 2 degrees at 12.883±0.035 ka BP, "suggesting
substantially drier and colder summer conditions." It is conceivable that this is reflecting the direct aerosol
effects of the LSE.

**Heiri et al. 2007:** The paper presents an interesting chironomid-based temperature record of the Younger Dryas
from the Netherlands. Unfortunately, it is too low resolution to be particularly useful to this manuscript. The
dating is also not as high precision as the records that we have chosen, though we note that the decrease in July
temperature starts just after the eruption based on their chronology. Still, we choose not to include this record
and others like it because of the low resolution and more uncertain chronology.

**Walker et al 2012:** Unfortunately the low resolution and ambiguous results make this paper a low priority for
inclusion. There are hundreds of Younger Dryas climate reconstructions globally, and including all of them is
simply not possible. Pollen reconstructions in particular are problematic due to the generally lower resolution of the datasets, the often less well-constrained chronologies, and the local nature of the proxy. We do not see how this paper provides a significant challenge to either a LSE, a bolide impact, or a meltwater trigger for GS-1 and the YD.

**Bondevik et al., 2006:** This is quite an interesting paper that we somehow had missed previously, and we thank the reviewer for bringing it to our attention. Their Figure 3 is particularly striking, and shows a pronounced inflection point in the radiocarbon concentrations precisely at the Laacher See eruption (Panels B, C, and D). They state that *'A high reservoir age during the YD could be explained by a combination of increased sea-ice*
*cover and reduced advection of surface water to the North Atlantic'*, both of which are entirely consistent with our proposed positive feedback mechanism. We have included this reference to the radiocarbon data within the context of a more substantial discussion of the positive feedback.

**Bakke et al., 2009:** The correlations between the data presented in this paper and the Laacher See eruption are
remarkably strong. In particular, Supplemental Figures S5 (noting of course that their data is in years b2k) and S7 show that the largest inflection point is coincident with the eruption. We thank the reviewer for bringing this supplemental material to our attention.

**Specific comments:**
L30: An alternative route has been proposed involving the drainage of the Baltic Ice Lake, the timing of which precisely coincide with the start of GS-1 (Muschitiello et al., 2016).
**Response:** This is now discussed.

L42-47: Without entering into the discussion on the credibility of the "impact" theory, I think that the evidence is still undermined by the poor chronological accuracy of the platinum/iridium anomaly.
**Response:** We now have an entire new section of almost 1,000 words discussing the other leading theories in
more detail.

L54: What stratigraphic frameworks? Please specify and provide reference.
**Response:** We have now provided more context regarding this statement.

L57-58: ": : :consistent with the LS eruption: : :" Please add here "as recorded in varved and 14C dated lake records".
**Response:** We have reworded this sentence.

L148: Please provide reference here on the impact of volcanic forcing on AMOC (e.g. Otterå et al., 2010).

**Response:** We have now added this reference as well as several others in a more detailed discussion of the feedback.

L162: Please specify what model. This is not clear.
**Response:** This has been rephrased to be clearer.

L176-192: This section is all very speculative. Please see my comment on the possibility of examining climate model output to support these claims.
**Response:** We have added a substantially strengthened section regarding relevant published climate model results.

L181: None of the studies cited here present direct evidence of sea ice changes (only indirect and mainly based on terrestrial reconstructions). Please consider referring to marine reconstructions (e.g. Cabedo-Sanz et al., 2013).
**Response:** We have now added this reference.

L223: I'm not convinced that the data in Figure 5a and b follow a Gaussian distribution. Rather, the frequency distributions seem skewed. Also the authors cannot claim that "a Gaussian distribution exist" when they fit a Gaussian-best-fit model to their data. They should use a resampling/bootstrap method to draw from their empirical distribution and only then establish its shape.

**Response:** We have changed the wording in the main text. We have left the wording the same in the figure caption, because the grey-filled curve is a Gaussian distribution, and this describes the shape of the data according to the software used.

L219-230: I suggest estimating the frequency of Greenland cooling relative to the number of volcanic events
occurring at the end of the preceding interstadial. This should somewhat inform on the potential link between volcanisms and the onset of stadial cooling.
**Response:** Previously published research (e.g., Zielinski et al., 1997; Sternai et al., 2016) has already made the connection between increased frequency of volcanism and deglaciation, and this is mentioned a couple of times in the text. The number of volcanic events relative to the number of Greenland cooling events is discussed in
detail in one of our prior publications (Baldini et al., 2015) and already mentioned in this manuscript.

L246: The rate of cooling (as seen in the d18O ice-core stratigraphies) is similar among most of the stadials, not exclusively between GS-1 and GS-20.
**Response:** Agreed, and we would argue that the cause is similar. Baldini et al. 2015 showed that every well-
dated, large volcanic eruption over the period 30-80 ka BP is within dating errors of a stadial. The post-eruptive positive feedback is likely to be identical between GS-1 (Laacher See), GS-20 (Toba), and many (most?) other stadials – GS-12 (Opala), GS-9 (Campi Flegrei), etc. Unfortunately most eruptions are still very poorly dated, but we would venture that every sulphur-rich magnitude 6 (or above) eruption occurring during intermediate ice volume conditions resulted in the positive feedback. We have added a substantial amount of text that will hopefully make this clearer.

L247-249: The initial cooling can be tested using ice-core (e.g. d15N) and other proxy based temperature reconstructions and compare the magnitude of the temperature change with that associated with large historical eruptions.

**Response:** We have added statements discussing Greenland d15N. We are unsure of the added value in looking for the initial cooling - there is no doubt that the eruption happened, and no doubt that it contained considerably more sulphur than the climatologically important Pinatubo eruption. So it almost certainly did result in cooling, but maximum cooling probably persisted for one year (more subdued cooling would have lasted for ~3 years). This would be difficult to detect in an ice core, and even more difficult to attribute to an eruption. Even if detectable, the cooling could also be ascribed to a meltwater pulse or a bolide impact. Regardless, existing d15N reconstructions (Buizert et al 2014, Science) are fully consistent with the manuscript, and we have included these references in the discussion.

L253: Is this claim based on ice-core d18O profiles? I don't see any substantial warming in Greenland during the second half of GS-1. d18O of ice can be misleading due to changes in moisture source of precipitation. Before making this claim I would check the ice-core temperature records ($\delta^{18}$O diffusion or d15N). As far as I can tell from the d18O data both GS-1 and GS-20 in Greenland were characterised by cold conditions throughout the stadial.

**Response:** There is abundant evidence that the maximum cooling during GS-1 occurred at around 12.65 ka BP, and that this was followed by moderate gradual warming. This is apparent not only in NGRIP and GISP2 $\delta^{18}$O, La Garma Cave (Spain) $\delta^{18}$O, Chauvet Cave $\delta^{18}$O (France), Lake Ammersee $\delta^{18}$O, etc., but also in ice core nitrogen isotope ratios and deuterium records. We have added add some references to this statement to clarify.

L258-260: Again, this is extremely speculative and I don't think there is any evidence supporting this claim.
**Response:** Please see our response to the point above. We do not see how this is 'extremely speculative' when there is abundant evidence supporting the claim. We have now added more references to further support the statement.

L275-284: The magnitude of the spike does not necessarily scale linearly with the magnitude of the eruption and could solely depend on the proximity of the volcanic source or the atmospheric circulation pattern.

**Response:** This is true, and we noted this in our previous submission when we state that the large spike
preceding our candidate LSE spike is a small Icelandic eruption of Hekla. We have added some extra text to the manuscript to ensure that readers are clear on this point.

L290: Age uncertainties for the LST and the sulphate spike should be reported on their respective time scales
(i.e. GICC05, IntCal13, MFM varve time scale, etc.). In addition, time scale offsets between the 14C time scale and GICC05 should be accounted when comparing radiocarbon-based and ice-core ages (e.g. Muscheler et al., 2014).
**Response:** The timing of the sulphate spike relative to the LSE is based on layer counting in the Meerfelder Maar core from the LST to the Vedde Ash, and then layer counting from the Vedde Ash to the position where
the LSE should be (and where we find a sulphate spike). We will clarify this in the section where we discuss the sulphate spike. Muscheler et al., 2014, note that *"…there is no evidence for any significant difference between the GICC05 ice-core and 14C time scales at around 13,000 yr BP."* and we now mention this.

L291: Please see my previous comment. Cooling in the North Atlantic started a few centuries before the onset of GS-1.
**Response:** It is true that cooling following the B-A warm interval had been occurring for some time prior to 12.9 ka BP. However, there is a clear increase in the rate of cooling at 12.9 ka BP, as discussed in innumerable publications on the YD. We now include a discussion on the B-A.
L305-306: As I mentioned above, it would be helpful to look into the frequency of stadials relative to the number of volcanic eruptions during the preceding interstadials.
**Response:** Please see our response to the reviewer's same comment above (comment on L219). This has already been previously, and is discussed and cited in the manuscript.

**Reviewer #2 Comments and Responses:**

**Comment #1:** I am not convinced by this paper, and so recommend rejection or major revisions. Perhaps the timing of the eruption is close to indicators of cooling (not my specialty), but the hand-waving arguments about why this eruption caused cooling and larger ones did not are not convincing.

**Response #1:** We are sorry to read that Reviewer #2 feels that our arguments are overly speculative. We feel that perhaps the reviewer has missed some key points, which probably reflects that these were poorly communicated on our part. We hope that the revised submission as well as our responses will help better communicate our hypothesis, and we welcome the opportunity to improve the manuscript.

Regarding the last part of Reviewer #2's comment, we are unclear as to which eruptions the reviewer is referring to when Reviewer #2 mentions '…larger ones did not…'? We state:

 *'…it is in fact the only known sulphur-rich high latitude eruption coinciding with the most sensitive ice volume*
*conditions during the last deglaciation. '*

and devote an entire section to other eruptions (Section 3.6), so the other large eruptions that the reviewer is referring to is not clear. We have published previous research indicating that there is a strong statistical significance between large Northern Hemisphere eruptions and long-lasting climate change (Baldini et al., 2015,
Scientific Reports) and there is now evidence that Toba also triggered long-lasting climate change (discussed below, in our responses to Comment #7). So there is strong evidence that other larger eruptions did cause cooling (Toba and the several others discussed in Baldini et al., 2015), and we argue that the Laacher See eruption occurred during a particularly sensitive climatological transition, so should also be expected to trigger cooling. We now also include a lengthy section discussing volcanic impacts on climate over the last millennium,
and an enhanced discussion regarding the amount of sulphur in the eruption, which was substantial. Part of that enhanced discussion is pasted below:

"*It is worth highlighting that a similar mechanism may have also contributed to Little Ice Age cooling (Miller et al., 2012), with research suggesting that a coupled sea ice/AMOC mechanism could extend the cooling effects of*
*volcanic aerosols by over 100 years during the Little Ice Age (Zhong et al., 2011). This perspective is supported by modelling results suggesting that a large volcanic forcing is required to explain Little Ice Age cooling (Slawinska and Robock, 2017). Lehner et al. (2013) identify a sea ice/AMOC/atmospheric feedback that amplified an initial negative radiative forcing to produce the temperature pattern characterising the Little Ice Age. Rahmstorf et al. (2015) argue that no identifiable change in AMOC occurred during the Little Ice Age, suggesting that the drivers*
*were related to sea ice and atmospheric rather than oceanic, although this requires further research to confirm. Large volcanic eruptions in 536, 540, and 547 AD are hypothesised to have triggered a coupled sea ice/AMOC feedback that led to an extended cold period (Buntgen et al., 2016). Recent research also highlights the*

*possibility that volcanism followed by a coupled sea ice/ocean circulation positive feedback triggered
hemispheric-wide centennial to millennial-scale variability during the Holocene (Kobashi et al., 2017). If a*

*feedback was active following volcanic eruptions during the 6th Century, the Little Ice Age, and the Holocene,
the intermediate ice volume and transitional climate characteristic of the last deglaciation should have amplified
their effects. This perspective is consistent with previous observations, including those of Zielinski et al. (1996)
who noted that when the climate system is in a state of flux it is more sensitive to external forcing, and that any
post-volcanic cooling would be longer lived. Importantly, Rampino and Self (1992) stated "Volcanic aerosols may*

*also contribute a negative feedback during glacial terminations, contributing to brief episodes of cooling and
glacial readvance such as the Younger Dryas Interval". Our results are entirely consistent with this perspective,
and here we highlight a sulphur-rich volcanic eruption whose timing coincided with the onset of YD-related
cooling. However, despite increasingly tangible evidence that eruptions can affect AMOC strength and sea ice
extent, the exact nature of any positive feedback is still unclear. Future research should prioritize the*

*identification and characterisation of this elusive, but potentially commonplace, feedback that amplifies
otherwise subtle NH temperature shifts."*

**Comment #2:** Indeed more work is required, including climate model simulations that include all the relevant
processes.

**Response #2:** We agree that climate model simulations would benefit the research, but we have decided not to
include them for three reasons. i) Neither the meltwater forcing nor the bolide impact hypotheses used climate model simulations to support the initial hypotheses initially, yet both hypotheses led to fruitful discussions and
future elaborations, including climate model research. We therefore feel that climate model simulations would
make for interesting future work, and indeed we are looking into this ourselves. But we do not feel that they are
required for this current submission. ii) There is a good chance that different climate models would return
different answers, in which case modelling is better left for future researchers who can devote considerable attention towards developing robust and replicable model outputs. iii) We may not know the details of all the
relevant feedbacks, and therefore any model might be incomplete. This issue is actually brought up by the
reviewer in their own phraseology: "…climate model simulations that include *all the relevant processes*." The
issue is that we may not know all the relevant processes. For example, if we knew that there was in fact a
pronounced positive feedback following Northern Hemisphere eruptions during intermediate ice volume conditions, we would know that this feedback was responsible for the Younger Dryas Event as well as other
Greenland stadial events. Possibly the reason that we do not know what forced these events is that we do not
know about the relevant process, and if that is the case, modelling could be misleading. This is point is further
illustrated by a recent paper by Diallo et al., 2017 GRL, that states *"Thus, climate model simulations need to
realistically take into account the effect of volcanic eruptions, including the minor eruptions after 2008, for a*

*reliable reproduction of observed stratospheric circulation changes."* If ambiguous modelling results, and the incorporation of unrealistic eruptions effects into models, are demonstrably an issue over the last decade, they will be even more of an issue during the less-well understood YD interval where a key feedback may remain unquantified. For these reasons, we feel strongly that modelling is best left for future work where intercomparison between different models is possible. We hope that this paper, if published, would encourage
modellers to consider the mechanisms and feedbacks proposed, and include these in future models.

**Comment #3:** Even if the timing was close, there is no proof that this was not just a coincidence. The authors
claim that the climate system was particularly sensitive to volcanic forcing at the time, but this is just speculation. Where are the model results to show this?
**Response #3:** Both of the other leading hypotheses for a Younger Dryas trigger also rely on the coincidence of a trigger with the advent of cooling. For example, the meltwater pulse hypothesis relies on the timing of a meltwater pulse with the start of the Younger Dryas. However, unlike the Laacher See hypothesis, there is little
agreement within the community regarding the source of that meltwater or in fact whether it even occurred simultaneously as the start of GS-1. The bolide impact hypothesis also relies on the coincidence of evidence for a meteor airburst or impact with the Younger Dryas initiation, but whether or not this even occurred is extremely controversial. The Laacher See hypothesis does rely on coincidence, but the event itself is much more clearly expressed than the other leading hypotheses. In fact, it is the only hypothesis that features a trigger that
is universally accepted as actually having happened. We have now included a lengthy (almost 1,000 words) new section ('3.7 Compatibility with other hypotheses') to better discuss how the Laacher See hypothesis is competitive relative to the other leading proposed triggers.

Please see our response to why we choose not to include models above (Response #2).
Finally, we are not the first to suggest that the climate system is particularly sensitive to volcanic forcing during climatological transitions, and this is not 'speculation'. It is well established that millennial-scale climate change was most sensitive to a forcing during intermediate ice volume conditions, and we simply propose that that forcing was volcanism. Zielinski et al. (1996) noted that when the climate system is in a state of flux it is more
sensitive to external forcing, and that any post-volcanic cooling would be longer lived. Importantly, Rampino and Self 1992 (*Nature*) stated that 'Volcanic aerosols may also contribute a negative feedback during glacial terminations, contributing to brief episodes of cooling and glacial readvance such as the Younger Dryas Interval.' Our research confirms and builds on this earlier work, and identifies a volcanic eruption potentially responsible for the YD. We have included the following text:
*"This perspective is consistent with previous observations, including those of Zielinski et al. (1996) who noted that when the climate system is in a state of flux it is more sensitive to external forcing, and that any post-volcanic cooling would be longer lived. Importantly, Rampino and Self (1992) stated 'Volcanic aerosols may also*

*contribute a negative feedback during glacial terminations, contributing to brief episodes of cooling and glacial*
*readvance such as the Younger Dryas Interval.'. Our results are entirely consistent with this perspective, and here*
*we identify the volcanic eruption responsible for the YD."*

**Comment #4:** In fact, in Fig. 4 there are two larger eruptions during the same period. Why did only Laacher See
produce cooling? They claim in the Fig. 4 caption that the Hekla eruption was more proximal, and therefore
should be discounted, but the way it works is that Icelandic eruptions into the westerlies have to go around the
world before the acid snow deposits on Greenland, and so there is no reason to think that it would have a
smaller climate impact than Laacher See.

**Response #4:** The volcanological information regarding the size of the Hekla eruption is from published sources
(Muschiatello et al., 2017, Nature Communications; Mortensen et al 2005, Journal of Quaternary Science), so
we are simply referring to previously published research when we refer to the fact that Hekla was substantially
smaller than the Laacher See eruption, and that it appears in the GISP2 ice core. Additionally, the reviewer is
incorrect about Icelandic eruptions, and it is well-established that sulphate from even small Icelandic eruptions
appears in Greenland ice cores (Muschiatello et al., 2017, Nature Communications; Abbott and Davies, 2012,
Earth Science Reviews; Abbott et al., 2012, QSR).  Muschiatello et al. (2017) state: "Icelandic volcanoes remain
the dominant source of volcanogenic aerosols in Greenland ice cores due to their relative proximity and high
eruptive frequency". The other eruption we mention in the Figure 4 caption is the Nevado de Toluca eruption;
we refer the reviewer to the text already in our manuscript that explains why its climate expression may not be
as clear:

*'The eruption was approximately the same size as the LSE, so the lack of climate cooling may reflect a different*
*climate response due to the eruption's latitude, which caused a more even distribution of aerosols across both*
*hemispheres, or a lower sulphur load. The 12.6 ka BP sulphate spike is associated with a short but dramatic*
*cooling; therefore the lack of long-term cooling may simply reflect the fact that temperatures had already*
*reached the lowest values possible under the insolation and carbon dioxide baseline conditions characteristic of*
*that time.'*

**Comment #5:** Even the size of the eruption is speculation, and the authors mix mass of SO2 with that of
elemental Sulphur with that of stratospheric aerosol. What do they claim actually was the stratospheric loading
for this eruption? And each time you talk about mass, please convert it to the same chemical so it can be
compared.

**Response #5:** We thank the reviewer for picking up on this. This partially stems from an ambiguity in a previously published paper that we used as a reference. We have now converted everything to the same chemical, except where the context requires us to discuss a different one.

**Comment #6:** The title is confusing. Why is it "re-evaluating?" There is no initial evaluation that is addressed in the abstract or in the paper.

**Response #6:** This issue was also brought up by another comment by another reader of the manuscript. The
Laacher See eruption was mentioned briefly (just a couple of sentences) in the late 1980s as a trigger for the YDE, before it was discarded due to evidence incorrectly showing that it occurred too early. However, the most recent lake core and ice core data indicate that the beginning of YD cooling (the start of GS-1) occurred synchronously with the LSE, so we feel that 'Re-evaluating' is the correct word to use here. We stated in the introduction of our originally submitted manuscript:

*"Instead, we re-introduce and provide new support for the hypothesis that the YD was triggered by the ~12.9 ka BP eruption of the Laacher See volcano, located in the East Eifel Volcanic Field (Germany). Early research considered the eruption as a possible causative mechanism for the YD (Berger, 1990). However, the concept was dismissed because lacustrine evidence across central Europe appeared to indicate that the YD's clearest*
*expression appeared ~200 years after the Laacher See Tephra within the same sediments (e.g., Brauer et al., 2008; Brauer et al., 1999; Hajdas et al., 1995)."*

We were not the first to suggest the eruption as a trigger, although we are 're-evaluating' the eruption's climatological consequences in a modern context and presenting far and away the most developed hypothesis
implicating the eruption as the YD trigger. Still, because two reviewers raise this same issue, we have now included the following text to better explain why we chose to use this term in the title:

*"Here we summarise but do not argue extensively for or against any of these established hypotheses. Instead, we investigate the hypothesis that the YD was triggered by the ~12.9 ka BP eruption of the Laacher See volcano,*
*located in the East Eifel Volcanic Field (Germany). Earlier research briefly alluded to the eruption as a possible causative mechanism for the YD (e.g., Berger, 1990; Bogaard et al., 1989). However, because the meltwater pulse hypothesis was already popularised, and because the effects of volcanic eruptions on climate would escape detailed quantification until after the 1991 Pinatubo eruption, the ~12.9 ka BP Laacher See eruption as a YD trigger never gained traction. Importantly, the concept was effectively dismissed after lacustrine evidence across*
*central Europe appeared to indicate that the YD's clearest expression appeared ~200 years after the Laacher See Tephra within the same sediments (e.g., Brauer et al., 2008; Brauer et al., 1999a; Hajdas et al., 1995). For example, Schmincke et al. (1999) state that "The Younger Dryas cooling period clearly was not triggered by LSE as formerly thought because it started ca. 180 years after the eruption", reflecting the accepted sequence of*

*events at that time. However, the identification of the Vedde Ash chronostratigraphic unit within Meerfelder*
*Maar (Germany) lake sediments has permitted improved correlation with Greenland ice core records, which also*
*contain the same ash (Lane et al., 2013a). This revised chronological framework now strongly suggests that the*
*12.880 ka BP Laacher See eruption was in fact synchronous with cooling associated with the YD onset (i.e., the*
*most recent abrupt Greenland millennial-scale cooling event, Greenland Stadial-1; 'GS-1'), but preceded major*
*atmospheric circulation shifts over central Europe (Rach et al., 2014).*

*Furthermore, we utilise ion data from the Greenland ice core GISP2 (Zielinski et al., 1997) on the most recent*
*chronological model for the core (the GICC05modelext chronology (Seierstad et al., 2014)) to identify a large*
*volcanogenic sulphate spike whose timing coincides with both the Laacher See eruption and the initiation of GS-*
*1 related cooling. We argue that the initial, short-lived volcanogenic aerosol cooling triggered a sea-ice/AMOC*
*positive feedback that caused both basin-wide cooling and the dynamical climate shifts most closely associated*
*with the YD. Therefore we are 're-evaluating' the Laacher See eruption's role in triggering the YD, building on*
*research that initially briefly mentioned the eruption as a causative mechanism for the YD, and later research*
*that dismissed the original version of the concept."*

**Comment #7:** Since the Laacher See eruption was high latitude, we would expect that for the same
stratospheric loading, it would have much less of a climate impact than an equivalent tropical eruption, since
the atmospheric lifetime would be much shorter, and there is less insolation at high latitudes. When you
compare to Toba, this must be addressed.

**Response #7:** We have added the following text to discuss this:

*"The residence time of aerosols within the atmosphere is not critical within the context of this model provided*
*the positive feedback is activated, and a sufficiently high aerosol-related cooling over only one summer and one*
*hemisphere could suffice. The strength of the feedback may also depend on the amount of hemispheric*
*temperature asymmetry caused by the eruption. Consequently, the high latitude LSE may actually have induced*
*a stronger hemispheric temperature asymmetry than the low latitude Toba eruption, although the Toba*
*eruption would have resulted in considerably more overall cooling. The long-term (e.g., hundreds to thousands*
*of years) climate response will depend on the climate background conditions; if a NH eruption occurs during an*
*orbitally-induced cooling trend (as may have been the case for Toba), the eruption will catalyse cooling towards*
*the insolation-mediated baseline."*

**Comment #8:** And if the eruption was in the fall or winter, most of the aerosol would have fallen out of the
stratosphere before the Sun comes up the next summer and there would be minimal impact on climate.

**Response #8:** For historical eruptions similar in size to the Laacher See eruption, aerosols have remained in the atmosphere for much longer than one year. For example, aerosols remained in the atmosphere for ~3 years after the Pinatubo eruption (15°N, 120°E) (Diallo et al., 2017), which is estimated to have released considerably less sulphur than the LSE (Sheng et al., 2015; Baales et al., 2002). Aerosols remained in the atmosphere for more than one year even after the 1980 Mount St. Helen's eruption (46°N, 122°W) (Pitari et al., 2016), which injected an order of magnitude less sulphur into the atmosphere than the LSE (Pitari et al., 2016; Baales et al., 2002) and erupted laterally (Eychenne et al., 2015, JGR-Solid Earth). Furthermore the eruption almost certainly occurred in the late spring. We have added the following text to discuss this in the manuscript:

*"Initiation of the positive feedback requires volcanic aerosols to remain in the atmosphere for at least one summer season. Evidence based on the seasonal development of vegetation covered by the LST suggests that the LSE occurred during late spring or early summer (Schmincke et al., 1999), and varve studies similarly suggest a late spring/early summer eruption (Merkt and Muller, 1999). Available evidence therefore suggests that the eruption occurred just prior to maximum summer insolation values, maximising the potential scattering effects*

*of the volcanogenic sulphate aerosols. Even if it were a winter eruption, for historical eruptions similar in size to the Laacher See eruption, aerosols have remained in the atmosphere far longer than one year, regardless of the eruption's latitude. For example, aerosols remained in the atmosphere for ~three years after the Pinatubo eruption (15°N, 120°E) (Diallo et al., 2017), which is estimated to have released approximately nine times less $SO_2$ than the LSE (~15-20 Mt versus 150 Mt) (Sheng et al., 2015; Baales et al., 2002). Aerosols remained in the*

*atmosphere for approximately three years even after the 1980 Mount St. Helen's eruption (46°N, 122°W) (Pitari et al., 2016), which injected almost 100x less $SO_2$ into the atmosphere than the LSE (2.1 Mt versus 150 Mt) (Baales et al., 2002; Pitari et al., 2016) and erupted laterally (Eychenne et al., 2015). In short, the LSE eruption probably occurred during the late spring or early summer, but even if the eruption were a winter eruption, the LSE's aerosols would have certainly persisted over at least the following summer, with the potential to catalyse*

*the positive feedback we invoke."*

**Comment #9:** The paper is replete with undefined acronyms, making it very confusing. All acronyms have to be defined the first time they are used. For example, what is LST? It is never defined. Is it LSE and a typo? What are

TOMS, NGRIP, GISP2, GICC05modelext, ITCZ, GS-20, GI-19 (in Fig. 6), ...? Please keep in mind that there will be readers not from your specific discipline, and so jargon needs to be defined. GS-1 is finally defined long after it is used, but the authors still never say what Greenland Stadial 1 is. What is a stadial? Why does Greenland have one? How many does it have?

**Response #9:** The reviewer is quite correct here, and we apologise for not defining these terms. We have gone through the manuscript and defined all acronyms and terms which might be confusing for a non-specialist. We have now also mentioned the total number of stadials as well at a relevant point in the manuscript.

**Comment #10:** The paper talks about magnitudes for volcanic eruptions, but never says what the scale is. Magnitude of what? If not of sulphur injection, then what is the point? And where do the data come from? There are no references to that.

**Response #10:** 'Magnitude' is a common term in volcanology that refers to the amount of tephra and lava erupted. It is very difficult to know for sure how much sulphur was in eruptions recorded in the geological past due to the fact that much of the sulphur existed in a volatile phase and is not preserved in the rock record. In general magnitude and sulphur concentrations are well correlated (Oppenheimer et al., 2003), and therefore magnitude is therefore often used as a surrogate for sulphur concentrations. All the available evidence suggests that if the LSE deviates from this trend, it is anomalously enriched in sulphur than expected. We have added the following text to the manuscript:

*"Both Pinatubo and the LSE were Magnitude 6 (M6) eruptions, where 'magnitude' is a measure of eruption size referring to the amount of material erupted (Deligne et al., 2010) on a logarithmic scale. However, the cooling effects of a volcanic eruption are controlled by the amount of sulphur released, and not necessarily the eruption size (Rampino and Self, 1982). In general, magnitude and erupted sulphur amounts are well correlated (Oppenheimer, 2003), and therefore magnitude is often used as a surrogate for sulphur yield. All the available evidence suggests that if the LSE deviates from this trend, it was anomalously enriched in sulphur relative to its magnitude (Baales et al., 2002; Scaillet et al., 2004), and that it therefore should have produced significant NH cooling."*

**Comment #11:** As for the Toba eruption, the paper is missing key references on the climate impact. Robock et al. (2009) found a larger short-term impact, but no long-term effect. Timmreck et al. (2010) claim that it would have had a small impact, as the particles would have grown and had a smaller impact per unit mass.

Robock, A., C. M. Ammann, L. Oman, D. Shindell, S. Levis, and G. Stenchikov, 2009: Did the Toba volcanic eruption of _74 ka B.P. produce widespread glaciation? J. Geophys. Res., 114, D10107, doi:10.1029/2008JD011652. Timmreck, C., et al., 2010: Aerosol size confines climate response to volcanic supereruptions. Geophys. Res. Lett., 37, L24705,  doi:10.1029/2010GL045464.

**Response #11:** We thank the reviewer for these suggestions, and we have now included these references as well as an enhanced discussion.

**Comment #12:** In any case, I find the Haslam and Petraglia (2010) Figure 1 very convincing that it got cold before the eruption. By the way, that reference is missing from the reference list. Why does the timing of the Toba eruption in Fig. 6 here differ from that in Fig. 1 of Haslam and Petraglia (2010)? Which is correct, and why?

**Response #12:** As we note in our previous submission's text, the timing of the Toba eruption S spike within the ice cores is based on Svensson et al., 2013, Climate of the Past, which uses a very thorough analysis of both Arctic and Antarctic ice core records. This represents the most up-to-date assessment of the timing of the Toba eruption relative to Greenland climate change, so we use this, in agreement with other recent publications (e.g., Polyak et al, 2017, Geology). The Haslam and Petraglia 2010 paper precedes the Svensson et al. 2013 analysis, and uses an older chronology. So our assessment is correct, and Haslam and Petraglia 2010 is incorrect, at least in terms of reflecting the most recently accepted chronology. Because the Haslam and Petraglia 2010 paper is demonstrably out-of-date (through no fault of their own – the paper preceded the chronological revisions we used), it is not worth discussing at length in this manuscript.

**Comment #13:** The paper ignores all the work that has shown that the 1257 Samalas eruption caused the Little Ice Age (Zhong et al., 2011; Miller et al., 2012; Slawinska and Robock, 2017). What does this tell us about the claim that a much smaller eruption of Laacher See caused a much larger climate response?

Miller, G. H., Á. Geirsdóttir, Y. Zhong, D. J. Larsen, B. L. Otto-Bliesner, M. M. Holland, D. A. Bailey, K. A. Refsnider, S. J. Lehman, J. R. Southon, Ch. Anderson, H. Björnsson, and T. Thordarson, 2012: Abrupt onset of the Little Ice Age triggered by volcanism and sustained by sea-ice/ocean feedbacks. Geophys. Res. Lett., 39, L02708, doi:10.1029/2011GL050168. Slawinska, J., and A. Robock, 2017: Impact of volcanic eruptions on decadal to centennial fluctuations of Arctic sea ice extent during the last millennium and on initiation of the Little Ice Age. J. Climate, doi:10.1175/JCLI-D-16-0498. http://journals.ametsoc.org/doi/abs/10.1175/JCLI-D-16-0498.1

Zhong, Y., G. H. Miller, B. L. Otto-Bliesner, M. M. Holland, D. A. Bailey, D. P. Schneider, and A. Geirsdottir, 2011: Centennial-scale climate change from decadally-paced explosive volcanism: a coupled sea ice-ocean mechanism. Clim. Dyn., 37, 2373-2387.

**Response #13:** We thank the reviewer for flagging this up; we have now included these suggested references, as well as some others as well. As for the second part of the comment, we refer the reviewer to our response to their Comment #7 (i.e., the size of the feedback is dependent on the background conditions, not the size of the eruption, provided the eruption is sufficiently large to trigger ice growth and interrupt ocean circulation). We have included a substantial amount of new text in the revised submission:

*'A similar mechanism may have also contributed to Little Ice Age cooling (Miller et al., 2012), with recent*
*research suggesting that a coupled sea ice/AMOC mechanism could extend the cooling effects of volcanic*
*aerosols by over 100 years during the Little Ice Age (Zhong et al., 2011). This perspective is supported by*
*modelling results suggesting that a large volcanic forcing is required to explain Little Ice Age cooling (Slawinska*
*and Robock, 2017). Lehner et al. (2013) identify a sea ice/AMOC/atmospheric feedback that amplified an initial*
*negative radiative forcing to produce the temperature pattern characterising the Little Ice Age. Similarly, large*
*volcanic eruptions in 536, 540, and 547 AD are hypothesised to have triggered a coupled sea ice/AMOC feedback*
*that led to an extended cold period (Buntgen et al., 2016). Recent research also highlights the possibility that*
*volcanism followed by a coupled sea ice/ocean circulation positive feedback triggered hemispheric-wide*
*centennial to millennial-scale variability during the Holocene (Kobashi et al., 2017). If a sea ice/AMOC feedback*
*was active following volcanic eruptions during the 6th Century, the Little Ice Age, and the Holocene, the*
*intermediate ice volume and transitional climate characteristic of the last deglaciation should have amplified*
*their effects. This perspective is consistent with previous observations, including those of Zielinski et al. (1996)*
*who noted that when the climate system is in a state of flux it is more sensitive to external forcing, and that any*
*post-volcanic cooling would be longer lived. Importantly, Rampino and Self (1992) stated "Volcanic aerosols may*
*also contribute a negative feedback during glacial terminations, contributing to brief episodes of cooling and*
*glacial readvance such as the Younger Dryas Interval". Our results are entirely consistent with this perspective,*
*and here we highlight a candidate volcanic eruption whose timing coincided with the onset of YD-related*
*cooling. However, despite increasingly tangible evidence that eruptions can affect AMOC strength and sea ice*
*extent, the exact nature of any positive feedback is still unclear. Future research should prioritize the*
*identification and characterisation of this elusive, but potentially commonplace, feedback that amplifies*
*otherwise subtle NH temperature shifts.'*

**Comment #14:** It would have been nice to have used hanging indents or additional spacing for the reference list to make it easier for the reader to find each paper in the list. In addition, there are another 35
comments in the attached annotated manuscript that need to be addressed.

**Response #14:** We agree with the reviewer, and have added a space between the references for increased legibility. We have addressed all of the comments contained in the annotated manuscript.

**Comment #1:** The manuscript "Re-evaluating the link between the Laacher See volcanic eruption and the Younger Dryas" by Baldini et al proposed Laacher See Eruption (LSE) as a potential trigger of the YD cold interval during the last deglaciation. The manuscript is good written and easy to follow.

**Response #1:** Thank you for these positive comments, which are much appreciated.

**Comment #2:** The authors argued that radiative effect of the LSE could lead to a cooling over the Northern Hemisphere, which eventually triggered the YD due to the existence of "sweet spot" of millennial-scale variability during glacial periods and positive feedbacks. I do see the potential of this mechanism, which enables an improvement of our understanding of YD dynamics.

**Response #2:** Again, we thank the reviewer for these positive comments and for summarising well what is
essentially a very simple idea. We agree that there is clear potential for the LSE to have triggered the YD, and we very strongly feel that the hypothesis merits further consideration.

**Comment #3:** However, I'm a bit suspicious of its reliability. Some points are summarized in the following: 1) Responses of ocean circulation (AMOC) to a Northern Hemisphere volcano eruption is not that supportive of
authors' argument. According to Pausata et al 2015 (PNAS), eruption's effect on AMOC is positive (strengthening) rather negative (weakening) at the first 20 years after the eruption, contrast to the weakening AMOC during YD.

**Response #3:** As we noted in our preliminary response online, the reviewer is correct that Pausata et al (2015) found that an eruption would strengthen AMOC. Other modelling studies based on historical data also suggest
that eruptions may strengthen AMOC (Ottera et al., 2010; Swingedouw et al., 2014; Ding et al., 2014). However, other models suggest that AMOC may intensify initially, but then weaken after about a decade (Mignot et al., 2011). A modelling study by Schleussner and Feulner (2013) suggested that volcanic eruptions occurring during the last millennium triggered increased Nordic Sea sea ice extent which weakened AMOC and eventually cooled the entire North Atlantic Basin. Other research finds that North Atlantic sea ice growth following a negative
forcing weakened oceanic convection and northward heat export during the Little Ice Age (Lehner et al., 2013). These are all studies focussing on eruptions that occurred over the last 1000 years, and they still yielded contradictory results. Therefore, we feel that how an eruption might affect AMOC at ~12.9 ka BP is still essentially unknown.

We thank the reviewer for raising this point, and we have now included the following text in an enhanced
discussion to address it:

*'Although the radiative effects associated with volcanic aerosols are reasonably well understood, the systematics of how volcanic eruptions affect atmospheric and oceanic circulation are less well constrained. Research suggests that volcanic eruptions affect a wide variety of atmospheric phenomenon, but the exact nature of these links remains unclear. For example, Pausata et al. (2015) used a climate model to conclude that high latitude NH eruptions trigger an El Niño event within 8-9 months by inducing a hemispheric temperature asymmetry leading to southward Intertropical Convergence Zone (ITCZ) migration and a restructuring of equatorial winds. The model also suggests that these eruptions could lead to AMOC shifts after several decades, consisting of an initial 25-year strengthening followed by a 35-year weakening, illustrating the potential for climate effects extending well beyond sulphate aerosol atmospheric residence times. Several modelling studies based on historical data suggest that eruptions may strengthen AMOC (Ottera et al., 2010; Swingedouw et al., 2014; Ding et al., 2014) but also increase North Atlantic sea ice extent for decades to centuries following the eruption due to the albedo feedback and reductions in surface heat loss (Ding et al., 2014; Swingedouw et al., 2014). Other models suggest that AMOC may intensify initially, but then weaken after about a decade (Mignot et al., 2011). A modelling study by Schleussner and Feulner (2013) suggested that volcanic eruptions occurring during the last millennium triggered increased Nordic Sea sea ice extent which weakened AMOC and eventually cooled the entire North Atlantic Basin. Importantly, Schleussner and Feulner (2013) concluded that short-lived volcanic aerosol forcings triggered "a cascade of sea ice-ocean feedbacks in the North Atlantic, ultimately leading to a persistent regime shift in the ocean circulation". Other research finds that North Atlantic sea ice growth following a negative forcing weakened oceanic convection and northward heat export during the Little Ice Age (Lehner et al., 2013). Quantifying the long-term influences of single volcanic eruptions is confounded by the effects of subsequent eruptions and other factors (e.g., solar variability, El Niño events), which can overprint more subtle feedbacks. For example, model results looking at recent eruptions found evidence that different types of eruptions can either constructively or destructively interfere with AMOC strength (Swingedouw et al., 2014). Therefore, despite increasingly clear indications that volcanic eruptions have considerable long-term consequences for atmospheric and oceanic circulation, the full scale of these shifts is currently not well understood even over the last two millennia, and are essentially unknown under Glacial boundary conditions.'*

**Comment #4:** 2) Effect of southward ITCZ shift will lead to an increase of salinity in the North Atlantic subtropics, which will also act as a negative feedback to a potential weakening AMOC (Schmidt et al 2006 Nature).

**Response #4:** We have now included the reference suggested along with a new paragraph of text:

*'MWP-1B may have cooled the SH and strengthened AMOC, prompting northward migration of the ITCZ and NH mid-latitude westerlies to achieve equilibrium with high insolation conditions, thereby rapidly reducing sea ice*

*extent and warming Greenland, but this requires further research, particularly because the source, duration, and timing of MWP-1B are still unclear. Reduced oceanic salt export within the North Atlantic subtropical gyre, as is characteristic of stadials, may have preconditioned the North Atlantic toward vigorous AMOC following the initial migration of atmospheric circulation back to the north (Schmidt et al., 2006).'*

**Comment #5:** 3) Although the ice volume during YD is beneficial to the occurrence of millennial-scale variability
(Zhang et al 2014 Nature), the high CO2 level (250 ppm) will shift the "sweet spot" to a lower level of global ice volume (Zhang et al 2017 Nature Geo). This will weaken the arguments proposed by the authors.

**Response #5:** We thank the reviewer for raising this potential complicating factor, and we now discuss this point:

*'The apparent high sensitivity of the climate system to millennial-scale climate change during times of*
*intermediate ice volume is well documented (e.g., Zhang et al., 2014; Zhang et al., 2017), and here we investigate this further by examining the timing of Greenland Stadials relative to ice volume (estimated using Red Sea sea level (Siddall et al., 2003)). The timings of 55 stadial initiations as compiled in the INTIMATE (INTegration of Ice core, MArine, and TErrestrial) initiative (Rasmussen et al., 2014) are compared relative to ice volume, and indeed a strong bias towards intermediate ice volume conditions exists, with 73% of the millennial*
*scale cooling events occurring during only 40% of the range of sea level across the interval from 0-120 ka BP (Figure 6). The distribution of events suggests that the most sensitive conditions are linked to ice volume associated with a sea level of -68.30 m below modern sea level. This intermediate ice volume was commonplace from 35-60 ka BP, and particularly from 50-60 ka BP. However, over the interval from 0-35 ka BP, these ideal intermediate ice volume conditions only existed during a short interval from 11.8-13.7 ka BP, and optimal*
*conditions were centred at 13.0 ka BP (Figure 6). These results are broadly consistent with previous research (e.g., Zhang et al., 2014; Zhang et al., 2017), but the timing and duration of the most sensitive interval of time, and the likelihood that a forcing produces a longer-term cooling event, may ultimately depend on a more complex interplay between ice volume and atmospheric CO2 (Zhang et al., 2017). Atmospheric pCO2 during the YD initiation was relatively high (~240 ppmv) and could therefore affect the timing of ideal conditions for abrupt*
*climate change in conjunction with ice volume, but their precise interdependence is still unclear. Finally, a frequency distribution of the sea level change rate associated with each stadial indicates that whatever mechanism is responsible for triggering a stadial operates irrespective of whether sea level (i.e., ice volume) is increasing or decreasing (Figure 6b). Because active ice sheet growth should discourage meltwater pulses, this observation seemingly argues against meltwater pulses as the sole trigger for initiating stadials.'*

**Comment #6:** Nevertheless, I do see a potential of LSE (or northern hemisphere volcanic eruption) as a trigger to YD

**Response #6:** We agree, and thank the reviewer for this supportive comment. We see no reason why a well-dated, very high-sulphur, and high-latitude eruption should not at least be considered as a trigger for a cold event.

**Comment #7:** Muschitiello et al (2017 Nature Comm, also cited by the authors) recently proposed that the volcanic eruptions can effectively influence the mass balance of ice sheet via altering its surface albedo. This will promote the ice-sheet melting, leading to freshwater input to the North Atlantic and weakening the AMOC. I'm not an expert on data and climate response to the volcanic eruptions.

**Response #7:** This is a good point, but our understanding of the mechanisms invoked by that paper are that the eruptions have to be fairly proximal to the ice sheet to cause melting, i.e., long-lasting Icelandic eruptions affected the Fennoscandian Ice Sheet. It is therefore unlikely that the LSE affected the albedo of any major ice sheet.

**Comment #8:** But I think if the author can well improve the robustness of their arguments (probably by rephrasing the mechanisms), this will be a nice manuscript for Clim Past.

**Response #8:** We have now rephrased and extended our text considerably, and we think that the arguments are indeed much stronger. We also include a substantially revised section on relevant published climate modelling results that also discuss possible mechanisms. We thank the reviewer for highlighting that this rephrasing might be necessary, and we believe that this has indeed helped.

**Comment #9:** Line 145: Citation "Pasauta et al 2015 Tellus B" is not proper here. It should be Pausata et al 2015 PNAS.

**Response #9:** This has been corrected.

**Comment #1:** The subject of the Younger Dryas cooling is one of considerable interest and fascination in the scientific community. Here, most research has been dominated by one theme that the cooling was triggered by a freshwater flood, or rerouting of meltwater, to the North Atlantic ocean. The idea that the YD cooling might have been triggered by a volcanic eruption has received much less attention and is very interesting.

**Response #1:** As the reviewer notes, there has been essentially no modern research on whether or not the Laacher See eruption could have triggered the YD, largely because the evidence available suggested that the LSE predated the onset of YD cooling (GS-1) by ~200 years. Given that the last few published review papers on the YD do not even mention volcanism, we feel that this is a valuable (and novel) contribution, and we are glad that the reviewer finds it interesting.

**Comment #2:** Overall, I really enjoyed the paper. It's very well written, easy to follow, and provides a nice break from the more typical meltwater-trigger hypothesis. Indeed, I found the discussion about the sensitive of climate to intermediate ice volume conditions, and the alignment of this 'ideal' configuration, to the timing of the YD very enlightening.

**Response #2:** We thank the reviewer for these supportive comments.

**Comment #3:** But whether a volcano actually triggered the YD is hard to tell from this paper. Yes, there was an eruption around the time of the YD cooling, but did it really produce a 1000-yr cooling? As such, the manuscript would have been vastly improved if the authors had done their own climate modeling. I think it would have been fantastic to try and see whether a volcano could have triggered a YD-like cooling. Indeed, the authors note that previous studies (fig 2) released 10-time LESS SO2 to the atmosphere than what is estimated here. Whether these experiments should be undertaken, I will leave that up to the authors, but I'm not going to rejecting this paper simply because they were not carried out.

**Response #3:** We really appreciate the reviewer's perspective on the inclusion (or not) of modelling. Although we agree that modelling is important, we feel that this is outside the scope of the current submission, and best left for future research. The reason is outlined at length in our response to Reviewer #2's comment #2, as well as in new text that we have added to discuss relevant models. Essentially, model simulations of the response of AMOC to volcanic forcing over the last 1000 years have yielded ambiguous results, with some models predicting that eruptions strengthen AMOC, and others predicting a weaker AMOC. Still others suggest initial strengthening, followed by long-term weakening. We strongly feel that under the considerably less-well constrained deglacial conditions, modelling results would be even more ambiguous, and results would not necessarily be robust. We hope that this manuscript would provide the motivation for substantial future
modelling work on the triggering of the YD by the LSE, and we feel that modelling support (or not) would need
to come from multiple climate studies conducted over several years.

**Comment #4:** Finally, I wasn't sure if the MWP-1b discussion was really needed. The existence of this period of
rapid sea level rise is still very much debated, as is its source, with various camps arguing back-and-forth over an
Antarctic or Laurentide contribution.

**Response #4:** We have toned down the MWP-1b discussion, and included more text, to emphasise that the
provenance of the meltwater, and even its existence, is still uncertain and controversial.

**Comment #5:** Anyway, my overall opinion is that this is a very interesting paper and it should be published with
minor corrections/edits.

**Response #5:** We thank the reviewer for this opinion, and for their comments. We have taken on board their
suggestions, as well as those from the other reviewers, and we feel that our revised manuscript is substantially
improved from the last submission.

**Responses to Short Comment by David Pyle**

**Comment #1:** For me the leading questions (which come out of this paper) are:
- was the LST eruption the source of a sulphur 'spike' in Greenland ice cores; and how
could we test this assertion?

**Response #1:** This could be tested by searching for, and fingerprinting, ash in the ice containing the sulphur spike. We now also mention using triple sulphur isotopes to detect stratospheric aerosols as future work. Although this would not pinpoint the LSE as the cause, it would suggest that it was a large eruption where aerosols reached the stratosphere, reducing the likelihood of a localised Icelandic eruption being the origin of the sulphur spike. We now state this as critical future work.

**Comment #2:** What was the total volatile yield from the LST eruption; and what new measurements are needed to improve on this assessment. (And did the halogen release have any impact?)

**Response #2:** The magma that fed the LSE was undoubtedly sulphur-rich (see Textor et al., 2003; and Table 5 in Scaillet et al. 2003), and would have injected significant amounts of $H_2S$ and/or $SO_2$ into the atmosphere, enabling it to have an effect on climate comparable to eruptions that are substantially larger in total erupted volume. Estimating volatile yields from ancient eruptions is difficult and complex (see the review in Scaillet et al., 2003), but we have updated this section (Section 2.0, Background) to include work that builds on the work of Schmincke et al., (1999) that was previously cited. A full petrological study of sulphur in the magma, using updated methodology (for example, that of Vidal et al., (2016)) is beyond the scope of this paper, but we hope that our work will stimulate others to undertake this important study. We have now included the most recent estimates of total sulphur produced by the eruption, as reported in the suggested papers. We have also calculated a new estimate based on the difference between the petrologic and actual sulphate yields of 17 explosive eruptions (mostly from Shinohara 2008, *Reviews in Geophysics*), which is consistent with the range reported by Textor et al., 2003. We also now mention that halogen release could also play a role.

**Comment #3:** What cascade of physical processes could lead to the observed pattern of response seen for the onset of the YD; and is a volcanic eruption a sufficient driver, on its own.

**Response #3:** The mechanisms that lead from the initial hemispheric cooling from the LSE sulphate cloud (itself uncontroversial) through to the YD as experienced across western Europe, is described in Section 3.3. Crucially, the LSE occurred during a period of intermediate ice cover, when we infer that the climate was particularly sensitive to short-term cooling events (as has been concluded previously). Thus, we consider that the sulphurrich LSE was capable of acting as the catalyst for longer term cooling and climatic reorganisation characteristic of the YD. We have also included a much more detailed discussion regarding the positive feedback, and outlining in detail the substantial amount of recent research that has similarly concluded that volcanic eruptions can have long-term consequences for climate.

**Detailed points**

Line 44 – there is also a documented enrichment in noble metals (e.g. platinum), at this stratigraphic level both in North America and Europe (Moore et al., Scientific Reports 7 Article Number: 44031, 2017; and papers by A Andronikov).

**Response:**  We have now included references to the paper by Moore et al. and Andronikov. We note that in the

Supplemental Material of Moore, the authors state that the LSE eruption occurred 200 years before the YD – which appears to reflect the consensus perspective, which unfortunately is also incorrect, as we discuss in the manuscript. In a more recent publication (Wolbach et al., 2018, J. of Geology), the eruption is also dismissed as being the source of the Pt spike, with the authors stating that the eruptions occurred 200 years too early. In their Figure 7 (and elsewhere), the timing of the spike is nearly indistinguishable from the timing of the eruption. We feel that there is a very reasonable possibility that the Pt spike is not derived from a cometary airburst, but instead is derived from the LSE. We now include text alluding to this possibility, but would like to reserve an in-depth discussion for after we are able to sample the LST for Pt concentrations directly.

Line 50 – 'new support' yes, but not much new evidence.

**Response:**  We explain how short-term, volcanogenic sulphate-induced hemispheric cooling could have acted as the trigger for the longer-term cooling and climatic changes associated with the YD. In the last few comprehensive reviews of the Younger Dryas event (e.g., Carlson 2010; Fiedel 2011) there was no mention of the LSE or any volcanic eruption as a trigger. We therefore feel that we use existing evidence to arrive at a very novel, provocative, and important conclusion. We also identify a candidate sulphur spike within the GISP2 ice core, and provide a new estimate for the total sulphate emitted by the eruption.

Line 50 – ages: some explanation is needed about the framing of time in the paper, as the model ages are derived from multiple approaches.

**Response:** We have now clarified the origin of the dates, as well as added dating uncertainties where needed.

Lines 53- 56: the emphasis here isn't quite right. The evidence for a 200-year time break between LST and the onset of 'YD' conditions in continental Europe remains firm. What has changed – since Lane et al., 2015, is the recognition that the onset of YD is time-transgressive. So – by inference – LST overlaps with GS-1 and the onset of YD as recorded in ice chemistry in Greenland.

**Response:** There is a lot of grey area here, but we believe that it is correct as we have it written. The YD is a term that has its origins in central European climate studies that detected the change in atmospheric circulation. So whereas GS-1 cooling did lead to the manifestation of the YD in central Europe, GS-1 cooling was not the YD itself.

Line 57 – does the 'GICC05modelext chronology' need a word or two of explanation
**Response:** We have now clarified this at the first occurrence in the text.

LST Impacts
Line 74: see also the extensive work by Felix Riede on the impacts of the LST:

Book, 2017 - Splendid Isolation : The eruption of the Laacher See volcano and southern Scandinavian Late
Glacial hunter-gatherers. / Riede, Felix. Aarhus : Aarhus Universitetsforlag, 2017. 214 p.

- Changes in mid- and far-field human landscape use following the Laacher See eruption (c. 13,000 years BP). / Riede, Felix. In: Quaternary International, Vol. 394, 02.2016, p. 37-50.

2012 - Bayesian radiocarbon models for the cultural transition during the Allerod in southern Scandinavia, Riede, Felix; Edinborough, Kevan, JOURNAL OF ARCHAEOLOGICAL SCIENCE 39, 744-756

- The Laacher See-eruption (12,920 BP) and material culture change at the end of the Allerod in northern Europe, Riede, Felix, JOURNAL OF ARCHAEOLOGICAL SCIENCE 35, 591-599

**Response:** We thank Professor Pyle for bringing these studies to our attention and we now reference Riede (2008) and Riede (2016) in the appropriate section.

Volcanic Emissions
Lines 78 - 85 – this section needs some critical revision and updating. Harms and Schmincke (2000) estimated, using mass balance, an SO2 yield of 20 Tg. Harms et al. (2004) did some experiments on LST magmas and determined the P, T , H2O conditions under which the magma was stored; you could revisit the calculations of

Harms & Schmincke to re-estimate the S and water budgets of the system – taking account of the work that Bruno Scaillet and colleagues have done on other systems. The '150 Mt' value should be cited as Schmincke et al (1999, Quaternary International, 61, 61-72) – it is, as the authors say 'highly speculative' and based on using the 'Pinatubo multiplier'; this can certainly be improved upon, rather than being taken as a starting point for the argument. Similar calculations have been attempted by Textor et al., 2003, (Geol Soc London Spec Pub, 'Volcanic Degassing', 213, 307-328); who also estimated the total halogen yield. Discussion on volcanic emissions Recent papers may also add a little to the discussion here: for example - - Colose, C.M., A.N. LeGrande, and M. Vuille, 2016: Hemispherically asymmetric volcanic forcing of tropical hydroclimate during the last millennium. Earth Syst. Dyn., 7, 681-696, doi:10.5194/esd-7-681-2016. - LeGrande, A.N., K. Tsigaridis, and S.E. Bauer, 2016: Role of atmospheric chemistry in the climate impacts of stratospheric volcanic injections. Nature Geosci., 9, no. 9, 652-655, doi:10.1038/ngeo2771.

**Response:** These are very good points and we have extensively updated this section. We now include the improved sulphur yield estimates of Textor et al., (2003), and note that these are still significantly greater than those emitted by the climatically-important Pinatubo eruption. A detailed petrological investigation of S contents in LS tephra would be a very interesting, substantial study, and we hope our research will encourage other researchers to look into this. As mentioned above, we have derived a new estimate for total sulphate released by using the mean of a range of petrologic to actual ratios reported in Shinohara 2008.

Line 178 – 'five years' may be an overestimate: in Graf and Timmreck's model, sulphate aerosol had an e-folding time of 11 months; and the detectable signal of volcanic stratospheric sulphate aerosol is usually considered to be less than three years.

**Response:** The text has been modified throughout.

Lines
– 195 – there's not really any new evidence here?

**Response:** Here we are drawing links between different a wide range of studies to establish a plausible mechanism to trigger long-term cooling following a short-term volcanic forcing of climate. We have added substantially more discussion to make this point clearer.

Line 208 – the magnitude of the eruption is not relevant, it's the magnitude of the gas release that is the key point. The LST magma is an unusual composition, so surely this is the starting point for why it may have had an exceptional impact?

**Response:** This is a very useful comment, and although we did appreciate that this was the case before, the manuscript did focus too much on magnitude and not enough on sulphur yield and the unusual composition of the magma. We have added a statement on the importance of the sulphur content of the erupting magma, as well as emphasising the sulphur yield rather than the magnitude throughout. We have also added a new figure, which clearly shows that the LSE was anomalous in terms of total sulphur emitted.

Lines 274 – 284: there still is no way of linking a sulphate peak in an ice core to a particular eruption, in the absence of any tephra so this remains speculative. It remains possible that sulphur mass-independent isotopic fractionation signals may help to identify plumes that entered the stratosphere (e.g. Martin et al., 2014, Volcanic sulfate aerosol formation in the troposphere, JOURNAL OF
GEOPHYSICAL RESEARCH-ATMOSPHERES Volume: 119 Issue: 22 Pages: 12660-12673), but this still won't help with source identification.

**Response:** This is an important point and we now discuss this in more depth, with the following text added to the conclusions:

*"Similarly, volcanic sulphate 'triple' isotope ratios of sulphur and oxygen provide information regarding the residence time of volcanic plumes in the stratosphere. The majority of atmospheric processes encourage mass dependent fractionation; however rare mass-independent fractionation processes produce isotope ratios that do not behave according to predictions based on mass dependent processes (Martin et al., 2014). Historical volcanic eruptions where sulphate aerosols reached the stratosphere have been successfully identified in ice cores (Baroni et al., 2008; Savarino et al., 2003), indicating that the technique is effective at distinguishing large explosive eruptions from smaller local ones. This technique could also determine if the sulphate in the potential LSE sulphate spike reached the stratosphere. Although this would not necessarily confirm the LSE as the source, it would strongly suggest that the sulphate was sourced from a climatologically important eruption rather than a smaller Icelandic one."*

**Responses to Short Comment by Evzen Stuchlik**

First, we would like to simply note that the goals of our submitted manuscript are straightforward: i) to highlight that the timing of the Laacher See eruption is indistinguishable from the initiation of cooling associated with the Younger Dryas, ii) to highlight the possibility that the effects of volcanic eruption can persist longer than just 1-3 years, and finally iii) that consequently the eruption should be viewed as a viable trigger for the Younger Dryas Event. In other words, if the LSE occurred at the correct time (and it appears that it did), and if an eruption of this scale and sulphur content could catalyse extended cooling (and it appears that it could), then logically the LSE should be considered a viable trigger for the Younger Dryas. Clearly more research needs to be conducted on this topic, but getting the idea out there is the key first step. We note that this is the only hypothesis where it is universally agreed that the proposed trigger actually occurred, and where it currently seems that the age coincides with the initiation of Younger Dryas cooling.

We have now added a section discussing the pros and cons of other proposed triggers. However, there are reams of papers discussing the pros and cons of the Younger Dryas Impact Hypothesis specifically, and providing a thorough review of all the evidence for or against this hypothesis is not possible or necessary. For example, it is true that the Laacher See Eruption would not account for the observed megafaunal extinctions across North America. However, recent papers [*Cooper et al.*, 2015; *Metcalf et al.*, 2016; *Rule et al.*, 2012; *van der Kaars et al.*, 2017] make an extremely strong case that this was caused by human migration (how else to explain the observation that the extinctions did not occur at the same time, and tracked human migration?), and that therefore the LSE (or an impact, or a meltwater pulse) would not have needed to cause any extinction. This perspective is also supported by the presence of other Younger Dryas-type events that apparently occurred during other Glacial terminations, e.g. TIII [*Broecker et al.*, 2010] but that were not associated with megafaunal extinctions, implying that neither YD-type climate change nor a bolide impact were the cause of the megafaunal extinctions. Furthermore, we do not argue that a bolide impact did not happen near the YD boundary (it may have), so defending the presence or absence of a Pt spike, shocked quartz, black carbon, nanodiamonds, etc. is well beyond the scope of the manuscript. That being said, we have added considerable more detail in the revised manuscript.

In response to your comments that are specific to our hypothesis:

1) Comment: Our 'statement indicating that Laacher see eruption (LSE) effect could last for some 5 years is in the contrast to surprisingly main conclusion not completely supported by own data and highly speculative that this event could trigger YD cooling.'

1) Response: We discuss a proposed ice/ocean feedback in detail in Sections 3.2, 3.3., and 3.4, and the concept of a positive feedback amplifying the original volcanic forcing is increasingly commonplace (see recent paper by Kobashi et al., Scientific Reports 2017 for example). There are now several papers that suggest the presence of a sea ice/oceanic circulation feedback that amplifies the initial short-lived aerosol cooling, and we will discuss these further in the revised manuscript as suggested by another reviewer.  We therefore feel that the concept of a longer-term volcanic forcing is well-defended already by several pages of text as well as previously published papers (these will be included and discussed in the revisions); we do not feel that it is highly speculative if you are familiar with this most recent literature. Upon any revision, we will revise this text to ensure that this message is clear, and describe the positive feedback mechanism in more detail.

2) Comment: 'Resulting the title of msc starting with "Reevaluation" is inappropriate to the msc content.'

2) Response: The Laacher See eruption was very briefly mentioned as a proposed trigger for the YDE, before it was discarded. However, the most recent lake core and ice core data suggest that the YD cooling occurred synchronously with the LSE, so we feel that 'Re-evaluating' is the correct word to use here. We were not the first to suggest the eruption as a trigger, although we are 're-evaluating' the eruption's climatological consequences in a modern context. Still, another reviewer raises this same issue, so although we feel that this is in fact the correct term, we clarify why we chose to use this term in the title.

Finally, we thank you for the papers that you have provided. We will include these in any revisions, where relevant and appropriate.

Both Pinatubo and the LSE were Magnitude 6 (M6) eruptions, where 'magnitude' is a measure of eruption size referring to the amount of tephra and lavamaterial erupted (Deligne et al., 2010) (Deligne et al., 2010 JGR) (on a logarithmic scale: ). However, the cooling effects of a volcanic eruption isare controlled by the amount of sulphursulphur released, and not necessarily the eruption size (Rampino and Self, 1982). Unfortunately, Tthe amount of sulphur released by prehistoric eruptions is often ambiguous due to the fact that much of the sulphur exists in a volatile phase and is not retained in the rock record. 
[revised manuscript text omitted]

---

## Referee Report (RR1)

First, I would like to thank Baldini and colleagues for their detailed point-by-point response to the reviewers' comments. I have now scrutinised their manuscript for the second time and generally I think the authors have gone a long way to address my concerns, primarily by rephrasing, clarifying and extending the main text. However, I'm still not convinced by the proposed mechanism, i.e. that a single high-latitude eruption caused a 1000-year long cold spell over the Northern Hemisphere.

Although the authors claim that "we are the first to detail the positive ice-AMOC feedback following the LSE within the context of the YDE", I'm sorry to say that I think this is not sufficiently supported by climate model simulations (e.g. see other reviewers' comments). The lengthy discussion on previously published model results is welcome but none of the simulations are appropriate to test whether the LS could have caused a YD-like event. On top of that, the published model results of the Common Era are discordant in the way they simulate climate and AMOC response to volcanic forcing and therefore cannot be used to rest the authors' case.

I would feel much more comfortable if the authors would back their claims with model experiments (as also requested by reviewer2), for instance by simulating a large volcanic eruption under deglacial (or glacial) boundary conditions. However, I understand this would take a considerable amount of extra time and the authors didn't envisage undertaking such task to begin with. Anyway, from a proxy point of view I am willing to encourage publication provided that the authors tone down their claims and clearly state in the abstract the speculative nature of the proposed mechanisms. As to the shortcomings with the modelling aspects of this study I will leave it to the other reviewers and the editor to decide whether this work is sufficiently suitable for publication even without support from ad-hoc climate model experiments.

---

## Author Response (AR2)

Department of Earth Sciences

Dr James U.L. Baldini
Associate Professor (Reader)
Department of Earth Sciences
University of Durham
Durham DH1 3LE, UK
+44 (0) 191 334 2334
james.baldini@durham.ac.uk

June 2018

Dear Dr Thornalley,

Thank you for your continued consideration of our manuscript entitled 'Evaluating the link between the sulphur-rich Laacher See volcanic eruption and the Younger Dryas climate anomaly' for publication in *Climate of the Past*. We have now addressed the minor comments made by the reviewers, as well as your editorial comments.

We greatly appreciate you taking the time to make in-depth comments, which have helped improve the manuscript and will undoubtedly increase the profile of any paper that may arise. Below this cover letter you will find our responses to the points raised. We have also made some unsolicited cosmetic changes that improve readability, and that generally improve the manuscript (please see the manuscript version with tracked changes, below as well).

Please do let us know if you require any further details on any of the changes made, if you would like to discuss any aspect further, or if you have any other questions regarding our revised resubmission.

Best regards,

Dr James Baldini
(Corresponding author)

**Responses to Reviewer and Editorial comments (responses are organised in the order presented in the editor's comments)**

We thank the reviewers for their additional comments that have continued to improve the manuscript. We would also like to thank the editor, whose in-depth and useful comments have significantly improved the manuscript. We have chosen to incorporate most of the recent set of comments into the text, but in some instances we have opted not to make these revisions. In these cases, we provide the reasons why we chose not to make the suggested revision, but we are happy to discuss these points further, particularly if we have misinterpreted any of the reviewer/editor comments.

**Comment 1:** Review 2 (Prof. David Pyle) provides some straight-forward additions. Please implement.

**Comment 1a:** The relationship between eruption size and sulphur release has been recently updated in a paper by Carn et al (Multi-decadal satellite measurements of global volcanic degassing, JVGR 311, 99 - 134, 2016 https://doi.org/10.1016/j.jvolgeores.2016.01.002) (their fig. 11). This supports the point made by the authors - broadly sulphur release scales with eruption size, but there is more than 2 orders of magnitude variation between high-S and low-S eruptions of a given size.

**Response**: We thank Prof. Pyle for suggesting this relevant new paper, and we have changed the text to include this reference. The new text reads:

"However, variability of almost three orders of magnitude exists in the amount of sulphur released amongst equivalently sized explosive eruptions (Carn et al., 2016), and consequently eruption size is not the only predictor of total sulphur released. In the case of the LSE, all the existing evidence suggests that it was anomalously enriched in sulphur relative to its magnitude (Baales et al., 2002; Scaillet et al., 2004), and that it therefore should have produced significant NH cooling."

**Comment 1b:** In the reference list the citation of Nowell et al. (2006) should be ' J. Quaternary Sci., Vol. 21 pp. 645–675'

**Response: Thank you.** This has been corrected.

**Comment 2:** Review 3 suggests that a condition for publication should be highlighting, in the abstract, that the volcanic trigger hypothesis needs testing and following up with rigorous modelling studies, similar to what was done for the meltwater hypothesis, since, in its current format, this idea is speculative. Review 1 makes a similar point, but also adds that the claims need to be toned down and the speculative nature made clear i.e. this idea is, as yet, conjecture. I think both these requests can be dealt with adequately by adding a couple of explicit sentences at the end of the abstract (and also the introduction) that clearly state the speculative/conjectural nature of this idea and the urgent need to test it with relevant, specific modelling studies.

**Response:** We have added the following text to the end of the abstract:

"However, we recommend that future studies prioritise climate modelling of high latitude volcanism during deglacial boundary conditions to test this hypothesis further."

We have also added the following text to the end of the introduction:

"Although more research is clearly needed to thoroughly investigate this hypothesis, the apparent coincidence of a large, very sulphur-rich eruption with the beginning of YD cooling is compelling, and well worth exploring further."

We still have a long discussion in the conclusions regarding future research, and the claims have been toned down throughout the text (please see tracked changes). We have avoided the use of the word 'speculative' in the abstract as we feel that, despite being under-researched, this hypothesis has just as much factual support as any of the others. Whereas the other hypotheses are far better researched, no universal agreement exists as to whether a meltwater pulse or a bolide impact even occurred, which is clearly not the case for the LSE. There is also very little doubt that the eruption would have caused substantial NH cooling for 1-3 years, or that the eruption coincided with the onset of YD cooling. The uncertainties with the hypothesis are mostly limited to the nature of the positive feedback, which is also the case for both of the other hypotheses that are normally not labelled by authors as 'speculative'. It is clear from our text where the uncertainties exist, and we are concerned that using the term 'speculative' in the abstract would discourage other researchers from following up our hypothesis, which we do feel has merit and is well supported. However, we have used this word in other places in the text where appropriate, and of course we are happy to discuss this further.

**Comment 3:** I would also add to this, that I strongly suggest towards the end of the introduction you mention something along the lines of: "A number of studies have suggested that the YD, like DO events, may be caused by internal oscillation of the climate system when it is an intermediate glacial state, such as midway through deglaciation. Yet, external forcing may also play an important role in controlling the precise timing of the transition into the YD and, by itself, external forcing has also been proposed as the underlying trigger for the YD. Here, we suggest that a volcanic eruption trigger (in this case the Laacher See eruption) should be considered as a viable hypothesis for triggering the onset of the YD." This would alert the reader to your later text, at the end of the discussion, where you discuss the possibility of the YD not requiring an abrupt external forcing, and it also provides a more complete synthesis of the state of the art (ie plenty of work negates the need for an abrupt external trigger for the YD (and DO events more generally) - as discussed earlier in the review process. As stated in the earlier review round, I am concerned about the general validity of a concept that calls exclusively upon external triggers for the YD, and DO events more generally (as is implied at times in your text).

**Response:** We have added text and a reference to Sima et al. 2004 in the introduction:

"Finally, still other research proposes that the YD resulted from internal oceanic processes, and that no external forcing was required to trigger the observed climate shifts (Sima et al., 2004)."

Of course you cannot have a pronounced cooling event unless the climate was warmer to begin with, and from this perspective D-O events (or any other event warmer than the glacial baseline) are critical. Thus, abundant literature does exist discussing internally forced D-O events, as discussed in the manuscript. We agree that there was cooling following on from the B-A, and that this cooling was probably largely internally forced. However, at 12.880 ka BP a very large number of records show an inflection point towards accelerated cooling (discussed in manuscript, and evident in many published figures, e.g., Fig. 1 in Buizert et al. 2014, *Science* and Fig. 2 in Bahr et al 2018 *Global and Planetary Change*). In most cases, the cooling into the YD is distinct from the preceding cooling (and sometimes warming) trend. In our manuscript, we are mostly discussing this conventionally defined YD (or GS-1) starting at 12.880 ka BP rather than the long-term cooling trend that started following peak B-A warming. We agree that this post-B-A cooling was probably internally forced, and this perspective is now discussed in more detail in the manuscript.

**Comment 5:** This is because there is a strong statistical link between the rate of cooling during preceding DO warm periods (for the YD, the BA) and their duration. This means that the approximate timing of the transition from DO interstadial to stadial is predictable. Theoretical concepts have been put forward to explain this relationship (from Schulz et al 2001 up to Buizert & Schmittner 2015, with many in between). It is relatively simple to incorporate a role for a common external forcing, like certain types of volcanic eruption, into these models, such that the general timing of the transition is determined by the evolving state/stability of components of the climate system, but the precise timing (ie why 100 years earlier not later etc) is affected by the occurrence of eg volcanic eruptions. What is difficult to reconcile would be a more dominant controlling role on the timing of DO stadials by volcanic events and a much more extensive theoretical framework would need to be developed to incorporate all the observations and features of DO events and the YD-BA. The problem is perhaps best put as: "The approximate timing of the end of a DO interstadial is predictable once its rate of cooling is established. If the underlying cause for this transition is invoked to be a volcanic eruption, it logically follows that the approximate timing of the eruption can be predicted. Ie during MIS 3 we could use cooling of DO interstadials to predict when the next major eruption will occur....this would certainly need to be explained/discussed, because it is a very provocative inference"

**Response:** We feel that perhaps the editor misinterprets some of our discussion, possibly because of ambiguity inherent in our text. The editorial comment states:

"The approximate timing of the end of a DO interstadial is predictable once its rate of cooling is established. If the underlying cause for this transition is invoked to be a volcanic eruption, it logically follows that the approximate timing of the eruption can be predicted. Ie during MIS 3 we could use cooling of DO interstadials to predict when the next major eruption will occur....this would certainly need to be explained/discussed, because it is a very provocative inference."

We do not believe that we can predict when a volcanic eruption may occur based on the characteristics of a D-O event. The end of D-O warming is to a certain extent predictable, but considerable uncertainties do exist, and a broad range of D-O cooling durations are possible for any given D-O warming, even after the cooling rate is established (see Figure 2 in Ganopolski and Rahmstorf, 2001, for example). We are not arguing that a volcanic eruption, like the LSE, immediately ends a D-O event. Rather, post D-O cooling (probably internally forced) occurs, permitting an estimation of the duration of that D-O warming event based on the initial cooling rate. However, any large, sulphur-rich northern hemisphere volcanic eruption increases the rate of cooling, and hastens the end of a D-O event. Clear inflection points exist during the cooling following several D-O temperature maxima (e.g., GI-1, GI-20, etc.). In the case of the transition into the YD, it is apparent from numerous studies that cooling did not occur at a constant rate following peak B-A warmth. We therefore argue that a volcanic eruption was responsible for some of the difference between the predicted time for the climate system to return to a glacial baseline state and the actual return time. We hope that the changes made during this most recent revision clarify these points.

**Comment, L1280:** Please may I suggest that you reword "The most widely accepted explanation for the YD" to "A common explanation for the YD". A minor change but there are many who do not accept freshwater pulses as the YD trigger. Technically fine as you have it though.

**Response:** This has been changed.

**Comment, L1618**: - You shouldn't rule out all ocean forcing just because AMOC might not weaken during the LIA (note - in addition to Rahmstorf et al 2015 showing no AMOC weakening during LIA, we have also recently shown no AMOC weakening during the LIA - Thornalley et al 2018, Nature). There are many ways the ocean can influence climate other than the AMOC, such as subpolar gyre circulation (as pointed out by the earlier suggested reference to Moreno-Chamarro et al 17, which, if you are discussing the LIA, I would still recommend including). Therefore change text to something along the lines of "related to sea-ice, atmosphere, or ocean changes other than the AMOC".

**Response:** We thank the editor for bringing their recent paper to our attention, and for highlighting the Moreno-Chamarro paper, and have now included the following text:

"This implies that the drivers of Little Ice Age cooling were related to sea-ice, atmosphere, or oceanic (other than the AMOC, e.g., horizontal subpolar gyre circulation (Moreno-Chamarro et al., 2017)) changes, although this requires further research to confirm."

**Comment, L1895:** Please alter this text because its underlying inference is incorrect: an abrupt cooling does not require an abrupt forcing. The fact that there is gradual cooling and then a switch to a more abrupt cooling at the onset of the YD does not imply that the gradual cooling had one cause, and then the more abrupt stage, another. As shown by many model studies, climate and the AMOC can display non-linear behaviour and tipping points, for which DO events and the BA-YD have been studied in this context. There can be gradual cooling, and then a threshold is crossed in which the climate can shift to another mode of operation. Ocean convection in the North Atlantic - weakening in the Nordic Seas then switching to south of Iceland - in relation to DO events is often thought of as a classic example of this (eg Ganopolski and Rahmstorf 2001 Nature).

**Response:** There are uncertainties associated with predicting the end of a D-O warming event; these uncertainties may be due to thresholds being crossed (as noted in this comment and many published papers) or a volcanic eruption (as we and others, e.g. Polyak et al., 2017 Geology, propose). We agree that there is more than one way of increasing the cooling rate, and have now clarified this in the text:

"However, the gradual cooling after the B-A differs from the rapid cooling observed at around 12.9 ka BP clearly visible in the NGRIP d18O and nitrogen isotope data as well as in numerous other North Atlantic records, raising the possibility that the LSE (or other external forcing) expedited the final cooling into the YD. It is also possible however that the increased cooling rate reflects non-linearity inherent in Atlantic oceanic circulation, potentially linked to transitions across key thresholds between different modes of oceanic circulation (e.g., Ganopolski and Rahmstorf, 2001)."

**Comment, L1903**: Your summary of why some terminations do not have a YD needs revising/updating so that it includes the concept that the anomaly in some terminations is not the YD-like events but rather the BA-type events. As described by Cheng et al 2009 (Science), but also in other earlier studies (see Carlson 2008, QSR, and refs therein), some terminations are characterised by a very long (Heinrich) stadial period, and then transition into interglacial. These terminations have strong insolation forcing that is thought to keep AMOC suppressed throughout the deglacial, possibly due to continued meltwater input. Other terminations (eg T1 and 3), where insolation forcing is weaker, enable a brief recovery of AMOC midway through deglaciation, in the form of a transient BA type event. Therefore, in addition to your speculation that whether a YD events occurs during a termination or not depends on if a volcanic eruption occurs during an intermediate climate state, you should also discuss the widely supported suggestion that the differing terminations are due to the occurrence (or lack)

of a BA-type event interrupting the (Heinrich) stadial event that is thought to be integral to a termination (e.g. Wolff et al 2009 Nat Geo., Barker et al 2010 and 2011).

**Response:** We thank the editor for highlighting this, and we agree that it is important to include some additional/revised text. We have now included the text:

"Additionally, the absence of a YD-type event during some terminations may simply reflect the absence of preceding B-A-type event and the presence of a very long stadial interval which then transitioned directly into an interglacial (Cheng et al., 2009; Carlson, 2008). Terminations lacking a YD-type event tend to have a strong insolation forcing that may have suppressed AMOC throughout the termination, preventing the development of a B-A-type event and decreasing the possibility of a YD-type event following after peak B-A-type warming (Wolff et al., 2010; Barker et al., 2011; Barker et al., 2010)."

**e-evaluating the link between the sulphur-rich Laacher See volcanic eruption and the Younger Dryas climate anomaly**

James U.L. Baldini[1], Richard J. Brown[1], and Natasha Mawdsley[1]
[1]Department of Earth Sciences, University of Durham, Durham, DH1 3LE, UK.

*Correspondence to*: James Baldini (james.baldini@durham.ac.uk)

**Abstract.** The Younger Dryas  is considered the  archetypal  millennial-scale  climate change event , and identifying its cause is fundamental for thoroughly understanding  climate systematics during deglaciations. However, the  mechanisms responsible for its initiation remain elusive, and both of the  most researched triggers (a meltwater pulse or a bolide impact) are controversial. Here we consider the problem from a different perspective, and explore a hypothesis that Younger Dryas climate shifts were catalysed by the unusually sulphur-rich 12.880 ka BP eruption of the Laacher See volcano (Germany). We use a recent chronology for GISP2 ice core ion dataset from the Greenland ice sheet to identify a large volcanic sulphur spike coincident with both the  Laacher See  eruption and the onset of Younger Dryas-related cooling in Greenland (i.e., the most recent abrupt Greenland millennial-scale cooling event, Greenland Stadial-1; 'GS-1'). Previously published lake sediment and stalagmite records confirm that the eruption's timing was indistinguishable from the onset of cooling across the North Atlantic, but that it preceded westerly wind repositioning over central Europe by ~200 years. We suggest that the initial short-lived volcanic sulphate aerosol cooling was amplified by ocean circulation shifts and/or sea ice expansion, gradually cooling the North Atlantic region and incrementally shifting the mid-latitude westerlies to the south. The aerosol-related cooling probably only lasted 1-3 years, and  the majority of Younger Dryas-related cooling may have been  due to the sea ice-ocean circulation positive feedback, which was particularly effective during the intermediate ice volume conditions characteristic of ~13 ka BP.  Furthermore, the eruption itself may have resulted from lithospheric unloading associated with the last deglaciation, and therefore similar volcanic forcings may not be spatiotemporally random, but instead may represent an integral part of deglaciation that often leads to YD-type events. 
[revised manuscript text omitted]

---

## Author Response (AR3)

Shaped by the past, creating the future

Dr James U.L. Baldini
Associate Professor (Reader)
Department of Earth Sciences
University of Durham
Durham DH1 3LE, UK
+44 (0) 191 334 2334
james.baldini@durham.ac.uk

12 June 2018

Dear Dr Thornalley,

Thank you for your recent comments regarding our manuscript entitled 'Evaluating the link between the sulphur-rich Laacher See volcanic eruption and the Younger Dryas climate anomaly' for publication in *Climate of the Past*. We have now made the two small revisions requested to the abstract. These revisions are marked in the manuscript with tracked changes below. We have also added to the acknowledgements, but have otherwise made no other changes to the manuscript.

Please do let us know if you have any other questions regarding our revised resubmission, and thank you once again for your editorial handling of this manuscript.

Best regards,

Dr James Baldini

(Corresponding author)

[revised manuscript text omitted]